# DUAL-OBJECTIVE REINFORCEMENT LEARNING WITH NOVEL HAMILTON-JACOBI-BELLMAN FORMULATIONS

**William Sharpless**\*, **Dylan Hirsch**\*, **Sander Tonkens**, **Nikhil Shinde**, **Sylvia Herbert**
University of California, San Diego, \*equal contribution, site

## ABSTRACT

Hard constraints in reinforcement learning (RL) often degrade policy performance. Lagrangian methods offer a way to blend objectives with constraints, but require intricate reward engineering and parameter tuning. In this work, we extend recent advances that connect Hamilton-Jacobi-Bellman (HJB) equations with RL to propose two novel value functions for dual-objective satisfaction. Namely, we address: 1) the **Reach-Always-Avoid** (RAA) problem – of achieving distinct reward and penalty thresholds – and 2) the **Reach-Reach** (RR) problem – of achieving thresholds of two distinct rewards. In contrast with temporal logic approaches, which typically involve representing an automaton, we derive explicit, tractable Bellman forms in this context via decomposition. Specifically, we prove that the RAA and RR problems may be rewritten as compositions of previously studied HJ-RL problems. We leverage our analysis to propose a variation of Proximal Policy Optimization (**DOHJ-PPO**), and demonstrate that it produces distinct behaviors from previous approaches, out-competing a number of baselines in success, safety and speed across a range of tasks for safe-arrival and multi-target achievement.

## 1 INTRODUCTION

The development of special Bellman equations from the Hamilton-Jacobi (HJ) perspective of dynamic programming (DP) has illustrated a novel route to safety and target-achievement in reinforcement learning (RL) Fisac et al. (2019); Hsu et al. (2021). In comparison with the canonical RL discounted-sum cost and corresponding additive DP update, these equations, namely the Safety Bellman Equation (SBE) and Reach-Avoid Bellman Equation (RABE), propagate the minimum (worst) penalty and maximum (best) reward, yielding a value function defined by the outlying performance of a trajectory. In mission-critical applications, where avoiding failure is a necessary condition, these equations have proved invaluable in the field of safe control Mitchell et al. (2005); Ames et al. (2016). By focusing on extremal values rather than discounted sums, the HJ-RL equations induce behaviors that act with respect to the best or worst outcomes in time-optimal fashions, performing far more safely than Lagrangian methods Ganai et al. (2023); So et al. (2024). Accordingly, these updates yield policies with significantly improved performance in target-achievement and obstacle-avoidance tasks over long horizons Yu et al. (2022a;b), relevant to fundamental and practical problems in many domains.

In this work, we advance the existing HJ-RL formulations by generalizing them to compositional problems. To date, the HJ-RL Bellman equations are limited to three operations: Reach (R), wherein the agent seeks to reach a goal (achieve a reward threshold), Avoid (A), wherein the agent seeks to avoid an obstacle (avoid a penalty threshold), and Reach-Avoid (RA), where the agent avoids obstacles *until* reaching the goal. In this light, we extend the HJ-RL Bellman equations to two complementary problems concerned with dual-satisfaction, namely the **Reach-Reach** (RR) problem for reaching two goals and the **Reach-Always-Avoid** (RAA) problem for continuing to avoid hazards after reaching a goal, demonstrated in Figure 1. We prove that the RAA and RR have a fundamental structure such that their Bellman equations may be decomposed into combinations of SBEs and RABEs. From this theory, we devise **DOHJ-PPO**, a novel algorithm for learning the RAA and RR values which bootstraps concurrently solved decompositions for coupling on-policy PPO roll-outs. Notably, this allows one to automatically learn to satisfy dual-objective tasks, for example, in the RAA, the F16 learns to fly into the desired airspace without crashing afterward (Figure 1, top middle-left), and in the RR case, the Hopper learns to jump into a target without diving so it may then achieve the second target (Figure 1, bottom left). The RAA and RR problems are distinct from both standard sum-of-reward values and the simpler HJ-RL formulations, providing new perspectives and performant tools for constrained decision-making.

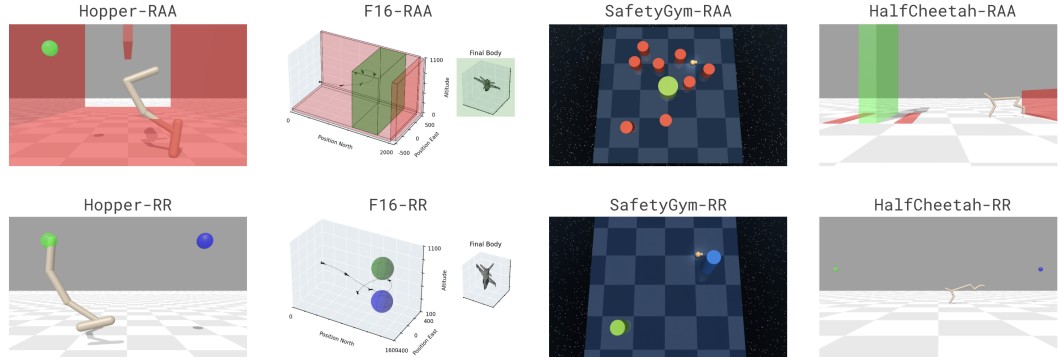

Figure 1: **Depiction of the Reach-Always-Avoid (RAA) and Reach-Reach (RR) Tasks.** In the RAA tasks, the zero-level set of the rewards (goals) and penalties (obstacles) are depicted in **green** and **red** respectively, while in the RR problem, the zero-level set of the two rewards (two goals) are depicted in **green** and **blue**. The RAA value is defined by the minimum of the minimum penalty and maximum reward, inducing the agents to enter the goals at some time without ever entering the obstacles. The RR value is defined by the minimum of the two maximum rewards, inducing the agents to enter both goals at some time.

**Our contributions include:**

1. We introduce novel value functions corresponding to the RAA and RR problems.

2. We prove that these value functions and their optimal policies can be decomposed into reach, avoid, and reach-avoid value functions (Theorems 1 and 2).

3. We demonstrate the nature of the RAA and RR values and their optimal policies in a simple grid-world example with deep $Q$-learning (DQN) (Figure 2).

4. We propose **DOHJ-PPO** to solve these value functions, which bootstraps concurrently solved decompositions for effectively coupling the on-policy rollouts (Section 7.2).

5. In continuous control tasks, we showcase that with *little to no* tuning, **DOHJ-PPO** is more successful, safer and faster than Lagrangian and existing HJ-RL baselines (Figure 4).

## 2 RELATED WORKS

This work involves aspects of safety (e.g. hazard avoidance), liveness (e.g. goal reaching), and balancing competing objectives. We summarize the relevant related works here.

**Constrained and Multi-Objective RL.** Constrained Markov decision processes (CMPDs) maximize the expected sum of discounted rewards subject to an expected sum of discounted costs, or an instantaneous safety violation function remaining below a set threshold Altman (2021); Achiam et al. (2017a); Wachi and Sui (2020). CMDPs are an effective way to incorporate state constraints into RL problems, and the efficient and accurate solution of the underlying optimization problem has been extensively researched, first by Lagrangian methods and later by an array of more sophisticated techniques Stooke et al. (2020); Li et al. (2024); Chen et al. (2021); Miryoosefi and Jin (2021); Yang et al. (2020). Multi-objective RL is an approach to designing policies that obtain *Pareto-optimal* expected sums of discounted *vector-valued* rewards Wiering et al. (2014); Van and Nowé (2014); Cai et al. (2023), including by deep-Q and other deep learning techniques Mossalam et al. (2016); Abels et al. (2019); Yang et al. (2019). By contrast, this work explicitly balances rewards and penalties in a way that does not require specifying a Lagrange multiplier or similar hyperparameter. Moreover, our work treats goal-reaching and hazard-avoidance as hard constraints, and the learned value function has a direct interpretation in terms of the constraint satisfaction.

**Goal-Conditioned RL (GCRL).** GCRL simultaneously learns optimal policies for a range of different (but typically related) tasks Liu et al. (2022); Plappert et al. (2018); Ren et al. (2019); Ma et al. (2022); Campero et al. (2020); Trott et al. (2019); Eysenbach et al. (2022); Ma et al. (2022); Campero et al. (2020). In GCRL, states are augmented with information on the current goal. While these goals are in their simplest form mostly independent, some work extends GCRL to more sophisticated composite tasks Chane-Sane et al. (2021). Our work primarily focuses on composing specific learned tasks rather than learning general tasks simultaneously.

**Linear Temporal Logic (LTL), Automatic State Augmentation, Automatons, and Generalized Objective Functions.** Many works have been explored that merge LTL and RL, canonically focused on Non-Markovian Reward Decision Processes (NMRDPs) Bacchus et al. (1996). Here, the reward gained at each time step may depend on the previous state history. Many of these works convert these NMRDPs to MDPs via state augmentation Bacchus et al. (1997); Thiebaux et al. (2006); Camacho et al. (2021); Icarte et al. (2018); Camacho et al. (2019). Often the augmented states are taken to be products between an ordinary state and an automaton state, where the automaton is used to determine "where" in the LTL specification an agent currently is. Other works using RL for LTL tasks involve MDP verification Brázdil et al. (2014), hybrid systems theory Cohen et al. (2023), GCRL with complex LTL tasks Qiu et al. (2023), almost-sure objective satisfaction Sadigh et al. (2014), incorporating (un)timed specifications Hamilton et al. (2022), and using truncated LTL Li et al. (2017). While the problems we attempt to solve (e.g. reaching multiple goals) can be thought of as specific instantiations of LTL specifications, our approach to solving these problems is fundamentally different from those in this line of work. Our state augmentation and subsequent decomposition of the problem are performed in a specific manner to leverage new HJ-based methods on the subproblems. Through our specific choice of state augmentation, we still prove that we can achieve an optimal policy in theory (and approximately so in practice) despite the non-NMRDP setup. There is also significant literature on generalized objective functions in RL Wang et al. (2020); Cui and Yu (2023); Tang et al. (2025), but these works are either not able to or are not tailored to simultaneously handle multiple rewards/penalties in the context of safe optimal control, which is where our decompositional approach becomes useful. On the other hand, works that do try to handle multiple rewards and penalties (including by decomposition) still use discounted-sum-of-rewards objectives van Seijen et al. (2017); Pitis (2023); Lin et al. (2020).

**Hamilton-Jacobi (HJ) Methods.** HJ is a dynamic programming-based framework for solving reach, avoid, and reach-avoid tasks Mitchell et al. (2005); Fisac et al. (2015). The value functions used in HJ have the advantage of directly specifying desired behavior, so that a positive value corresponds to task achievement and a negative value corresponds to task failure. Recent works use RL to find corresponding optimal policies by leveraging the unconventional Bellman updates associated with these value functions So et al. (2024); Hsu et al. (2021); Fisac et al. (2019). We build on these works by extending these advancements to more complex tasks, superficially mirroring the progression from MDPs to NMRDPs in the LTL-RL literature. Additional works merge HJ and RL, but do not concern themselves with such composite tasks Ganai et al. (2023); Yu et al. (2022a); Zhu et al. (2024).

## 3 Problem Definition

Consider a Markov decision process (MDP) $\mathcal{M} = \langle \mathcal{S}, \mathcal{A}, f \rangle$ consisting of finite state and action spaces $\mathcal{S}$ and $\mathcal{A}$, and *unknown* discrete dynamics $f$ that define the deterministic transition $s_{t+1} = f(s_t, a_t)$. Let an agent interact with the MDP by selecting an action with policy $\pi : \mathcal{S} \to \mathcal{A}$ to yield a state trajectory $s_t^\pi$, i.e. $s_{t+1}^\pi = f(s_t^\pi, \pi(s_t^\pi))$.

In this work, we consider the **Reach-Always-Avoid** (RAA) and **Reach-Reach** (RR) problems, which both involve the composition of two objectives, which are each specified in terms of the best reward and worst penalty encountered over time. In the RAA problem, let $r, p : \mathcal{S} \to \mathbb{R}$ represent a reward to be maximized and a penalty to be minimized. We will let $q = -p$ for mathematical convenience, hence, our aim to minimize the largest-over-time (worst) penalty $p$ becomes the aim to maximize the smallest-over-time $q$. In the RR problem, let $r_1, r_2 : \mathcal{S} \to \mathbb{R}$ be two distinct rewards to be maximized. The agent's overall objective is to maximize the *worst-case* outcome between the best-over-time reward and worst-over-time penalty (in RAA) and the two best-over-time rewards (in RR), i.e.

$$(\text{RAA}) \begin{cases} \text{maximize (w.r.t. } \pi) & \min\left\{ \max_t r(s_t^\pi), \ \min_t q(s_t^\pi) \right\} \\ \quad\quad\quad \text{s.t.} & s_{t+1}^\pi = f(s_t^\pi, \pi(s_t^\pi)), \\ & s_0^\pi = s, \end{cases}$$

$$(\text{RR}) \begin{cases} \text{maximize (w.r.t. } \pi) & \min\left\{ \max_t r_1(s_t^\pi), \ \max_t r_2(s_t^\pi) \right\} \\ \quad\quad\quad \text{s.t.} & s_{t+1}^\pi = f(s_t^\pi, \pi(s_t^\pi)), \\ & s_0^\pi = s. \end{cases}$$

As the names suggest, these optimization problems are inspired by — but not limited to — tasks involving goal reaching and hazard avoidance. More specifically, the RAA problem is motivated by a

Figure 2: **DQN Grid-World Demonstration of the RAA & RR Problems.** We compare our novel formulations with previous HJ-RL formulations (RA & R) in a simple grid-world problem with DQN. The zero-level sets of $q$ (*hazards*) are highlighted in **red**, those of $r$ (*goals*) in **blue**, and trajectories in **black** (starting at the bottom). In both models, the agents actions are limited to {left, right, straight} and the system flows upwards over time.

task in which an agent wishes to both reach a goal $\mathcal{G}$ and perennially avoid a hazard $\mathcal{H}$ (even after it reaches the goal). The RR problem is motivated by a task in which an agent wishes to reach two goals, $\mathcal{G}_1$ and $\mathcal{G}_2$, in either order. While these problems are thematically distinct, they are mathematically complementary (differing by a single $\max/\min$ operation), and hence we tackle them together.

The values for any policy in these problems then take the forms $V_{\text{RAA}}^{\pi}$ and $V_{\text{RR}}^{\pi}$,

$$V_{\text{RAA}}^{\pi}(s) = \min\left\{\max_t r(s_t^{\pi}),\ \min_t q(s_t^{\pi})\right\} \quad \text{and} \quad V_{\text{RR}}^{\pi}(s) = \min\left\{\max_t r_1(s_t^{\pi}),\ \max_t r_2(s_t^{\pi})\right\}.$$

One may observe that these values are fundamentally different from the infinite-sum value commonly employed in RL Sutton and Barto (2018), and do not accrue over the trajectory but, rather, are determined by certain points. Moreover, while each return considers two objectives, these objectives are combined in worst-case fashion to ensure *dual-satisfaction*. Although many of the related works discussed approach similar tasks (e.g. goal reaching and hazard avoidance) via traditional sum-of-discounted-rewards formulations, these novel value functions have a more direct interpretation in the following sense: if $r$ is positive (only) within $\mathcal{G}$ and $q$ is positive (only) inside $\mathcal{H}$, $V_{\text{RAA}}^{\pi}(s)$ will be positive if and only if the RAA task will be accomplished by the policy $\pi$. Similarly if $r_1$ and $r_2$ are positive within $\mathcal{G}_1$ and $\mathcal{G}_2$, respectively, $V_{\text{RR}}^{\pi}(s)$ will be positive if and only if the RR task will be accomplished by the policy $\pi$.

## 4 REACHABILITY AND AVOIDABILITY IN RL

The reach $V_{\text{R}}^{\pi}$, avoid $V_{\text{A}}^{\pi}$, and reach-avoid $V_{\text{RA}}^{\pi}$ values, respectively defined by

$$V_{\text{R}}^{\pi}(s) = \max_t r(s_t^{\pi}), \quad V_{\text{A}}^{\pi}(s) = \min_t q(s_t^{\pi}), \quad V_{\text{RA}}^{\pi}(s) = \max_t \min\left\{r(s_t^{\pi}), \min_{\tau \leq t} q(s_\tau^{\pi})\right\},$$

have been previously studied Fisac et al. (2019) leading to the derivation of special Bellman equations. To put these value functions in context, assume the goal $\mathcal{G}$ is the set of states for which $r(s)$ is positive and the hazard $\mathcal{H}$ is the set of states for which $q(s)$ is non-positive. See Figure 2 for a simple grid-world demonstration comparing the RAA and RR values with the previously existing RA and R values. Then $V_{\text{R}}^{\pi}$, $V_{\text{A}}^{\pi}$, and $V_{\text{RA}}^{\pi}$ are positive if and only if $\pi$ causes the agent to eventually reach $\mathcal{G}$, to always avoid $\mathcal{H}$, and to reach $\mathcal{G}$ without hitting $\mathcal{H}$ prior to the reach time, respectively. The Reach-Avoid Bellman Equation (RABE), for example, takes the form Hsu et al. (2021)

$$V_{\text{RA}}^{*}(s) = \min\left\{\max\left\{\max_{a \in \mathcal{A}} V_{\text{RA}}^{*}\left(f(s, a)\right), r(s)\right\}, q(s)\right\},$$

and is associated with optimal policy $\pi_{\text{RA}}^{*}(s)$ (without the need for state augmentation, see the appendix). This formulation does not naturally induce a contraction, but may be discounted to induce contraction by defining $V_{\text{RA}}^{\gamma}(z)$ implicitly via

$$V_{\text{RA}}^{\gamma}(s) = (1 - \gamma)\min\{r(s), q(s)\} + \gamma \min\left\{\max\left\{\max_{a \in \mathcal{A}} V_{\text{RA}}^{\gamma}\left(f(s, a)\right), r(s)\right\}, q(s)\right\},$$

## Reach-Reach

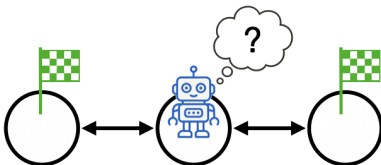

## Reach-Always-Avoid

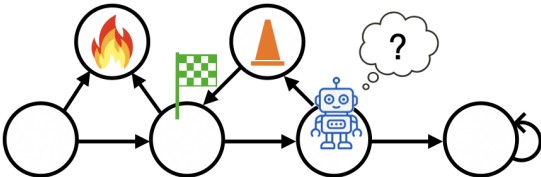

Figure 3: **Examples where a Non-Augmented Policy is Flawed.** In both MDPs, consider an agent with no memory. (Left) For a deterministic policy based on the current state, the agent can only achieve one target (RR), as this policy must associate the middle state with either of the two possible actions. (Right) The RAA case is slightly more complex. Assume the robot will make sure to avoid the fire at all costs (which is easily done from the current state). It would also prefer to not encounter the cone hazard, but will do so if needed to achieve the target. From its current state the robot cannot determine whether to pursue the target by crossing the cone or move to the right. The correct decision depends on state history, specifically on whether the robot has already reached the target state or not (e.g. imagine the initial state is on the target state).

for each $\gamma \in [0, 1)$. A fundamental result (Proposition 3 in Hsu et al. (2021)) is that

$$\lim_{\gamma \to 1} V_{\text{RA}}^{\gamma}(s) = V_{\text{RA}}(s).$$

These prior value functions and corresponding Bellman equations have proven powerful for these simple reach/avoid/reach-avoid problem formulations. In this work, we generalize the aforementioned results to the broader class involving $V_{\text{RAA}}$ (assure no penalty after the reward threshold is achieved) and $V_{\text{RR}}$ (achieve multiple rewards optimally). Through this generalization, we are able to train an agent to accomplish more complex tasks with noteworthy performance.

## 5 THE NEED FOR AUGMENTING STATES WITH HISTORICAL INFORMATION

We here discuss a small but important detail regarding the problem formulation. The value functions we introduce may appear similar to the simpler HJ-RL value functions discussed in the previous section; however, in these new formulations the goal of choosing a policy $\pi : \mathcal{S} \to \mathcal{A}$ is inherently flawed without state augmentation. In considering multiple objectives over an infinite horizon, situations arise in which the optimal action depends on more than the current state, but rather the **history** the trajectory. This complication is not unique to our problem formulation, but also occurs for NMDPs (see the Related Works section). To those unfamiliar with NMDPs, this at first may seem like a paradox as the MDP is by definition Markov, but the problem occurs not due to the state-transition dynamics but the nature of the reward. An example clarifying the issue is shown in Figure 3.

To allow the agent to use relevant aspects of its history, we will henceforth consider an augmentation of the MDP with auxiliary variables. A theoretical result in the next section states that this choice of augmentation is sufficient in that no additional information will be able to improve performance under the optimal policy. Note that the state augmentation is needed because of the use of HJR-style optimization objectives (rather than discounted sum-of-rewards). The point of the state augmentation is not to make the rewards and penalties Markovian (indeed, they are already Markovian as they are deterministic functions of the current state).

### 5.1 AUGMENTATION OF THE RAA PROBLEM

We consider an augmentation of the MDP defined by $\overline{\mathcal{M}} = \langle \overline{\mathcal{S}}, \mathcal{A}, f \rangle$ consisting of augmented states $\overline{\mathcal{S}} = \mathcal{S} \times \mathcal{Y} \times \mathcal{Z}$ and the same actions $\mathcal{A}$. For any initial state $s$, let the augmented states be initialized as $y = r(s)$ and $z = q(s)$, and let the transition of $\overline{\mathcal{M}}$ be defined by

$$s_{t+1}^{\bar{\pi}} = f\left(s_t^{\bar{\pi}}, \bar{\pi}\left(s_t^{\bar{\pi}}, y_t^{\bar{\pi}}, z_t^{\bar{\pi}}\right)\right); \quad y_{t+1}^{\bar{\pi}} = \max\left\{r\left(s_{t+1}^{\bar{\pi}}\right), y_t^{\bar{\pi}}\right\}; \quad z_{t+1}^{\bar{\pi}} = \min\left\{q\left(s_{t+1}^{\bar{\pi}}\right), z_t^{\bar{\pi}}\right\},$$

such that $y_t$ and $z_t$ track the best reward and worst penalty up to any point. Hence, the policy for $\overline{\mathcal{M}}$ given by $\bar{\pi} : \overline{\mathcal{S}} \to \mathcal{A}$ may now consider information regarding the history of the trajectory.

By definition, the RAA value for $\overline{\mathcal{M}}$,

$$V_{\text{RAA}}^{\bar{\pi}}(s) = \min \left\{ \max_t r(s_t^{\bar{\pi}}), \min_t q(s_t^{\bar{\pi}}) \right\},$$

is equivalent to that of $\mathcal{M}$ except that it allows for a policy $\bar{\pi}$ which has access to historical information. We seek to find $\bar{\pi}$ that maximizes this value.

## 5.2 AUGMENTATION OF THE RR PROBLEM

For the Reach-Reach problem, we augment the system similarly, except that $z_t$ is updated using a max operation instead of a min:

$$s_{t+1}^{\bar{\pi}} = f\left(s_t^{\bar{\pi}}, \bar{\pi}\left(s_t^{\bar{\pi}}, y_t^{\bar{\pi}}, z_t^{\bar{\pi}}\right)\right); \quad y_{t+1}^{\bar{\pi}} = \max\left\{r_1\left(s_{t+1}^{\bar{\pi}}\right), y_t^{\bar{\pi}}\right\}; \quad z_{t+1}^{\bar{\pi}} = \max\left\{r_2\left(s_{t+1}^{\bar{\pi}}\right), z_t^{\bar{\pi}}\right\}.$$

Again, by definition,

$$V_{\text{RR}}^{\bar{\pi}}(s) = \min\left\{\max_t r_1(s_t^{\bar{\pi}}), \max_t r_2(s_t^{\bar{\pi}})\right\}.$$

The RR problem is again to find an augmented policy $\bar{\pi}$ which maximizes this value.

# 6 OPTIMAL POLICIES FOR RAA AND RR BY VALUE DECOMPOSITION

We now discuss our first theoretical contributions. We refer the reader to the appendix for the proofs of the theorems.

## 6.1 DECOMPOSITION OF RAA INTO AVOID AND REACH-AVOID PROBLEMS

Our main theoretical result for the RAA problem shows that we can solve this problem by first solving the avoid problem corresponding to the penalty $q(s)$ to obtain the optimal value function $V_A^*(s)$ and then solving a reach-avoid problem with the negated penalty function $q(s)$ and a modified reward function $\tilde{r}_{\text{RAA}}(s)$.

**Theorem 1.** *Let $V_{\text{RAA}}^*(s) := \max_{\bar{\pi}} V_{\text{RAA}}^{\bar{\pi}}(s)$. For all initial states $s \in \mathcal{S}$,*

$$V_{\text{RAA}}^*(s) = \max_\pi \max_t \min\left\{\tilde{r}_{\text{RAA}}\left(s_t^\pi\right), \min_{\tau \leq t} q\left(s_\tau^\pi\right)\right\}, \tag{1}$$

*where $\tilde{r}_{\text{RAA}}(s) := \min\left\{r(s), V_A^*(s)\right\}$, with*

$$V_A^*(s) := \max_\pi \min_t q\left(s_t^\pi\right).$$

This decomposition is significant, as methods customized to solving avoid and reach-avoid problems were recently explored in Fisac et al. (2019); Hsu et al. (2021); So et al. (2024); So and Fan (2023), allowing us to effectively solve the optimization problem defining $V_A^*(s)$ as well as the optimization problem that defines the right-hand-side of 1.

**Corollary 1.** *The value function $V_{\text{RAA}}^*$ satisfies the Bellman equation*

$$V_{\text{RAA}}^*(s) = \min\left\{\max\left\{\max_{a \in \mathcal{A}} V_{\text{RAA}}^*\left(f(s,a)\right), \tilde{r}_{\text{RAA}}(s)\right\}, q(s)\right\}.$$

Readers familiar with temporal logic (TL) may be interested in how these decompositions relate to decompositions of predicates in TL. We discuss the distinction between these two classes of decompositions in Sec. L of the Appendix, and how the TL predicate algebra is insufficient for safe optimal control.

## 6.2 DECOMPOSITION OF THE RR PROBLEM INTO THREE REACH PROBLEMS

Our main result for the RR problem shows that we can solve this problem by first solving two reach problems corresponding to the rewards $r_1(s)$ and $r_2(s)$ to obtain reach value functions $V_{\text{R}_1}^*(s)$ and $V_{\text{R}_2}^*(s)$, respectively. We then solve a third reach problem with a modified reward $\tilde{r}_{\text{RR}}(s)$.

**Theorem 2.** *Let $V_{\text{RR}}^*(s) := \max_{\bar{\pi}} V_{\text{RR}}^{\bar{\pi}}(s)$. For all initial states $s \in \mathcal{S}$,*

$$V_{\text{RR}}^*(s) = \max_{\pi} \max_{t} \tilde{r}_{\text{RR}}\left(s_t^{\pi}\right), \tag{2}$$

*where $\tilde{r}_{\text{RR}}(s) := \max\left\{\min\left\{r_1(s), V_{\text{R}_2}^*(s)\right\}, \min\left\{r_2(s), V_{\text{R}_1}^*(s)\right\}\right\}$, with*

$$V_{\text{R}_1}^*(s) := \max_{\pi} \max_{t} r_1\left(s_t^{\pi}\right), \quad V_{\text{R}_2}^*(s) := \max_{\pi} \max_{t} r_2\left(s_t^{\pi}\right).$$

**Corollary 2.** *The value function $V_{\text{RR}}^*$ satisfies the Bellman equation*

$$V_{\text{RR}}^*(s) = \max\left\{\max_{a \in \mathcal{A}} V_{\text{RR}}^*\left(f(s,a)\right), \tilde{r}_{\text{RR}}(s)\right\}.$$

## 6.3 Optimality of the augmented problems

We previously motivated the choice to consider an augmented MDP $\overline{\mathcal{M}}$ over the original MDP in the context of the RAA and RR problems. In this section, we justify our particular choice of augmentation. Indeed, the following theoretical result shows that further augmenting the states with additional historical information cannot improve performance under the optimal policy.

**Theorem 3.** *Let $s \in \mathcal{S}$. Then*

$$\max_{\pi} V_{\text{RAA}}^{\pi}(s) \le \max_{\bar{\pi}} V_{\text{RAA}}^{\bar{\pi}}(s) = \max_{a_0,a_1,\dots} \min\left\{\max_{t} r(s_t), \min_{t} q(s_t)\right\},$$

*and*

$$\max_{\pi} V_{\text{RR}}^{\pi}(s) \le \max_{\bar{\pi}} V_{\text{RR}}^{\bar{\pi}}(s) = \max_{a_0,a_1,\dots} \min\left\{\max_{t} r_1(s_t), \max_{t} r_2(s_t)\right\}$$

*where $s_{t+1} = f(s_t, a_t)$ and $s_0 = s$.*

The terms on the right of the lines above reflect the best possible sequence of actions to solve the RAA or RR problem, and the theorem states that the optimal augmented policy achieves that value, represented by the middle terms. This value will generally be less than or equal to the outcome from using a non-augmented policy, represented by the terms on the left.

# 7 DOHJ-PPO: Solving RAA and RR with RL

In the previous sections, we demonstrated that the RAA and RR problems can be solved through decomposition of the values into formulations amenable to existing RL methods. However, we make a few assumptions in the derivation that would limit performance and generalization, namely, the determinism of the values as well as access to the decomposed values (by solving them beforehand). In this section, we propose relaxations to the RR and RAA theory and devise a custom variant of Proximal Policy Optimization, **DOHJ-PPO**, to solve this broader class of problems, and demonstrate its performance.

## 7.1 Stochastic Reach-Avoid Bellman Equation

It is well known that the most performative RL methods allow for stochastic learning. In So et al. (2024), the Stochastic Reachability Bellman Equation (SRBE) is described for Reach problems and used to design a specialized PPO algorithm. We first generalize this notion to a Stochastic Reach-Avoid Bellman Equation (SRABE). Using Theorems 1 and 2, the SRBE and SRABE offer the necessary tools for designing a PPO variant for solving the RR and RAA problems.

By anology to the SRBE, the SRABE is given by

$$\hat{V}_{\text{RAA}}^{\pi}(s) = \mathbb{E}_{a \sim \pi}\left[\min\left\{\max\left\{\hat{V}_{\text{RAA}}^{\pi}\left(f(s,a)\right), \tilde{r}_{\text{RAA}}(s)\right\}, q(s)\right\}\right]. \tag{SRABE}$$

More rigorously, we actually consider the discounted SRABE, which is contractive, and the corresponding quality function below in the limit $\gamma \to 1^-$ (as in Hsu et al. (2021)),

$$\hat{V}_{\text{RAA}}^{\gamma,\pi}(s) = (1-\gamma)\min\left\{\tilde{r}_{\text{RAA}}(s), q(s)\right\} + \gamma\mathbb{E}_{a \sim \pi}\left[\min\left\{\max\left\{\hat{V}_{\text{RAA}}^{\gamma,\pi}\left(f(s,a)\right), \tilde{r}_{\text{RAA}}(s)\right\}, q(s)\right\}\right].$$

$$\hat{Q}_{\text{RAA}}^{\gamma,\pi}(s,a) = (1-\gamma)\min\left\{\tilde{r}_{\text{RAA}}(s), q(s)\right\} + \gamma\min\left\{\max\left\{\hat{V}_{\text{RAA}}^{\gamma,\pi}\left(f(s,a)\right), \tilde{r}_{\text{RAA}}(s)\right\}, q(s)\right\}.$$

Theoretically speaking, the use of the SRABE is justified by Theorem 4 in the Appendix. With this action-value function we then follow So et al. to derive the corresponding policy gradient result with an augmented version of the dynamics; for details, see Prop. 1 in the Appendix. The PPO advantage function is then given by $\hat{A}_{\text{RAA}}^{\pi} = \hat{Q}_{\text{RAA}} - \hat{V}_{\text{RAA}}$ Schulman et al. (2017). Although, this approximation may be poor in highly noisy settings, we show this approach yields conservative estimates of the value with stochastic policies (Appendix sec. D), and validate it empirically with stochastic dynamics in Sec. 8.3.

## 7.2 Algorithm

We introduce **DOHJ-PPO** for solving the RAA and RR problems, which integrates the SRABE and SRBE via three minimal modifications to PPO Schulman et al. (2017) (see appendix for more).

**Additional actor and critics are introduced to represent the decomposed objectives.** Per Theorems 1 and 2, one may know that the RAA and RR values are given by a composition of the simpler R, A and RA values. Therefore, we learn these decompositions with their own networks and integrate them into the composed actor and critic training, namely via the GAE and target with the special RAA and RR reward functions in Theorems 1 and 2.

**The composed actor and critic are learned concurrently to the decomposed actor and critics by bootstrapping the current values.** Rather than learning the decomposed and composed representations sequentially, DOHJ-PPO bootstraps to learn them simultaneously. Namely, at each iteration, we rollout trajectories for composed and decomposed updates with each actor. In the update of the composed representation specifically, the decomposed values are inferred from the current decomposed critic(s) along the composed trajectories. This design choice allows us to couple the on-policy learning of PPO in the following way.

**Trajectories for training the decomposed actor and critic(s) are initialized with states sampled from the composed trajectories**, which we refer to as *coupled resets*. While it is possible to estimate the decomposed objectives independently—i.e., prior to solving the composed task—this approach might lead to inaccurate or irrelevant value estimates in on-policy settings. For example, in the RAA problem, the avoid decomposition will solely prioritize avoiding penalties and, hence, might converge to an optimal strategy within a reward-irrelevant region, misaligned with the overall task.

## 8 Experiments

### 8.1 DQN Demonstration

We begin by demonstrating the utility of our theoretical results (Theorems 1 and 2) through a simple 2D grid-world experiment using DQN (Figure 2). In this environment, the agent can move left, right, or remain stationary, while drifting upward at a constant rate. Throughout, reward regions are shown in blue and penalty regions in red. On the left, we compare the optimal value functions learned under the classic Reach-Avoid (RA) formulation with those from the Reach-Always-Avoid (RAA) setting. In the RA scenario, trajectories successfully avoid the obstacle but may terminate in regions from which future collisions are inevitable, as there is no incentive to consider what happens after reaching the minimum reward threshold. In contrast, under the RAA formulation, where the objective involves maximizing cumulative reward while accounting for future penalties (as per Theorem 1), the agent learns to reach the target while remaining in safe regions thereafter. On the right, we consider a similar environment without obstacles but with two distinct targets. Here, the Reach-Reach (RR) formulation induces trajectories that visit both targets, unlike simple reach tasks in which the agent halts after reaching a single goal. These qualitative results highlight the behavioral distinctions induced by the RAA and RR objectives compared to their simpler counterparts. Additional algorithmic and experimental details are provided in the Apendix.

### 8.2 Continuous Control Tasks with **DOHJ-PPO**

To evaluate the method under more complex and less structured conditions, we extend our analysis to continuous control settings. Specifically, we consider RAA and RR tasks in the `Hopper`, `F16`, `SafetyGym`, and `HalfCheetah` environments, depicted in Figure 1. In the RAA tasks, the penalty

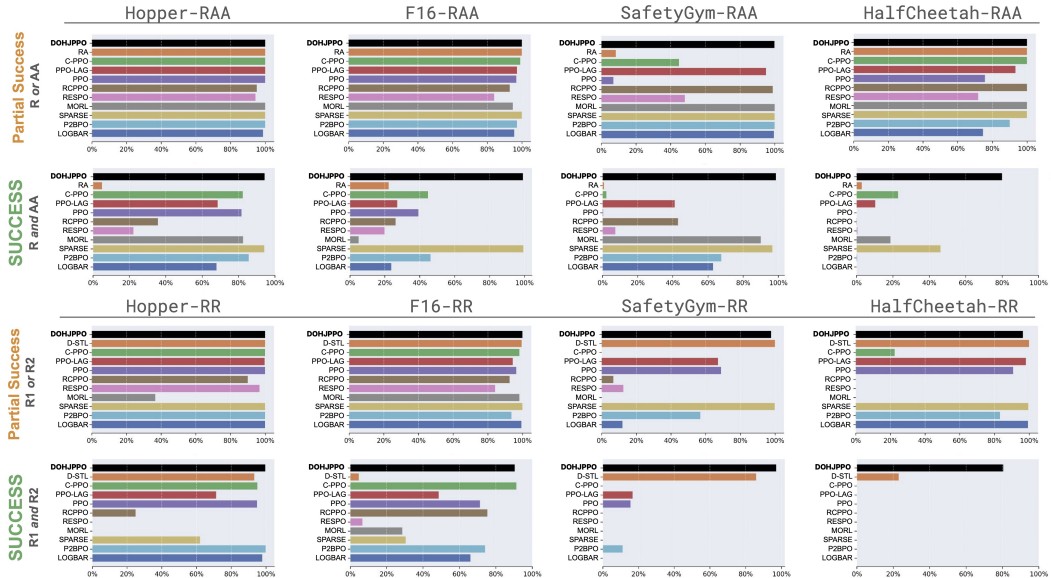

Figure 4: **Success (→) and Partial Success (→) in RAA and RR Tasks for DOHJ-PPO and Baselines.** We evaluate **DOHJ-PPO in black** against baselines over 1,000 trajectories in the `Hopper`, `F16`, `SafetyGym` and `HalfCheetah` environments. In the first and third row, the **Partial Success** percentage of each algorithm is given, defined by the number of trajectories to achieve one objective (reaching or always-avoiding in the RAA, reaching either in the RR). In the second and fourth rows, **SUCCESS** percentage is given, defined by the number of trajectories to achieve both objectives. Most baselines achieve partial success, however, few achieve total success as the environment becomes more difficult, underscoring the difficulty of balancing objectives in RL.

function generally characterizes regions of states where the agent (or its body parts) is intended to avoid, while the reward characterizes regions of states where the agent is intended to reach.

As baselines, we compare **DOHJ-PPO** against a variety of classes of RL algorithms. We include several augmented Lagrangian methods which transform constraints (either for reaching both or always avoiding) into mixed objectives, such as Constrained PPO (CPPO) Achiam et al. (2017b), PPO-LAG Ray et al. (2019), P2BPO Dey et al. (2024), and LOGBAR Zhang et al. (2024). Additionally, we include three HJ-RL baselines designed for the previous R and RA problems, RESPO Ganai et al. (2023), RCPPO So et al. (2024) and RA Hsu et al. (2021). Lastly, we also include a few methods based on approaches in STL/LTL-RL and MORL, including a decomposed STL (D-STL) PPO, a sparse-reward STL PPO (SPARSE) and a MORL-based PPO. All algorithms are trained on random initial conditions and then evaluated on new random initial conditions within distribution. To quantify performance of the dual-objective tasks, we measure (1) the percent of trajectories which achieve at least both tasks successfully, (2) the percent of trajectories which achieve at least one of the tasks (dubbed partial success), and (3) the mean steps in each trajectory until success.

Empirically, we find that our method performs at the top-level, achieving **first or second place** among all tasks and environments (Figure 4). In fact, for the multi-target (RR) or safe-achievement (RAA) **as dynamics become more complex, our algorithm increasingly dominates** the 10 state-of-the-art baselines (e.g. the `HalfCheetah`). Note, that almost all algorithms can achieve partial success at a high rate in each dual-objective task, highlighting the difficulty of mixed or competing objectives, particularly with discounted-sum rewards. Moreover, DOHJ-PPO is the **sole performant algorithm in both** RAA and RR tasks, displaying the fastest achievement times across tasks (see appendix).

These results underscore the challenging nature of composing multiple satisfaction objectives using traditional baselines with discounted-sum rewards. In contrast, DOHJ-PPO provides a direct and robust solution to handling these complex tasks, **with little to no tuning**. Our algorithm enjoys these benefits because of the structure of the novel Bellman updates, which propagate the extreme (maximum and minimum) values as opposed to the short-term average (discounted-sum) values.

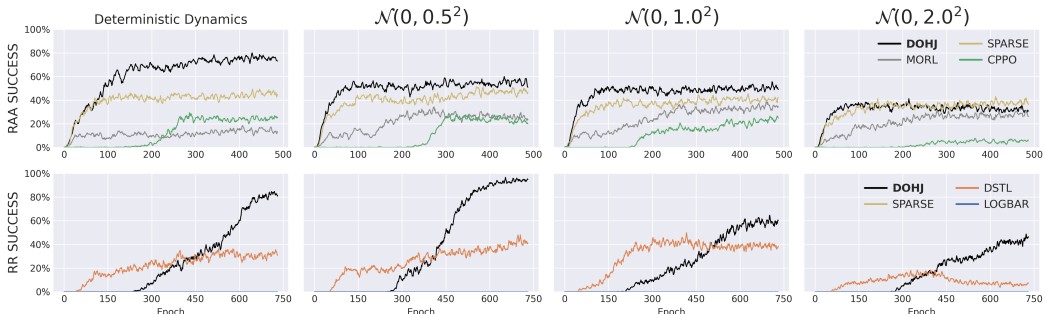

Figure 5: **Success (↑) in the** `HalfCheetah` **RAA and RR Tasks with Increasingly Stochastic Dynamics.** We plot the learning trajectories of **DOHJ-PPO in black** and the top baselines for the `HalfCheetah` environment with an affine Gaussian noise added to the dynamics. Task achievement (success) is given by the percentage of 256 trajectories that either reach the target and always-avoid the obstacles or reach both targets (corresponding to $V_{\text{RAA}} > 0$ and $V_{\text{RR}} > 0$). Each column corresponds to a different scale of noise – null, low (0.5), moderate (1.) and high (2.) – which is added to the velocities and angular velocities of the `HalfCheetah` dynamics. In the RAA task, DOHJ-PPO outperforms all baselines up to the highest noise settings where all algorithms perform equivalently poorly. In the RR task, DOHJ-PPO outperforms all algorithms significantly. In summary, this ablation demonstrates the robustness of DOHJ-PPO to certain stochasticity in the dynamics and the validity of the SRBE and SRABE approximations.

### 8.3 COMPARISON IN STOCHASTIC DYNAMICS

To design an algorithm robust to randomness, **DOHJ-PPO** employs the SRBE and SRABE discussed in Sec. 7.1 in place of their analogous deterministic forms. This choice equates to an approximation of the decompositional results (Thms. 1 and 2) that interchanges the extrema and expectation operators. The empirical results in Fig. 4 justify this approximation with stochastic policies, however, this noise is introduced for exploration and ultimately attenuated in training. To interrogate the behavior of DOHJ-PPO with stochastic dynamics, we inject affine Gaussian noise into the evolution of the `HalfCheetah` dynamics for both RAA and RR tasks. Note, only the velocities and angular velocities of the agent are perturbed to protect contact physics. We compare our algorithm against the top three baselines in each task along a scale of standard deviations of low (0.5), moderate (1.) and high (2.) quantity, plotting the maximum learning curve over three seeds in Fig. 5.

In this ablation, we find that the proposed approach, using the novel Bellman equations with stochastic relaxations, offers a significant performance improvement even in the face of significant noise. In the RAA task, DOHJ-PPO dominates the top performing baselines with a 8%-22% peak-performance gap between it and the second best algorithm (and is the fastest to peak-performance) for moderate noise levels, beyond which all algorithms perform equally poorly. In the RR case, we find an even starker result, with all but one experiment demonstrating a >30% improvement in peak-performance even in the high-noise regimes, with an exception of the moderate noise case where DOHJ-PPO still performs >15% than the best baseline, DSTL. Interestingly, DOHJ-PPO is slower than DSTL to peak performance, but performs twice as well at best in three of the four settings. These results demonstrate that despite certain highly-noisy dynamics DOHJ-PPO is competitive at worst and optimal in majority. See Appendix Sec. D for further analysis and discussion of the usage of the SRBE and SRABE approximations.

## 9 CONCLUSIONS

In this work, we introduced two novel Bellman formulations for new problems (RAA and RR) which generalize those considered in several recent publications. We derive decomposition results to break them into simpler Bellman equations, which can then be composed to obtain the corresponding value functions and optimal policies. We use these results to design DOHJ-PPO, which shows to be the most performant and balanced algorithm in safe-arrival and multi-target achievement. DOHJ-PPO employs the stochastic relaxations of the simpler Bellman equations (the SRBE and SRABE), for which we offer rigorous justification and empirical validation in the case of stochastic policies. As expectation and extrema operations do not commute, more work is needed to provide guarantees under stochastic dynamics. Nonetheless, we demonstrate through an artificial ablation that DOHJ-

PPO can be successful in the face of certain dynamic randomness. With regard to more complex objectives, it appears one might employ our results to iteratively decompose layered objectives corresponding to temporal logic specifications into a graph of Bellman values. However, doing so would require deriving generalized decomposition principles for nontrivial compositions of logical operations. Moreover, a practical algorithm for solving the decomposed graph of values might benefit from a more efficient representation, mechanisms to guarantee convergence, and heuristics to improve sampling efficiency, but we leave this to future work. By solving the RAA and RR values, this work provides a road-map to extend complex Bellman formulations, via decomposing higher-level problems into lower-level ones, establishing a foundation for nuanced tasks in real-world environments and safe RL.

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

## 10  ETHICS STATEMENT

This project was conducted and completed on entirely responsible and ethical grounds, and meets the highest standard of the ICLR Code of Ethics. We believe the rigor, investigation and communication not only upholds scientific ideals whilst avoiding societal harm, but advances machine learning for the betterment of all society, namely by improving learning to be more performant with much less hyper-tuning, and thus better for the planet and human race. Moreover, the work fundamentally improves the safety and reliability of reinforcement learning, and thus greatly improves a society in which machine learning is heavily integrated. Above all, the work is honest, noting limitations and caveats, while depicting the strengths we believe make this work invaluable for the field.

## 11  REPRODUCIBILITY STATEMENT

All theorems, algorithms and parameters for this work are totally explained in this paper (partially in the appendix) in what the authors believe to be a clear and understandable form. All theorems have been proven in detail, including all necessary lemmas, propositions and references in a manner the authors believe is intelligible. The algorithm proposed in the work is explained clearly in the main text and written line-by-line in the appendix, along with all hyper-parameters for each environment. The code for this work has been inherited from another group with security clearance and we are awaiting their response to publicize it, but will do so as soon as possible as we are committed to fair and open resources without bias or discrimination.

# Appendix

## Contents

## ACHIEVEMENT SPEED RESULTS FROM DOHJ-PPO EXPERIMENTS

Here we present additional results for RAA and RR problems solved with **DOHJ-PPO**. In both settings, DOHJ-PPO out-performs or matches the best of baselines with less tuning and faster arrival. Notably as the difficulty of the problem increases the gap increases significantly with DOHJ-PPO remaining the sole algorithm that can achieve the task in reasonable time and in both RAA and RR categories.

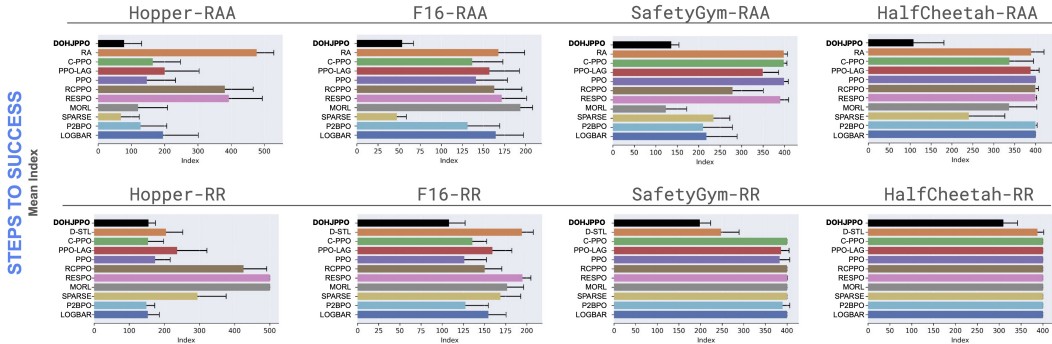

Figure 6: **Steps to Success ($\leftarrow$) in RAA and RR Tasks for DOHJ-PPO and Baselines** For the same 1000 trajectories in Figure 4, we quantify here the number of steps until achievement of both tasks: reaching without crash afterward in the RAA, reaching both goal in the RR. DOHJ-PPO is not only competitive but consistently achieves the dual-objective problems in the fewest number of steps.

PROOF NOTATION

Throughout the theoretical sections of this supplement, we use the following notation.

We let $\mathbb{N} = \{0, 1, \dots\}$ be the set of whole numbers.

We let $\mathbb{A}$ be the set of maps from $\mathbb{N}$ to $\mathcal{A}$. In other words, $\mathbb{A}$ is the set of sequences of actions the agent can choose. Given $\mathbf{a}_1, \mathbf{a}_2 \in \mathbb{A}$, and $\tau \in \mathbb{N}$, we let $[\mathbf{a}_1, \mathbf{a}_2]_\tau$ be the element of $\mathbb{A}$ for which

$$[\mathbf{a}_1, \mathbf{a}_2]_\tau(t) = \begin{cases} \mathbf{a}_1(t) & t < \tau, \\ \mathbf{a}_2(t - \tau) & t \geq \tau. \end{cases}$$

Similarly, given $a \in \mathcal{A}$ and $\mathbf{a} \in \mathbb{A}$, we let $[a, \mathbf{a}]$ be the element of $\mathbb{A}$ for which

$$[a, \mathbf{a}](t) = \begin{cases} a & t = 0, \\ \mathbf{a}(t - 1) & t \geq 1. \end{cases}$$

Additionally, given $\mathbf{a} \in \mathbb{A}$ and $\tau \in \mathbb{N}$, we let $\mathbf{a}|_\tau$ be the element of $\mathbb{A}$ for which

$$\mathbf{a}|_\tau(t) = \mathbf{a}(t + \tau) \quad \forall t \in \mathbb{N}.$$

The $[\cdot, \cdot]_\tau$ operation corresponds to concatenating two action sequences (using only the $0^{\text{th}}$ to $(\tau - 1)^{\text{st}}$ elements of the first sequence), the $[\cdot, \cdot]$ operation corresponds to prepending an action to an action sequence, and the $\cdot|_\tau$ operation corresponds to removing the $0^{\text{th}}$ to $(\tau - 1)^{\text{st}}$ elements of an action sequence.

We let $\Pi$ be the set of policies $\pi : \mathcal{S} \to \mathcal{A}$. Given $s \in \mathcal{S}$ and $\pi \in \Pi$, we let $\xi_s^\pi : \mathbb{N} \to \mathcal{S}$ be the solution of the evolution equation

$$\xi_s^\pi(t + 1) = f\left(\xi_s^\pi(t), \pi\left(\xi_s^\pi(t)\right)\right)$$

for which $\xi_s^\pi(0) = s$. In other words, $\xi_s^\pi(\cdot)$ is the state trajectory over time when the agent begins at state $s$ and follows policy $\pi$.

We will also "overload" this trajectory notation for signals rather than policies: given $\mathbf{a} \in \mathbb{A}$, we let $\xi_s^{\mathbf{a}} : \mathbb{N} \to \mathcal{S}$ be the solution of the evolution equation

$$\xi_s^{\mathbf{a}}(t + 1) = f\left(\xi_s^{\mathbf{a}}(t), \mathbf{a}(t)\right)$$

for which $\xi_s^{\mathbf{a}}(0) = s$. In other words, $\xi_s^{\mathbf{a}}(\cdot)$ is the state trajectory over time when the agent begins at state $s$ and follows action sequence $\mathbf{a}$.

# A  PROOF OF RAA MAIN THEOREM

We first define the value functions, $V_{\text{A}}^*, \tilde{V}_{\text{RA}}^*, V_{\text{RAA}}^* : \mathcal{S} \to \mathbb{R}$ by

$$V_{\text{A}}^*(s) = \max_{\pi \in \Pi} \min_{\tau \in \mathbb{N}} q\left(\xi_s^\pi(\tau)\right),$$

$$\tilde{V}_{\text{RA}}^*(s) = \max_{\pi \in \Pi} \max_{\tau \in \mathbb{N}} \min \left\{ \tilde{r}_{\text{RAA}}\left(\xi_s^\pi(\tau)\right), \min_{\kappa \leq \tau} q\left(\xi_s^\pi(\kappa)\right) \right\},$$

$$V_{\text{RAA}}^*(s) = \max_{\pi \in \Pi} \min \left\{ \max_{\tau \in \mathbb{N}} r\left(\xi_s^\pi(\tau)\right), \min_{\kappa \in \mathbb{N}} q\left(\xi_s^\pi(\kappa)\right) \right\},$$

where $\tilde{r}_{\text{RAA}}$ is as in Theorem 1.

We next define the value functions, $v_{\text{A}}^*, \tilde{v}_{\text{RA}}^*, v_{\text{RAA}}^* : \mathcal{S} \to \mathbb{R}$, which maximize over action sequences rather than policies:

$$v_{\text{A}}^*(s) = \max_{\mathbf{a} \in \mathbb{A}} \min_{\tau \in \mathbb{N}} q\left(\xi_s^{\mathbf{a}}(\tau)\right),$$

$$\tilde{v}_{\text{RA}}^*(s) = \max_{\mathbf{a} \in \mathbb{A}} \max_{\tau \in \mathbb{N}} \min \left\{ \tilde{r}_{\text{RAA}}\left(\xi_s^{\mathbf{a}}(\tau)\right), \min_{\kappa \leq \tau} q\left(\xi_s^{\mathbf{a}}(\kappa)\right) \right\},$$

$$v_{\text{RAA}}^*(s) = \max_{\mathbf{a} \in \mathbb{A}} \min \left\{ \max_{\tau \in \mathbb{N}} r\left(\xi_s^{\mathbf{a}}(\tau)\right), \min_{\kappa \in \mathbb{N}} q\left(\xi_s^{\mathbf{a}}(\kappa)\right) \right\},$$

Observe that for each $s \in \mathcal{S}$,

$$v_{\text{A}}^*(s) \geq V_{\text{A}}^*(s), \quad \tilde{v}_{\text{RA}}^*(s) \geq \tilde{V}_{\text{RA}}^*(s), \quad v_{\text{RAA}}^*(s) \geq V_{\text{RAA}}^*(s).$$

We now prove a series of lemmas that will be useful in the proof of the main theorem.

**Lemma 1.** *There is a $\pi \in \Pi$ such that*

$$v_{\mathrm{A}}^*(s) = \min_{\tau \in \mathbb{N}} q\left(\xi_s^\pi(\tau)\right)$$

*for all $s \in \mathcal{S}$.*

*Proof.* Choose $\pi \in \Pi$ such that

$$\pi(s) \in \arg\max_{a \in \mathcal{A}} v_{\mathrm{A}}^*\left(f(s,a)\right) \quad \forall s \in \mathcal{S}.$$

Fix $s \in \mathcal{S}$. Note that for each $\tau \in \mathbb{N}$,

$$
\begin{aligned}
v_{\mathrm{A}}^*\left(\xi_s^\pi(\tau+1)\right) &= v_{\mathrm{A}}^*\left(f\left(\xi_s^\pi(\tau), \pi\left(\xi_s^\pi(\tau)\right)\right)\right) \\
&= \max_{a \in \mathcal{A}} v_{\mathrm{A}}^*\left(f\left(\xi_s^\pi(\tau), a\right)\right) \\
&= \max_{a \in \mathcal{A}} \max_{\mathbf{a} \in \mathbb{A}} \min_{\kappa \in \mathbb{N}} q\left(\xi_{f\left(\xi_s^\pi(\tau), a\right)}^{\mathbf{a}}(\kappa)\right) \\
&= \max_{a \in \mathcal{A}} \max_{\mathbf{a} \in \mathbb{A}} \min_{\kappa \in \mathbb{N}} q\left(\xi_{\xi_s^\pi(\tau)}^{[a,\mathbf{a}]}(\kappa+1)\right) \\
&= \max_{\mathbf{a} \in \mathbb{A}} \min_{\kappa \in \mathbb{N}} q\left(\xi_{\xi_s^\pi(\tau)}^{\mathbf{a}}(\kappa+1)\right) \\
&\geq \max_{\mathbf{a} \in \mathbb{A}} \min_{\kappa \in \mathbb{N}} q\left(\xi_{\xi_s^\pi(\tau)}^{\mathbf{a}}(\kappa)\right) \\
&\geq v_{\mathrm{A}}^*\left(\xi_s^\pi(\tau)\right).
\end{aligned}
$$

It follows by induction that $v_{\mathrm{A}}^*\left(\xi_s^\pi(\tau)\right) \geq v_{\mathrm{A}}^*\left(\xi_s^\pi(0)\right)$ for all $\tau \in \mathbb{N}$, so that

$$v_{\mathrm{A}}^*(s) \geq \min_{\tau \in \mathbb{N}} q\left(\xi_s^\pi(\tau)\right) \geq \min_{\tau \in \mathbb{N}} v_{\mathrm{A}}^*\left(\xi_s^\pi(\tau)\right) = v_{\mathrm{A}}^*\left(\xi_s^\pi(0)\right) = v_{\mathrm{A}}^*(s).$$

$\square$

**Corollary 3.** *For all $s \in \mathcal{S}$, we have $V_{\mathrm{A}}^*(s) = v_{\mathrm{A}}^*(s)$.*

**Lemma 2.** *There is a $\pi \in \Pi$ such that*

$$\tilde{v}_{\mathrm{RA}}^*(s) = \max_{\tau \in \mathbb{N}} \min\left\{\tilde{r}_{\mathrm{RAA}}\left(\xi_s^\pi(\tau)\right), \min_{\kappa \leq \tau} q\left(\xi_s^\pi(\kappa)\right)\right\}$$

*for all $s \in \mathcal{S}$.*

*Proof.* First, let us note that in this proof we will use the standard conventions that

$$\max \varnothing = -\infty \quad \text{and} \quad \min \varnothing = +\infty.$$

We next introduce some notation. First, for convenience, we set $v^* = \tilde{v}_{\mathrm{RA}}^*$ and $V^* = \tilde{V}_{\mathrm{RA}}^*$. Given $s \in \mathcal{S}$ and $\mathbf{a} \in \mathbb{A}$, we write

$$v^{\mathbf{a}}(s) = \max_{\tau \in \mathbb{N}} \min\left\{\tilde{r}_{\mathrm{RAA}}\left(\xi_s^{\mathbf{a}}(\tau)\right), \min_{\kappa \leq \tau} q\left(\xi_s^{\mathbf{a}}(\kappa)\right)\right\}.$$

Similarly, given $s \in \mathcal{S}$ and $\pi \in \Pi$, we write

$$V^\pi(s) = \max_{\tau \in \mathbb{N}} \min\left\{\tilde{r}_{\mathrm{RAA}}\left(\xi_s^\pi(\tau)\right), \min_{\kappa \leq \tau} q\left(\xi_s^\pi(\kappa)\right)\right\}.$$

Then

$$V^*(s) = \max_{\pi \in \Pi} \max_{\tau \in \mathbb{N}} \min\left\{\tilde{r}_{\mathrm{RAA}}\left(\xi_s^\pi(\tau)\right), \min_{\kappa \leq \tau} q\left(\xi_s^\pi(\kappa)\right)\right\} = \max_{\pi \in \Pi} V^\pi(s),$$

and

$$v^*(s) = \max_{\mathbf{a} \in \mathbb{A}} \max_{\tau \in \mathbb{N}} \min\left\{\tilde{r}_{\mathrm{RAA}}\left(\xi_s^{\mathbf{a}}(\tau)\right), \min_{\kappa \leq \tau} q\left(\xi_s^{\mathbf{a}}(\kappa)\right)\right\} = \max_{\mathbf{a} \in \mathbb{A}} v^{\mathbf{a}}(s).$$

It is immediate that $v^*(s) \geq V^*(s)$ for each $s \in \mathcal{S}$, so it suffices to show the reverse inequality. Toward this end, it suffices to show that there is a $\pi \in \Pi$ for which $V^\pi(s) = v^*(s)$ for each $s \in \mathcal{S}$. Indeed, in this case, $V^*(s) \geq V^\pi(s) = v^*(s)$.

We now construct the desired policy $\pi$. Let $\alpha_0 = +\infty$, $S_0 = \varnothing$, and $v_0^* : \mathcal{S} \to \mathbb{R} \cup \{-\infty\}, s \mapsto -\infty$. We recursively define $\alpha_t \in \mathbb{R}$, $S_t \subseteq \mathcal{S}$, and $v_t^* : \mathcal{S} \to \mathbb{R} \cup \{-\infty\}$ for $t = 1, 2, \ldots$ by

$$\alpha_{t+1} = \max_{s \in \mathcal{S} \setminus S_t} \min \left\{ \max \left\{ \tilde{r}_{\mathrm{RAA}}(s), \max_{a \in \mathcal{A}} v_t^*(f(s,a)) \right\}, q(s) \right\}, \tag{3}$$

$$S_{t+1} = S_t \cup \left\{ s \in \mathcal{S} \setminus S_t \,\middle|\, \min \left\{ \max \left\{ \tilde{r}_{\mathrm{RAA}}(s), \max_{a \in \mathcal{A}} v_t^*(f(s,a)) \right\}, q(s) \right\} = \alpha_{t+1} \right\}, \tag{4}$$

$$v_{t+1}^*(s) = \begin{cases} v_t^*(s) & s \in S_t, \\ \alpha_{t+1} & s \in S_{t+1} \setminus S_t, \\ -\infty & s \in \mathcal{S} \setminus S_{t+1}. \end{cases} \tag{5}$$

From (4) it follows that

$$S_0 \subseteq S_1 \subseteq S_2 \subseteq \ldots, \tag{6}$$

which together with (3) shows that

$$\alpha_0 \geq \alpha_1 \geq \alpha_2 \geq \ldots. \tag{7}$$

Also, whenever $\mathcal{S} \setminus S_t$ is non-empty, the set being appended to $S_t$ in (4) is non-empty so

$$\bigcup_{t=0}^{\infty} S_t = \mathcal{S}. \tag{8}$$

For each $s \in \mathcal{S}$, let $\sigma(s)$ be the smallest $t \in \mathbb{N}$ for which $s \in S_t$. We choose the policy $\pi \in \Pi$ of interest by insisting

$$\pi(s) \in \arg\max_{a \in \mathcal{A}} v_{\sigma(s)-1}^*(f(s,a)) \quad \forall s \in \mathcal{S}. \tag{9}$$

In the remainder of the proof, we show that $V^\pi(s) = v^*(s)$ for each $s \in \mathcal{S}$ by induction. Let $n \in \mathbb{N}$ and suppose the following induction assumptions hold:

$$V^\pi(s) = v^*(s) = v_n^*(s) \geq \alpha_n \quad \forall s \in S_n, \tag{10}$$

$$v^*(s') \leq \alpha_n \quad \forall s' \in \mathcal{S} \setminus S_n. \tag{11}$$

Note that the above hold trivially when $n = 0$ since $S_0 = \varnothing$ and $\alpha_0 = +\infty$. Fix some particular $y \in S_{n+1}$ and some $z \in \mathcal{S} \setminus S_{n+1}$. We must show that

$$V^\pi(y) = v^*(y) = v_{n+1}^*(y) \geq \alpha_{n+1}, \tag{12}$$

$$v^*(z) \leq \alpha_{n+1}. \tag{13}$$

In this case, induction then shows that $V^\pi(s) = v^*(s)$ for all $s \in \cup_{n=0}^{\infty} S_t$. Since this union is equal to $\mathcal{S}$ by (8), the desired result then follows.

To show (12)-(13), we first demonstrate the following three claims.

1. Let $x \in \mathcal{S}$ and $w \in \mathcal{A}$ be such that $f(x,w) \in S_n$ and $q(x) \geq \alpha_{n+1}$. We claim $x \in S_{n+1}$.

   We can assume $x \notin S_n$, for otherwise the claim follows immediately from (6). Since $f(x,w) \in S_n$, we have $v_n^*(f(x,w)) \geq \alpha_n$ by (10). Thus

   $$\alpha_{n+1} \geq \min \left\{ \max \left\{ \tilde{r}_{\mathrm{RAA}}(x), \max_{a \in \mathcal{A}} v_n^*(f(x,a)) \right\}, q(x) \right\}$$
   $$\geq \min \left\{ \max\{\tilde{r}_{\mathrm{RAA}}(x), \alpha_n\}, \alpha_{n+1} \right\}$$
   $$= \alpha_{n+1},$$

   where the first inequality follows from (3), and the equality follows from (7). Thus

   $$\alpha_{n+1} = \min \left\{ \max \left\{ \tilde{r}_{\mathrm{RAA}}(x), \max_{a \in \mathcal{A}} v_n^*(f(x,a)) \right\}, q(x) \right\},$$

   so the claim follows from (4).

2. Let $x \in S_{n+1} \setminus S_n$ and $w \in \mathcal{A}$ be such that $f(x, w) \in S_n$. We claim that

$$V^\pi(x) = v^*(x) = \alpha_{n+1}. \tag{14}$$

To show this claim, we will make use of the dynamic programming principle

$$v^{\mathbf{a}}(s) = \min \left\{ \max \left\{ \tilde{r}_{\mathrm{RAA}}(s), v^{\mathbf{a}|_1}\left(f(s, \mathbf{a}(0))\right) \right\}, q(s) \right\}, \quad \forall s \in \mathcal{S}, \mathbf{a} \in \mathbb{A},$$

from which it follows that

$$V^\pi(s) = \min \left\{ \max \left\{ \tilde{r}_{\mathrm{RAA}}(s), V^\pi\left(f(s, \pi(s))\right) \right\}, q(s) \right\}, \quad \forall s \in \mathcal{S}, \tag{15}$$

and

$$v^*(s) = \min \left\{ \max \left\{ \tilde{r}_{\mathrm{RAA}}(s), \max_{a \in \mathcal{A}} v^*\left(f(s, a)\right) \right\}, q(s) \right\}, \quad \forall s \in \mathcal{S}. \tag{16}$$

Since $x \in S_{n+1} \setminus S_n$, then $\sigma(x) = n + 1$ by definition of $\sigma$, so $\pi(x) \in \arg\max_{a \in \mathcal{A}} v_n^*(f(x, a))$ by (9). Thus

$$v_n^*\left(f(x, \pi(x))\right) = \max_{a \in \mathcal{A}} v_n^*\left(f(x, a)\right). \tag{17}$$

But then

$$v_n^*\left(f(x, \pi(x))\right) \geq v_n^*\left(f(x, w)\right) \geq \alpha_n \geq \alpha_{n+1} > -\infty,$$

where the second inequality comes from (10), the third comes from (7), and the final inequality comes from (3) ($\mathcal{S} \setminus S_n$ is non-empty because $x \in \mathcal{S} \setminus S_n$). Thus $f(x, \pi(x)) \in S_n$ by (5). It then follows from (10) that

$$V^\pi\left(f(x, \pi(x))\right) = v^*\left(f(x, \pi(x))\right) = v_n^*\left(f(x, \pi(x))\right). \tag{18}$$

Now, observe that for all $s \in S_n$ and $s' \in \mathcal{S} \setminus S_n$,

$$v^*(s) = v_n^*(s) \geq \alpha_n \geq v^*(s') \geq -\infty = v_n^*(s'), \tag{19}$$

where the first equality and inequality are from (10), the second inequality is from (11), and the final equality is from (5). Moreover, $f(x, a) \in S_n$ for at least one $a$ (in particular $a = w$). Letting $\mathcal{A}' = \{a \in \mathcal{A} \mid f(x, a) \in S_n\}$, it follows from (19) that

$$\max_{a \in \mathcal{A}} v^*\left(f(x, a)\right) = \max_{a \in \mathcal{A}'} v^*\left(f(x, a)\right) = \max_{a \in \mathcal{A}'} v_n^*\left(f(x, a)\right) = \max_{a \in \mathcal{A}} v_n^*\left(f(x, a)\right). \tag{20}$$

From (17)-(20) we have

$$V^\pi\left(f(x, \pi(x))\right) = \max_{a \in \mathcal{A}} v^*\left(f(x, a)\right) = \max_{a \in \mathcal{A}} v_n^*\left(f(x, a)\right). \tag{21}$$

Now observe that

$$V^\pi(x) = \min \left\{ \max \left\{ \tilde{r}_{\mathrm{RAA}}(x), V^\pi\left(f(x, \pi(x))\right) \right\}, q(x) \right\},$$

$$v^*(x) = \min \left\{ \max \left\{ \tilde{r}_{\mathrm{RAA}}(x), \max_{a \in \mathcal{A}} v^*\left(f(x, a)\right) \right\}, q(x) \right\},$$

$$\alpha_{n+1} = \min \left\{ \max \left\{ \tilde{r}_{\mathrm{RAA}}(x), \max_{a \in \mathcal{A}} v_n^*\left(f(x, a)\right) \right\}, q(x) \right\},$$

where the first equation is from (15), the second is from (16), and the third is from (4). But then (14) follows from the above equations together with (21).

3. Let $x \in \mathcal{S} \setminus S_n$. We claim that $v^*(x) \leq \alpha_{n+1}$. Suppose otherwise. Then we can choose $\mathbf{a} \in \mathbb{A}$ and $\tau \in \mathbb{N}$ such that

$$\min \left\{ \tilde{r}_{\mathrm{RAA}}\left(\xi_x^{\mathbf{a}}(\tau)\right), \min_{\kappa \leq \tau} q\left(\xi_x^{\mathbf{a}}(\kappa)\right) \right\} > \alpha_{n+1}. \tag{22}$$

It follows that $\xi_x^{\mathbf{a}}(\tau) \in S_n$, for otherwise

$$\alpha_{n+1} \geq \min \left\{ \tilde{r}_{\mathrm{RAA}}(\xi_x^{\mathbf{a}}(\tau)), q(\xi_x^{\mathbf{a}}(\tau)) \right\}$$

by (3), creating a contradiction.

So $x \notin S_n$ and $\xi_x^{\mathbf{a}}(\tau) \in S_n$, indicating that there is some $\theta \in \{0, \ldots, \tau - 1\}$ such that $\xi_x^{\mathbf{a}}(\theta) \notin S_n$ and $f\left(\xi_x^{\mathbf{a}}(\theta), \mathbf{a}(\theta)\right) = \xi_x^{\mathbf{a}}(\theta + 1) \in S_n$. Moreover, $q\left(\xi_x^{\mathbf{a}}(\theta)\right) > \alpha_{n+1}$ by (22). It follows from claim 1 that $\xi_x^{\mathbf{a}}(\theta) \in S_{n+1}$.

But then it follows from claim 2 that $v^*\left(\xi_x^{\mathbf{a}}(\theta)\right) = \alpha_{n+1}$. However,

$$
v^*\left(\xi_x^{\mathbf{a}}(\theta)\right) \geq \min\left\{\tilde{r}_{\text{RAA}}\left(\xi_{\xi_x^{\mathbf{a}}(\theta)}^{\mathbf{a}}(\tau - \theta)\right), \min_{\kappa \leq \tau - \theta} q\left(\xi_{\xi_x^{\mathbf{a}}(\theta)}^{\mathbf{a}}(\kappa)\right)\right\}
$$

$$
= \min\left\{\tilde{r}_{\text{RAA}}\left(\xi_x^{\mathbf{a}}(\tau - \theta + \theta)\right), \min_{\kappa \leq \tau - \theta} q\left(\xi_x^{\mathbf{a}}(\kappa + \theta)\right)\right\}
$$

$$
= \min\left\{\tilde{r}_{\text{RAA}}\left(\xi_x^{\mathbf{a}}(\tau)\right), \min_{\kappa \in \{\theta, \theta+1, \ldots, \tau\}} q\left(\xi_x^{\mathbf{a}}(\kappa)\right)\right\}
$$

$$
> \alpha_{n+1},
$$

giving the desired contradiction.

Having established these claims, we return to proving (12) and (13) hold. In fact, (13) follows immediately from claim 3, so we actually only need to show (12).

If $y \in S_n$, then from (5) and (10), we have that $V^\pi(y) = v^*(y) = v_n^*(y) = v_{n+1}^*(y)$, and from (7) and (10), we also have that $v_n^*(y) \geq \alpha_n \geq \alpha_{n+1}$. Together these establish (12) when $y \in S_n$.

So suppose $y \in S_{n+1} \setminus S_n$. First, observe that $v_{n+1}^*(y) = \alpha_{n+1}$ by (5). There are now two possibilities. If there is some $a \in \mathcal{A}$ for which $f(y, a) \in S_n$, then (12) follows from claim 2. If instead, $f(y, a) \notin S_n$ for each $a \in \mathcal{A}$, then $\max_{a \in \mathcal{A}} v_n^*(f(y, a)) = -\infty$ by (5) (or if $n = 0$ by definition of $v_0^*$). Thus $\alpha_{n+1} = \min\{\tilde{r}_{\text{RAA}}(y), q(y)\}$ by (4), so

$$
v^*(y) \geq V^\pi(y) \geq \min\{\tilde{r}_{\text{RAA}}(y), q(y)\} = \alpha_{n+1} \geq v^*(y),
$$

where the final inequality follows from claim 3. This completes the proof. $\qquad\square$

**Corollary 4.** *For all $s \in \mathcal{S}$, we have $\tilde{V}_{\text{RA}}^*(s) = \tilde{v}_{\text{RA}}^*(s)$.*

**Lemma 3.** *Let $F : \mathbb{A} \times \mathbb{N} \to \mathbb{R}$. Then*

$$
\sup_{\mathbf{a} \in \mathbb{A}} \sup_{\tau \in \mathbb{N}} \sup_{\mathbf{a}' \in \mathbb{A}'} F\left([\mathbf{a}, \mathbf{a}']_\tau, \tau\right) = \sup_{\mathbf{a} \in \mathbb{A}} \sup_{\tau \in \mathbb{N}} F\left(\mathbf{a}, \tau\right). \tag{23}
$$

*Proof.* We proceed by showing both inequalities corresponding to (23) hold.

($\geq$) Given any $\mathbf{a} \in \mathbb{A}$ and $\tau \in \mathbb{N}$, we have $\sup_{\mathbf{a}' \in \mathbb{A}'} F\left([\mathbf{a}, \mathbf{a}']_\tau, \tau\right) \geq F\left(\mathbf{a}, \tau\right)$. Taking the suprema over $\mathbf{a} \in \mathbb{A}$ and $\tau \in \mathbb{N}$ on both sides of this inequality gives the desired result.

($\leq$) Given any $\mathbf{a} \in \mathbb{A}$ and $\tau \in \mathbb{N}$, we have

$$
\sup_{\mathbf{a}' \in \mathbb{A}'} F\left([\mathbf{a}, \mathbf{a}']_\tau, \tau\right) \leq \sup_{\mathbf{a}'' \in \mathbb{A}} F\left(\mathbf{a}'', \tau\right),
$$

so that the result follows from taking the suprema over $\mathbf{a} \in \mathbb{A}$ and $\tau \in \mathbb{N}$ on both sides of this inequality.

$\qquad\square$

**Lemma 4.** *For each $s \in \mathcal{S}$,*

$$
v_{\text{RAA}}^*(s) = \tilde{v}_{\text{RA}}^*(s).
$$

*Proof.* For each $s \in \mathcal{S}$, we have

$$\tilde{v}_{\text{RA}}^*(s) = \max_{\mathbf{a} \in \mathbb{A}} \max_{\tau \in \mathbb{N}} \min \left\{ \tilde{r}_{\text{RAA}} \left( \xi_s^{\mathbf{a}}(\tau) \right), \min_{\kappa \leq \tau} q \left( \xi_s^{\mathbf{a}}(\kappa) \right) \right\} \tag{24}$$

$$= \max_{\mathbf{a} \in \mathbb{A}} \max_{\tau \in \mathbb{N}} \min \left\{ r \left( \xi_s^{\mathbf{a}}(\tau) \right), v_{\text{A}}^* \left( \xi_s^{\mathbf{a}}(\tau) \right), \min_{\kappa \leq \tau} q \left( \xi_s^{\mathbf{a}}(\kappa) \right) \right\} \tag{25}$$

$$= \max_{\mathbf{a} \in \mathbb{A}} \max_{\tau \in \mathbb{N}} \min \left\{ r \left( \xi_s^{\mathbf{a}}(\tau) \right), \max_{\mathbf{a}' \in \mathbb{A}} \min_{\kappa' \in \mathbb{N}} q \left( \xi_{\xi_s^{\mathbf{a}}(\tau)}^{\mathbf{a}'}(\kappa') \right), \min_{\kappa \leq \tau} q \left( \xi_s^{\mathbf{a}}(\kappa) \right) \right\}$$

$$= \max_{\mathbf{a} \in \mathbb{A}} \max_{\tau \in \mathbb{N}} \min \left\{ r \left( \xi_s^{\mathbf{a}}(\tau) \right), \max_{\mathbf{a}' \in \mathbb{A}} \min_{\kappa' \in \mathbb{N}} q \left( \xi_s^{[\mathbf{a},\mathbf{a}']_\tau}(\tau + \kappa') \right), \min_{\kappa \leq \tau} q \left( \xi_s^{\mathbf{a}}(\kappa) \right) \right\}$$

$$= \max_{\mathbf{a} \in \mathbb{A}} \max_{\tau \in \mathbb{N}} \max_{\mathbf{a}' \in \mathbb{A}} \min \left\{ r \left( \xi_s^{\mathbf{a}}(\tau) \right), \min_{\kappa' \in \mathbb{N}} q \left( \xi_s^{[\mathbf{a},\mathbf{a}']_\tau}(\tau + \kappa') \right), \min_{\kappa \leq \tau} q \left( \xi_s^{\mathbf{a}}(\kappa) \right) \right\}$$

$$= \max_{\mathbf{a} \in \mathbb{A}} \max_{\tau \in \mathbb{N}} \max_{\mathbf{a}' \in \mathbb{A}} \min \left\{ r \left( \xi_s^{[\mathbf{a},\mathbf{a}']_\tau}(\tau) \right), \min_{\kappa' \in \mathbb{N}} q \left( \xi_s^{[\mathbf{a},\mathbf{a}']_\tau}(\tau + \kappa') \right), \min_{\kappa \leq \tau} q \left( \xi_s^{[\mathbf{a},\mathbf{a}']_\tau}(\kappa) \right) \right\} \tag{26}$$

$$= \max_{\mathbf{a} \in \mathbb{A}} \max_{\tau \in \mathbb{N}} \min \left\{ r \left( \xi_s^{\mathbf{a}}(\tau) \right), \min_{\kappa' \in \mathbb{N}} q \left( \xi_s^{\mathbf{a}}(\tau + \kappa') \right), \min_{\kappa \leq \tau} q \left( \xi_s^{\mathbf{a}}(\kappa) \right) \right\} \tag{27}$$

$$= \max_{\mathbf{a} \in \mathbb{A}} \max_{\tau \in \mathbb{N}} \min \left\{ r \left( \xi_s^{\mathbf{a}}(\tau) \right), \min_{\kappa \in \mathbb{N}} q \left( \xi_s^{\mathbf{a}}(\kappa) \right) \right\}$$

$$= \max_{\mathbf{a} \in \mathbb{A}} \min \left\{ \max_{\tau \in \mathbb{N}} r \left( \xi_s^{\mathbf{a}}(\tau) \right), \min_{\kappa \in \mathbb{N}} q \left( \xi_s^{\mathbf{a}}(\kappa) \right) \right\}$$

$$= v_{\text{RAA}}^*(s),$$

where the equality between (24) and (25) follows from Corollary 3, and where the equality between (26) and (27) follows from Lemma 3. $\qquad\square$

Before the next lemma, we need to introduce two last pieces of notation. First, we let $\overline{\overline{\Pi}}$ be the set of augmented policies $\bar{\pi} : \mathcal{S} \times \mathcal{Y} \times \mathcal{Z} \to \mathcal{A}$, where

$$\mathcal{Y} = \{ r(s) \mid s \in \mathcal{S} \} \quad \text{and} \quad \mathcal{Z} = \{ q(s) \mid s \in \mathcal{S} \}.$$

Next, given $s \in \mathcal{S}$, $y \in \mathcal{Y}$, $z \in \mathcal{Z}$, and $\bar{\pi} \in \overline{\overline{\Pi}}$, we let $\bar{\xi}_s^{\bar{\pi}} : \mathbb{N} \to \mathcal{S}$, $\bar{\eta}_s^{\bar{\pi}} : \mathbb{N} \to \mathcal{Y}$, and $\bar{\zeta}_s^{\bar{\pi}} : \mathbb{N} \to \mathcal{Z}$, be the solution of the evolution

$$\bar{\xi}_s^{\bar{\pi}}(t+1) = f \left( \bar{\xi}_s^{\bar{\pi}}(t), \bar{\pi} \left( \bar{\xi}_s^{\bar{\pi}}(t), \bar{\eta}_s^{\bar{\pi}}(t), \bar{\zeta}_s^{\bar{\pi}}(t) \right) \right),$$
$$\bar{\eta}_s^{\bar{\pi}}(t+1) = \max \left\{ r \left( \bar{\xi}_s^{\bar{\pi}}(t+1) \right), \bar{\eta}_s^{\bar{\pi}}(t) \right\},$$
$$\bar{\zeta}_s^{\bar{\pi}}(t+1) = \min \left\{ q \left( \bar{\xi}_s^{\bar{\pi}}(t+1) \right), \bar{\zeta}_s^{\bar{\pi}}(t) \right\},$$

for which $\bar{\xi}_s^{\bar{\pi}}(0) = s$, $\bar{\eta}_s^{\bar{\pi}}(0) = r(s)$, and $\bar{\zeta}_s^{\bar{\pi}}(0) = q(s)$.

**Lemma 5.** *There is a $\bar{\pi} \in \overline{\overline{\Pi}}$ such that*

$$v_{\text{RAA}}^*(s) = \min \left\{ \max_{\tau \in \mathbb{N}} r \left( \bar{\xi}_s^{\bar{\pi}}(\tau) \right), \min_{\tau \in \mathbb{N}} q \left( \bar{\xi}_s^{\bar{\pi}}(\tau) \right) \right\} \tag{28}$$

*for all $s \in \mathcal{S}$.*

*Proof.* By Lemmas 1 and 2 together with Corollary 3, we can choose $\pi, \theta \in \Pi$ such that

$$\tilde{v}_{\text{RA}}^*(s) = \max_{\tau \in \mathbb{N}} \min \left\{ r \left( \xi_s^\pi(\tau) \right), v_{\text{A}}^* \left( \xi_s^\pi(\tau) \right), \min_{\kappa \leq \tau} q \left( \xi_s^\pi(\kappa) \right) \right\} \quad \forall s \in \mathcal{S},$$
$$v_{\text{A}}^*(s) = \min_{\tau \in \mathbb{N}} q \left( \xi_s^\theta(\tau) \right) \quad \forall s \in \mathcal{S}.$$

We introduce some useful notation we will use throughout the rest of the proof. For each $s \in \mathcal{S}$, let $[s]^+ = f(s, \pi(s))$, $[y]_s^+ = \max\{y, r([s]^+)\}$, $[z]_s^+ = \min\{z, q([s]^+)\}$.

We define an augmented policy $\bar{\pi} \in \bar{\Pi}$ by

$$\bar{\pi}(s, y, z) = \begin{cases} \pi(s) & \min\{[y]_s^+, [z]_s^+, v_{\mathrm{A}}^*([s]^+)\} \geq \min\{y, z, v_{\mathrm{A}}^*(s)\}, \\ \theta(s) & \text{otherwise.} \end{cases}$$

Now fix some $s \in \mathcal{S}$. For all $t \in \mathbb{N}$, set $\bar{x}_t = \bar{\xi}_s^{\bar{\pi}}(t)$, $\bar{y}_t = \bar{\eta}_s^{\bar{\pi}}(t) = \max_{\tau \leq t} r(\bar{x}_\tau)$, and $\bar{z}_t = \bar{\zeta}_s^{\bar{\pi}}(t) = \min_{\tau \leq t} q(\bar{x}_\tau)$, and also set $x_t^\circ = \xi_s^\pi(t)$, $y_t^\circ = \max_{\tau \leq t} r(x_\tau^\circ)$, and $z_t^\circ = \min_{\tau \leq t} q(x_\tau^\circ)$.

First, assume that $t$ is such that $\min\{[\bar{y}_t]_{\bar{x}_t}^+, [\bar{z}_t]_{\bar{x}_t}^+, v_{\mathrm{A}}^*([\bar{x}_t]^+)\} < \min\{\bar{y}_t, \bar{z}_t, v_{\mathrm{A}}^*(\bar{x}_t)\}$. In this case, $\bar{\pi}(\bar{x}_t, \bar{y}_t, \bar{z}_t) = \theta(\bar{x}_t)$, so that

$$\min\{\bar{z}_t, v_{\mathrm{A}}^*(\bar{x}_t)\} = \min\{\bar{z}_{t+1}, v_{\mathrm{A}}^*(\bar{x}_{t+1})\}$$

by our choice of $\theta$. Since $\bar{y}_t$ is non-decreasing in $t$, thus have

$$\min\{\bar{y}_t, \bar{z}_t, v_{\mathrm{A}}^*(\bar{x}_t)\} \leq \min\{\bar{y}_{t+1}, \bar{z}_{t+1}, v_{\mathrm{A}}^*(\bar{x}_{t+1})\}.$$

Next, assume that $t$ is such that $\min\{[\bar{y}_t]_{\bar{x}_t}^+, [\bar{z}_t]_{\bar{x}_t}^+, v_{\mathrm{A}}^*([\bar{x}_t]^+)\} \geq \min\{\bar{y}_t, \bar{z}_t, v_{\mathrm{A}}^*(\bar{x}_t)\}$. In this case, we have that $\bar{\pi}(\bar{x}_t, \bar{y}_t, \bar{z}_t) = \pi(\bar{x}_t)$, so

$$\min\{\bar{y}_t, \bar{z}_t, v_{\mathrm{A}}^*(\bar{x}_t)\} \leq \min\{[\bar{y}_t]_{\bar{x}_t}^+, [\bar{z}_t]_{\bar{x}_t}^+, v_{\mathrm{A}}^*([\bar{x}_t]^+)\} = \min\{\bar{y}_{t+1}, \bar{z}_{t+1}, v_{\mathrm{A}}^*(\bar{x}_{t+1})\}.$$

It thus follows from these two cases that $\min\{\bar{y}_t, \bar{z}_t, v_{\mathrm{A}}^*(\bar{x}_t)\}$ is non-decreasing in $t$. Let

$$T = \min\left\{t \in \mathbb{N} \mid \min\{[\bar{y}_t]_{\bar{x}_t}^+, [\bar{z}_t]_{\bar{x}_t}^+, v_{\mathrm{A}}^*([\bar{x}_t]^+)\} < \min\{\bar{y}_t, \bar{z}_t, v_{\mathrm{A}}^*(\bar{x}_t)\}\right\}.$$

There are again two cases:

$(T < \infty)$ In this case, $\bar{\pi}(\bar{x}_t, \bar{y}_t, \bar{z}_t) = \pi(\bar{x}_t)$ for $t < T$. Then $\bar{x}_t = x_t^\circ$, $\bar{y}_t = y_t^\circ$, and $\bar{z}_t = z_t^\circ$ for all $t \leq T$. It follows that $[\bar{x}_t]^+ = x_{t+1}^\circ$, $[\bar{y}_t]_{\bar{x}_t}^+ = y_{t+1}^\circ$, and $[\bar{z}_t]_{\bar{x}_t}^+ = z_{t+1}^\circ$ for all $t \leq T$. Thus by definition of $T$,

$$\min\left\{y_{t+1}^\circ, z_{t+1}^\circ, v_{\mathrm{A}}^*\left(x_{t+1}^\circ\right)\right\} \geq \min\left\{y_t^\circ, z_t^\circ, v_{\mathrm{A}}^*\left(x_t^\circ\right)\right\} \quad \forall t < T.$$

and

$$\min\left\{y_{T+1}^\circ, z_{T+1}^\circ, v_{\mathrm{A}}^*\left(x_{T+1}^\circ\right)\right\} < \min\left\{y_T^\circ, z_T^\circ, v_{\mathrm{A}}^*\left(x_T^\circ\right)\right\}.$$

But since $y_t^\circ$ is non-decreasing and $\min\{z_t^\circ, v_{\mathrm{A}}^*(x_t^\circ)\}$ is non-increasing in $t$, it follows that $\min\{y_t^\circ, z_t^\circ, v_{\mathrm{A}}^*(x_t^\circ)\}$ must achieve its maximal value at the smallest $t$ for which it strictly decreases from $t$ to $t + 1$, i.e.

$$\begin{aligned} \min\left\{\bar{y}_T, \bar{z}_T, v_{\mathrm{A}}^*\left(\bar{x}_T\right)\right\} &= \min\left\{y_T^\circ, z_T^\circ, v_{\mathrm{A}}^*\left(x_T^\circ\right)\right\} \\ &= \max_{t \in \mathbb{N}} \min\left\{y_t^\circ, z_t^\circ, v_{\mathrm{A}}^*\left(x_t^\circ\right)\right\} \\ &\geq \max_{t \in \mathbb{N}} \min\left\{r\left(x_t^\circ\right), z_t^\circ, v_{\mathrm{A}}^*\left(x_t^\circ\right)\right\} \\ &= \tilde{v}_{\mathrm{RA}}^*(s). \end{aligned}$$

where the final equality follows from our choice of $\pi$. Since $\min\{\bar{y}_t, \bar{z}_t, v_{\mathrm{A}}^*(\bar{x}_t)\}$ is non-decreasing in $t$, then

$$\min\{\bar{y}_t, \bar{z}_t\} \geq \min\{\bar{y}_t, \bar{z}_t, v_{\mathrm{A}}^*(\bar{x}_t)\} \geq \min\{\bar{y}_T, \bar{z}_T, v_{\mathrm{A}}^*(\bar{x}_T)\} = \tilde{v}_{\mathrm{RA}}^*(s) \quad \forall t \geq T.$$

Thus

$$v_{\mathrm{RAA}}^*(s) \geq \min\left\{\max_{t \in \mathbb{N}} r\left(\bar{x}_t\right), \min_{t \in \mathbb{N}} q\left(\bar{x}_t\right)\right\} = \lim_{t \to \infty} \min\{\bar{y}_t, \bar{z}_t\} \geq \tilde{v}_{\mathrm{RA}}^*(s) = v_{\mathrm{RAA}}^*(s),$$

where the final equality follows from Lemma (4). Thus the proof is complete in this case.

$(T = \infty)$ In this case, $\bar{\pi}(\bar{x}_t, \bar{y}_t, \bar{z}_t) = \pi(\bar{x}_t)$ for all $t \in \mathbb{N}$. Then $\bar{x}_t = x_t^\circ$, $\bar{y}_t = y_t^\circ$, and $\bar{z}_t = z_t^\circ$ for all $t \in \mathbb{N}$. Also $[\bar{x}_t]^+ = x_{t+1}^\circ$, $[\bar{y}_t]_{\bar{x}_t}^+ = y_{t+1}^\circ$, and $[\bar{z}_t]_{\bar{x}_t}^+ = z_{t+1}^\circ$ for all $t \in \mathbb{N}$. Thus by definition of $T$,

$$\min\left\{y_{t+1}^\circ, z_{t+1}^\circ, v_{\mathrm{A}}^*\left(x_{t+1}^\circ\right)\right\} \geq \min\left\{y_t^\circ, z_t^\circ, v_{\mathrm{A}}^*\left(x_t^\circ\right)\right\} \quad \forall t \in \mathbb{N}.$$

Let $T' \in \arg\max_{t\in\mathbb{N}} \min\{y_t^\circ, z_t^\circ, v_A^*(x_t^\circ)\}$. Then

$$\min\{\bar{y}_{T'}, \bar{z}_{T'}, v_A^*(\bar{x}_{T'})\} = \min\{y_{T'}^\circ, z_{T'}^\circ, v_A^*(x_{T'}^\circ)\}$$
$$= \max_{t\in\mathbb{N}} \min\{y_t^\circ, z_t^\circ, v_A^*(x_t^\circ)\}$$
$$\geq \max_{t\in\mathbb{N}} \min\{r(x_t^\circ), z_t^\circ, v_A^*(x_t^\circ)\}$$
$$= \tilde{v}_{RA}^*(s).$$

The rest of the proof the follows the same as the previous case with $T$ replaced by $T'$.

$\square$

**Corollary 5.** *For all $s \in \mathcal{S}$, we have $V_{RAA}^*(s) = v_{RAA}^*(s)$.*

*Proof of Theorem 1.* Theorem 1 is now a direct consequence of the previous corollary together with Corollary 4 and Lemma 4. $\square$

### A.1 A DIRECT DERIVATION OF THE RAA BELLMAN EQUATION

Here, we offer a direct derivation for the RAA Bellman equation. Note, this derivation does not guarantee that the resulting Bellman equation is unique, and is just for intuition for the rigor above.

$$v_{RAA}^*(s) := \max_{a_0,a_1,\dots} \min\left\{ \max_{\tau\in\{0,1,\dots\}} r(\mathbf{x}_s^{a_0,a_1,\dots}(\tau)), \min_{\kappa\in\{0,1,\dots\}} q(\mathbf{x}_s^{a_0,a_1,\dots}(\kappa)) \right\}$$
$$= \min\left\{ q(s), \max_{a_0,a_1,\dots} \min\left\{ \max_{\tau\in\{0,1,\dots\}} r(\mathbf{x}_s^{a_0,a_1,\dots}(\tau)), \min_{\kappa\in\{1,2,\dots\}} q(\mathbf{x}_s^{a_0,a_1,\dots}(\kappa)) \right\} \right\}$$
$$= \min\left\{ q(s), \max_{a_0,a_1,\dots} \min\left\{ \max\left\{ r(s), \max_{\tau\in\{1,2,\dots\}} r(\mathbf{x}_s^{a_0,a_1,\dots}(\tau)) \right\}, \min_{\kappa\in\{1,2,\dots\}} q(\mathbf{x}_s^{a_0,a_1,\dots}(\kappa)) \right\} \right\},$$
(29)

where $\mathbf{x}_s^{a_0,a_1,\dots}$ is the system trajectory starting from state $s$ under the sequence of actions $a_0, a_1, \dots$. Using the identity $\min\{\max\{a,b\},c\} = \max\{\min\{a,c\},\min\{b,c\}\}$,

$$v_{RAA}^*(s) = \min\left\{ q(s), \max_{a_0,a_1,\dots} \max\left\{ \min\left\{ r(s), \min_{\kappa\in\{1,2,\dots\}} q(\mathbf{x}_s^{a_0,a_1,\dots}(\kappa)) \right\}, \right.\right.$$
$$\left.\left. \min\left\{ \min_{\kappa\in\{1,2,\dots\}} q(\mathbf{x}_s^{a_0,a_1,\dots}(\kappa)), \max_{\tau\in\{1,2,\dots\}} r(\mathbf{x}_s^{a_0,a_1,\dots}(\tau)) \right\} \right\} \right\}$$
$$= \min\left\{ q(s), \max\left\{ \min\left\{ r(s), \max_{a_0,a_1,\dots} \min_{\kappa\in\{1,2,\dots\}} q(\mathbf{x}_s^{a_0,a_1,\dots}(\kappa)) \right\}, \right.\right.$$
$$\left.\left. \max_{a_0,a_1,\dots} \min\left\{ \min_{\kappa\in\{1,2,\dots\}} q(\mathbf{x}_s^{a_0,a_1,\dots}(\kappa)), \max_{\tau\in\{1,2,\dots\}} r(\mathbf{x}_s^{a_0,a_1,\dots}(\tau)) \right\} \right\} \right\}$$
$$= \min\left\{ q(s), \max\left\{ \min\left\{ r(s), \max_{a_0,a_1,\dots} \min_{\kappa\in\{1,2,\dots\}} q(\mathbf{x}_s^{a_0,a_1,\dots}(\kappa)) \right\}, \max_a v_{RAA}^*(f(s,a)) \right\} \right\}$$
$$= \min\left\{ q(s), \max\left\{ \min\left\{ q(s), r(s), \max_{a_0,a_1,\dots} \min_{\kappa\in\{0,1,\dots\}} q(\mathbf{x}_s^{a_0,a_1,\dots}(\kappa)) \right\}, \max_a v_{RAA}^*(f(s,a)) \right\} \right\}$$
$$= \min\left\{ q(s), \max\left\{ \min\left\{ r(s), \max_{a_0,a_1,\dots} \min_{\kappa\in\{0,1,\dots\}} q(\mathbf{x}_s^{a_0,a_1,\dots}(\kappa)) \right\}, \max_a v_{RAA}^*(f(s,a)) \right\} \right\},$$

where in the penultimate step we used the identity $\min\{a, \max\{b,c\}\} = \min\{a, \max\{\min\{a,b\}, c\}\}$. Noticing that $v_A^*(s) = \max_{a_0,a_1,\dots} \min_{\kappa\in\{0,1,\dots\}} q(\mathbf{x}_s^{a_0,a_1,\dots}(\kappa))$, we have

$$v_{RAA}^*(s) = \min\left\{ q(s), \max\left\{ \min\{r(s), v_A^*(s)\}, \max_a v_{RAA}^*(f(s,a)) \right\} \right\}. \tag{30}$$

This completes the derivation of the RAA Bellman equation.

Lastly, we may note that if define $\tilde{r}(s) := \min\{r(s), v_A^*(s)\}$, the above becomes the RA Bellman equation,

$$v_{RAA}^*(s) = \min\left\{ q(s), \max\left\{ \tilde{r}(s), \max_a v_{RAA}^*(f(s,a)) \right\} \right\}. \tag{31}$$

## B    PROOF OF RR MAIN THEOREM

We first define the value functions, $V_{\mathrm{R1}}^*, V_{\mathrm{R2}}^*, \tilde{V}_{\mathrm{R}}^*, V_{\mathrm{RR}}^* : \mathcal{S} \to \mathbb{R}$ by

$$V_{\mathrm{R1}}^*(s) = \max_{\pi \in \Pi} \max_{\tau \in \mathbb{N}} r_1\left(\xi_s^\pi(\tau)\right),$$

$$V_{\mathrm{R2}}^*(s) = \max_{\pi \in \Pi} \max_{\tau \in \mathbb{N}} r_2\left(\xi_s^\pi(\tau)\right),$$

$$\tilde{V}_{\mathrm{R}}^*(s) = \max_{\pi \in \Pi} \max_{\tau \in \mathbb{N}} \tilde{r}_{\mathrm{RR}}\left(\xi_s^{\mathbf{a}}(\tau)\right),$$

$$V_{\mathrm{RR}}^*(s) = \max_{\pi \in \Pi} \min\left\{ \max_{\tau \in \mathbb{N}} r_1\left(\xi_s^\pi(\tau)\right), \max_{\tau \in \mathbb{N}} r_2\left(\xi_s^\pi(\tau)\right) \right\}.$$

We next define the value functions, $v_{\mathrm{R1}}^*, v_{\mathrm{R2}}^*, \tilde{v}_{\mathrm{R}}^*, v_{\mathrm{RR}}^* : \mathcal{S} \to \mathbb{R}$, which maximize over action sequences rather than policies:

$$v_{\mathrm{R1}}^*(s) = \max_{\mathbf{a} \in \mathbb{A}} \max_{\tau \in \mathbb{N}} r_1\left(\xi_s^{\mathbf{a}}(\tau)\right),$$

$$v_{\mathrm{R2}}^*(s) = \max_{\mathbf{a} \in \mathbb{A}} \max_{\tau \in \mathbb{N}} r_2\left(\xi_s^{\mathbf{a}}(\tau)\right),$$

$$\tilde{v}_{\mathrm{R}}^*(s) = \max_{\mathbf{a} \in \mathbb{A}} \max_{\tau \in \mathbb{N}} \tilde{r}_{\mathrm{RR}}\left(\xi_s^{\mathbf{a}}(\tau)\right),$$

$$v_{\mathrm{RR}}^*(s) = \max_{\mathbf{a} \in \mathbb{A}} \min\left\{ \max_{\tau \in \mathbb{N}} r_1\left(\xi_s^{\mathbf{a}}(\tau)\right), \max_{\tau \in \mathbb{N}} r_2\left(\xi_s^{\mathbf{a}}(\tau)\right) \right\},$$

where $\tilde{r}_{\mathrm{RR}}$ is as in Theorem 2. Observe that for each $s \in \mathcal{S}$,

$$v_{\mathrm{R1}}^*(s) \geq V_{\mathrm{R1}}^*(s), \quad v_{\mathrm{R2}}^*(s) \geq V_{\mathrm{R2}}^*(s), \quad \tilde{v}_{\mathrm{R}}^*(s) \geq \tilde{V}_{\mathrm{R}}^*(s), \quad v_{\mathrm{RR}}^*(s) \geq V_{\mathrm{RR}}^*(s).$$

We now prove a series of lemmas that will be useful in the proof of the main theorem.

**Lemma 6.** *There are $\pi_1, \pi_2 \in \Pi$ such that*

$$v_{\mathrm{R1}}^*(s) = \max_{\tau \in \mathbb{N}} r_1\left(\xi_s^{\pi_1}(\tau)\right) \ \text{and} \ v_{\mathrm{R2}}^*(s) = \max_{\tau \in \mathbb{N}} r_2\left(\xi_s^{\pi_2}(\tau)\right)$$

*for all $s \in \mathcal{S}$.*

*Proof.* We will just prove the result for $v_{\mathrm{R1}}^*(s)$ since the other result follows identically. For each $s \in \mathcal{S}$, let $\tau_s$ be the smallest element of $\mathbb{N}$ for which

$$\max_{\mathbf{a} \in \mathbb{A}} r_1\left(\xi_s^{\mathbf{a}}(\tau_s)\right) = v_{\mathrm{R1}}^*(s).$$

Moreover, for each $s \in \mathcal{S}$, let $\mathbf{a}_s$ be such that

$$r_1\left(\xi_s^{\mathbf{a}_s}(\tau_s)\right) = v_{\mathrm{R1}}^*(s).$$

Let $\pi_1 \in \Pi$ be given by $\pi_1(s) = \mathbf{a}_s(0)$. It suffices to show that

$$r_1\left(\xi_s^{\pi_1}(\tau_s)\right) = v_{\mathrm{R1}}^*(s) \tag{32}$$

for all $s \in \mathcal{S}$, for in this case, we have

$$v_{\mathrm{R1}}^*(s) \geq \max_{\tau \in \mathbb{N}} r_1\left(\xi_s^{\pi_1}(\tau)\right) \geq r_1\left(\xi_s^{\pi_1}(\tau_s)\right) = v_{\mathrm{R1}}^*(s) \quad \forall s \in \mathcal{S}.$$

We show (32) holds for each $s \in \mathcal{S}$ by induction on $\tau_s$. First, suppose that $s \in \mathcal{S}$ is such that $\tau_s = 0$. Then

$$r_1\left(\xi_s^{\pi_1}(\tau_s)\right) = r_1(s) = r_1\left(\xi_s^{\mathbf{a}_s}(\tau_s)\right) = v_{\mathrm{R1}}^*(s).$$

For the induction step, let $n \in \mathbb{N}$ and suppose that

$$r_1\left(\xi_s^{\pi_1}(\tau_s)\right) = v_{\mathrm{R1}}^*(s) \quad \forall s \in \mathcal{S} \text{ such that } \tau_s \leq n.$$

Now fix some $x \in \mathcal{S}$ such that $\tau_x = n + 1$. Notice that

$$
\begin{aligned}
v_{\mathrm{R1}}^*(x) &\geq v_{\mathrm{R1}}^* \left( f\left(x, \pi_1(x)\right) \right) \\
&\geq \max_{\mathbf{a} \in \mathbb{A}} r_1 \left( \xi_{f(x, \pi_1(x))}^{\mathbf{a}}(n) \right) \\
&\geq r_1 \left( \xi_{f(x, \pi_1(x))}^{\mathbf{a}_x|_1}(n) \right) \\
&= r_1 \left( \xi_x^{[\pi_1(x), \mathbf{a}_x|_1]}(n+1) \right) \\
&= r_1 \left( \xi_x^{\mathbf{a}_x}(\tau_x) \right) \\
&= v_{\mathrm{R1}}^*(x),
\end{aligned}
$$

so that $v_{\mathrm{R1}}^* \left( f\left(x, \pi_1(x)\right) \right) = v_{\mathrm{R1}}^*(x)$ and $\tau_{f(x, \pi_1(x))} \leq n$. It suffices to show

$$
\tau_{f(x, \pi_1(x))} = n, \tag{33}
$$

for then, by the induction assumption, we have

$$
r_1 \left( \xi_x^{\pi_1}(\tau_x) \right) = r_1 \left( \xi_{f(x, \pi_1(x))}^{\pi_1}(n) \right) = v_{\mathrm{R1}}^* \left( f\left(x, \pi_1(x)\right) \right) = v_{\mathrm{R1}}^*(x).
$$

To show (33), assume instead that

$$
\tau_{f(x, \pi_1(x))} < n.
$$

But

$$
\begin{aligned}
v_{\mathrm{R1}}^*(x) &\geq \max_{\mathbf{a} \in \mathbb{A}} r_1 \left( \xi_x^{\mathbf{a}} \left( \tau_{f(x, \pi_1(x))} + 1 \right) \right) \\
&\geq r_1 \left( \xi_x^{\left[ \pi_1(x), \mathbf{a}_{f(x, \pi_1(x))} \right]} \left( \tau_{f(x, \pi_1(x))} + 1 \right) \right) \\
&= r_1 \left( \xi_{f(x, \pi_1(x))}^{\mathbf{a}_{f(x, \pi_1(x))}} \left( \tau_{f(x, \pi_1(x))} \right) \right) \\
&= v_{\mathrm{R1}}^* \left( f\left(x, \pi_1(x)\right) \right) \\
&= v_{\mathrm{R1}}^*(x),
\end{aligned}
$$

so that

$$
v_{\mathrm{R1}}^*(x) = \max_{\mathbf{a} \in \mathbb{A}} r_1 \left( \xi_x^{\mathbf{a}} \left( \tau_{f(x, \pi_1(x))} + 1 \right) \right)
$$

and thus

$$
\tau_x \leq \tau_{f(x, \pi_1(x))} + 1 < n + 1,
$$

giving our desired contradiction. $\qquad \square$

**Corollary 6.** *For all $s \in \mathcal{S}$, we have $V_{\mathrm{R1}}^*(s) = v_{\mathrm{R1}}^*(s)$ and $V_{\mathrm{R2}}^*(s) = v_{\mathrm{R2}}^*(s)$.*

**Lemma 7.** *There is a $\pi \in \Pi$ such that*

$$
\tilde{v}_{\mathrm{R}}^*(s) = \max_{\tau \in \mathbb{N}} \tilde{r}_{\mathrm{RR}} \left( \xi_s^\pi(\tau) \right).
$$

*for all $s \in \mathcal{S}$.*

*Proof.* This lemma follows by precisely the same proof as the previous lemma, with $r_1$, $v_{\mathrm{R1}}^*$, and $\pi_1$ replaced with $\tilde{r}_{\mathrm{RR}}$, $\tilde{v}_{\mathrm{R}}^*$, and $\pi$ respectively. $\qquad \square$

**Corollary 7.** *For all $s \in \mathcal{S}$, we have $\tilde{V}_{\mathrm{R}}^*(s) = \tilde{v}_{\mathrm{R}}^*(s)$.*

**Lemma 8.** *Let $\zeta_1 : \mathbb{N} \to \mathbb{R}$ and $\zeta_2 : \mathbb{N} \to \mathbb{R}$. Then*

$$
\sup_{\tau \in \mathbb{N}} \max \left\{ \min \left\{ \zeta_1(\tau), \sup_{\tau' \in \mathbb{N}} \zeta_2(\tau + \tau') \right\}, \min \left\{ \sup_{\tau' \in \mathbb{N}} \zeta_1(\tau + \tau'), \zeta_2(\tau) \right\} \right\}
$$
$$
= \min \left\{ \sup_{\tau \in \mathbb{N}} \zeta_1(\tau), \sup_{\tau \in \mathbb{N}} \zeta_2(\tau) \right\}.
$$

*Proof.* We proceed by showing both inequalities corresponding to the above equality hold.

($\leq$) Observe that

$$\sup_{\tau \in \mathbb{N}} \max \left\{ \min \left\{ \zeta_1(\tau), \sup_{\tau' \in \mathbb{N}} \zeta_2(\tau + \tau') \right\}, \min \left\{ \sup_{\tau' \in \mathbb{N}} \zeta_1(\tau + \tau'), \zeta_2(\tau) \right\} \right\}$$

$$\leq \max \left\{ \min \left\{ \sup_{\tau \in \mathbb{N}} \zeta_1(\tau), \sup_{\tau \in \mathbb{N}} \sup_{\tau' \in \mathbb{N}} \zeta_2(\tau + \tau') \right\}, \min \left\{ \sup_{\tau \in \mathbb{N}} \sup_{\tau' \in \mathbb{N}} \zeta_1(\tau + \tau'), \sup_{\tau \in \mathbb{N}} \zeta_2(\tau) \right\} \right\}$$

$$= \min \left\{ \sup_{\tau \in \mathbb{N}} \zeta_1(\tau), \sup_{\tau \in \mathbb{N}} \zeta_2(\tau) \right\}$$

($\geq$) Fix $\varepsilon > 0$. Choose $\tau_1, \tau_2 \in \mathbb{N}$ such that $\zeta_1(\tau_1) \geq \sup_{\tau \in \mathbb{N}} \zeta_1(\tau) - \varepsilon$ and $\zeta_2(\tau_2) \geq \sup_{\tau \in \mathbb{N}} \zeta_2(\tau) - \varepsilon$. Without loss of generality, we can assume $\tau_1 \leq \tau_2$. Then

$$\sup_{\tau \in \mathbb{N}} \max \left\{ \min \left\{ \zeta_1(\tau), \sup_{\tau' \in \mathbb{N}} \zeta_2(\tau + \tau') \right\}, \min \left\{ \sup_{\tau' \in \mathbb{N}} \zeta_1(\tau + \tau'), \zeta_2(\tau) \right\} \right\}$$

$$\geq \sup_{\tau \in \mathbb{N}} \min \left\{ \zeta_1(\tau), \sup_{\tau' \in \mathbb{N}} \zeta_2(\tau + \tau') \right\}$$

$$\geq \min \left\{ \zeta_1(\tau_1), \sup_{\tau' \in \mathbb{N}} \zeta_2(\tau_1 + \tau') \right\}$$

$$\geq \min \left\{ \zeta_1(\tau_1), \zeta_2(\tau_2) \right\}$$

$$\geq \min \left\{ \sup_{\tau \in \mathbb{N}} \zeta_1(\tau) - \varepsilon, \sup_{\tau \in \mathbb{N}} \zeta_2(\tau) - \varepsilon \right\}$$

$$= \min \left\{ \sup_{\tau \in \mathbb{N}} \zeta_1(\tau), \sup_{\tau \in \mathbb{N}} \zeta_2(\tau) \right\} - \varepsilon.$$

But since $\varepsilon > 0$ was arbitrary, the desired inequality follows.

$\square$

**Lemma 9.** *For each $s \in \mathcal{S}$,*

$$\tilde{v}_\mathrm{R}^*(s) = v_\mathrm{RR}^*(s).$$

*Proof.* For each $s \in \mathcal{S}$,

$$\tilde{v}_{\mathrm{R}}^*(s) = \max_{\mathbf{a} \in \mathbb{A}} \max_{\tau \in \mathbb{N}} \tilde{r}_{\mathrm{RR}} \left( \xi_s^{\mathbf{a}}(\tau) \right) \tag{34}$$

$$= \max_{\mathbf{a} \in \mathbb{A}} \max_{\tau \in \mathbb{N}} \max \left\{ \min \left\{ r_1 \left( \xi_s^{\mathbf{a}}(\tau) \right), v_{\mathrm{R2}}^* \left( \xi_s^{\mathbf{a}}(\tau) \right) \right\}, \min \left\{ v_{\mathrm{R1}}^* \left( \xi_s^{\mathbf{a}}(\tau) \right), r_2 \left( \xi_s^{\mathbf{a}}(\tau) \right) \right\} \right\} \tag{35}$$

$$= \max_{\mathbf{a} \in \mathbb{A}} \max_{\tau \in \mathbb{N}} \max \left\{ \min \left\{ r_1 \left( \xi_s^{\mathbf{a}}(\tau) \right), \max_{\mathbf{a}' \in \mathbb{A}} \max_{\tau' \in \mathbb{N}} r_2 \left( \xi_{\xi_s^{\mathbf{a}}(\tau)}^{\mathbf{a}'}(\tau') \right) \right\}, \right.$$
$$\left. \min \left\{ \max_{\mathbf{a}' \in \mathbb{A}} \max_{\tau' \in \mathbb{N}} r_1 \left( \xi_{\xi_s^{\mathbf{a}}(\tau)}^{\mathbf{a}'}(\tau') \right), r_2 \left( \xi_s^{\mathbf{a}}(\tau) \right) \right\} \right\}$$

$$= \max_{\mathbf{a} \in \mathbb{A}} \max_{\tau \in \mathbb{N}} \max \left\{ \min \left\{ r_1 \left( \xi_s^{\mathbf{a}}(\tau) \right), \max_{\mathbf{a}' \in \mathbb{A}} \max_{\tau' \in \mathbb{N}} r_2 \left( \xi_s^{[\mathbf{a},\mathbf{a}']_\tau}(\tau + \tau') \right) \right\}, \right.$$
$$\left. \min \left\{ \max_{\mathbf{a}' \in \mathbb{A}} \max_{\tau' \in \mathbb{N}} r_1 \left( \xi_s^{[\mathbf{a},\mathbf{a}']_\tau}(\tau + \tau') \right), r_2 \left( \xi_s^{\mathbf{a}}(\tau) \right) \right\} \right\}$$

$$= \max_{\mathbf{a} \in \mathbb{A}} \max_{\tau \in \mathbb{N}} \max_{\mathbf{a}' \in \mathbb{A}} \max \left\{ \min \left\{ r_1 \left( \xi_s^{\mathbf{a}}(\tau) \right), \max_{\tau' \in \mathbb{N}} r_2 \left( \xi_s^{[\mathbf{a},\mathbf{a}']_\tau}(\tau + \tau') \right) \right\}, \right.$$
$$\left. \min \left\{ \max_{\tau' \in \mathbb{N}} r_1 \left( \xi_s^{[\mathbf{a},\mathbf{a}']_\tau}(\tau + \tau') \right), r_2 \left( \xi_s^{\mathbf{a}}(\tau) \right) \right\} \right\}$$

$$= \max_{\mathbf{a} \in \mathbb{A}} \max_{\tau \in \mathbb{N}} \max_{\mathbf{a}' \in \mathbb{A}} \max \left\{ \min \left\{ r_1 \left( \xi_s^{[\mathbf{a},\mathbf{a}']_\tau}(\tau) \right), \max_{\tau' \in \mathbb{N}} r_2 \left( \xi_s^{[\mathbf{a},\mathbf{a}']_\tau}(\tau + \tau') \right) \right\}, \right.$$
$$\left. \min \left\{ \max_{\tau' \in \mathbb{N}} r_1 \left( \xi_s^{[\mathbf{a},\mathbf{a}']_\tau}(\tau + \tau') \right), r_2 \left( \xi_s^{[\mathbf{a},\mathbf{a}']_\tau}(\tau) \right) \right\} \right\} \tag{36}$$

$$= \max_{\mathbf{a} \in \mathbb{A}} \max_{\tau \in \mathbb{N}} \max \left\{ \min \left\{ r_1 \left( \xi_s^{\mathbf{a}}(\tau) \right), \max_{\tau' \in \mathbb{N}} r_2 \left( \xi_s^{\mathbf{a}}(\tau + \tau') \right) \right\}, \right.$$
$$\left. \min \left\{ \max_{\tau' \in \mathbb{N}} r_1 \left( \xi_s^{\mathbf{a}}(\tau + \tau') \right), r_2 \left( \xi_s^{\mathbf{a}}(\tau) \right) \right\} \right\} \tag{37}$$

$$= \max_{\mathbf{a} \in \mathbb{A}} \min \left\{ \max_{\tau \in \mathbb{N}} r_1 \left( \xi_s^{\mathbf{a}}(\tau) \right), \max_{\tau \in \mathbb{N}} r_2 \left( \xi_s^{\mathbf{a}}(\tau) \right) \right\} \tag{38}$$

$$= v_{\mathrm{RR}}^*(s),$$

where the equality between 34 and 35 follows from Corollary 6, the equality between 36 and 37 follows from Lemma 3, and the equality between 37 and 38 follows from Lemma 8. $\quad\square$

Before the next lemma, we need to introduce two last pieces of notation. First, we let $\overline{\overline{\Pi}}$ be the set of augmented policies $\bar{\pi} : \mathcal{S} \times \mathcal{Y} \times \mathcal{Z} \to \mathcal{A}$, as in the previous section, but where

$$\mathcal{Y} = \{ r_1(s) \mid s \in \mathcal{S} \} \quad \text{and} \quad \mathcal{Z} = \{ r_2(s) \mid s \in \mathcal{S} \}.$$

Next, given $s \in \mathcal{S}$, $y \in \mathcal{Y}$, $z \in \mathcal{Z}$, and $\bar{\pi} \in \overline{\overline{\Pi}}$, we let $\bar{\xi}_s^{\bar{\pi}} : \mathbb{N} \to \mathcal{S}$, $\bar{\eta}_s^{\bar{\pi}} : \mathbb{N} \to \mathcal{Y}$, and $\bar{\zeta}_s^{\bar{\pi}} : \mathbb{N} \to \mathcal{Z}$, be the solution of the evolution

$$\bar{\xi}_s^{\bar{\pi}}(t+1) = f \left( \bar{\xi}_s^{\bar{\pi}}(t), \bar{\pi} \left( \bar{\xi}_s^{\bar{\pi}}(t), \bar{\eta}_s^{\bar{\pi}}(t), \bar{\zeta}_s^{\bar{\pi}}(t) \right) \right),$$
$$\bar{\eta}_s^{\bar{\pi}}(t+1) = \max \left\{ r_1 \left( \bar{\xi}_s^{\bar{\pi}}(t+1) \right), \bar{\eta}_s^{\bar{\pi}}(t) \right\},$$
$$\bar{\zeta}_s^{\bar{\pi}}(t+1) = \max \left\{ r_2 \left( \bar{\xi}_s^{\bar{\pi}}(t+1) \right), \bar{\zeta}_s^{\bar{\pi}}(t) \right\},$$

for which $\bar{\xi}_s^{\bar{\pi}}(0) = s$, $\bar{\eta}_s^{\bar{\pi}}(0) = r_1(s)$, and $\bar{\xi}_s^{\bar{\pi}}(0) = r_2(s)$.

**Lemma 10.** *There is a $\bar{\pi} \in \overline{\overline{\Pi}}$ such that*

$$v_{\mathrm{RR}}^*(s) = \min \left\{ \max_{\tau \in \mathbb{N}} r_1 \left( \bar{\xi}_s^{\bar{\pi}}(\tau) \right), \max_{\tau \in \mathbb{N}} r_2 \left( \bar{\xi}_s^{\bar{\pi}}(\tau) \right) \right\}$$

*for all $s \in \mathcal{S}$.*

*Proof.* By Lemmas 6 and 7 together with Corollary 6, we can choose $\pi, \theta_1, \theta_2 \in \Pi$ such that

$$v_{\text{R1}}^*(s) = \max_{\tau \in \mathbb{N}} r_1\left(\xi_s^{\theta_1}(\tau)\right) \quad \forall s \in \mathcal{S},$$

$$v_{\text{R2}}^*(s) = \max_{\tau \in \mathbb{N}} r_2\left(\xi_s^{\theta_2}(\tau)\right) \quad \forall s \in \mathcal{S},$$

$$\tilde{v}_{\text{R}}^*(s) = \max_{\tau \in \mathbb{N}} \max\left\{\min\left\{r_1\left(\xi_s^{\pi}(\tau)\right), v_{\text{R2}}^*\left(\xi_s^{\pi}(\tau)\right)\right\}, \min\left\{r_2\left(\xi_s^{\pi}(\tau)\right), v_{\text{R1}}^*\left(\xi_s^{\pi}(\tau)\right)\right\}\right\} \quad \forall s \in \mathcal{S}.$$

Define $\bar{\pi} \in \overline{\Pi}$ by

$$\bar{\pi}(s, y, z) = \begin{cases} \pi(s) & \max\{y, z\} < \tilde{v}_{\text{R}}^*(s) \\ \theta_1(s) & \max\{y, z\} \geq \tilde{v}_{\text{R}}^*(s) \text{ and } y \leq z, \\ \theta_2(s) & \max\{y, z\} \geq \tilde{v}_{\text{R}}^*(s) \text{ and } y > z. \end{cases}$$

Now fix some $s \in \mathcal{S}$. For all $t \in \mathbb{N}$, set $\bar{x}_t = \bar{\xi}_s^{\bar{\pi}}(t)$, $\bar{y}_t = \bar{\eta}_s^{\bar{\pi}}(t) = \max_{\tau \leq t} r_1(\bar{x}_\tau)$, and $\bar{z}_t = \bar{\zeta}_s^{\bar{\pi}}(t) = \max_{\tau \leq t} r_2(\bar{x}_\tau)$, and also set $x_t^\circ = \xi_s^\pi(t)$. It suffices to show

$$v_{\text{RR}}^*(s) \leq \min\left\{\max_{\tau \in \mathbb{N}} r_1\left(\bar{x}_\tau\right), \max_{\tau \in \mathbb{N}} r_2\left(\bar{x}_\tau\right)\right\}, \tag{39}$$

since the reverse inequality is immediate. We proceed in three steps.

1. We claim there exists a $t \in \mathbb{N}$ such that $\max\left\{r_1(\bar{x}_t), r_2(\bar{x}_t)\right\} \geq \tilde{v}_{\text{R}}^*(\bar{x}_t)$.

   Suppose otherwise. Then $\bar{\pi}(\bar{x}_t, \bar{y}_t, \bar{z}_t) = \pi(\bar{x}_t)$ so that $\bar{x}_t = x_t^\circ$ for all $t \in \mathbb{N}$. Thus

   $$\max_{t \in \mathbb{N}} \max\left\{r_1(\bar{x}_t), r_2(\bar{x}_t)\right\} < \max_{t \in \mathbb{N}} \tilde{v}_{\text{R}}^*(\bar{x}_t)$$
   $$= \tilde{v}_{\text{R}}^*(s)$$
   $$= \max_{\tau \in \mathbb{N}} \max\left\{\min\left\{r_1\left(x_\tau^\circ\right), v_{\text{R2}}^*\left(x_\tau^\circ\right)\right\}, \min\left\{r_2\left(x_\tau^\circ\right), v_{\text{R1}}^*\left(x_\tau^\circ\right)\right\}\right\}$$
   $$= \max_{\tau \in \mathbb{N}} \max\left\{\min\left\{r_1\left(\bar{x}_\tau\right), v_{\text{R2}}^*\left(\bar{x}_\tau\right)\right\}, \min\left\{r_2\left(\bar{x}_\tau\right), v_{\text{R1}}^*\left(\bar{x}_\tau\right)\right\}\right\}$$
   $$\leq \max_{\tau \in \mathbb{N}} \max\left\{r_1(\bar{x}_\tau), r_2(\bar{x}_\tau)\right\},$$

   providing the desired contradiction.

2. Let $T$ be the smallest element of $\mathbb{N}$ for which

   $$\max\left\{r_1(\bar{x}_T), r_2(\bar{x}_T)\right\} \geq v_{\text{R}}^*(\bar{x}_T),$$

   which must exist by the previous step, and let $T'$ be the smallest element of $\mathbb{N}$ for which

   $$\max\left\{\min\left\{r_1\left(x_{T'}^\circ\right), v_{\text{R2}}^*\left(x_{T'}^\circ\right)\right\}, \min\left\{r_2\left(x_{T'}^\circ\right), v_{\text{R1}}^*\left(x_{T'}^\circ\right)\right\}\right\} = \tilde{v}_{\text{R}}^*(s),$$

   which must exist by our choice of $\pi$. We claim $T' \geq T$.

   Suppose otherwise. Since $\bar{x}_t = x_t^\circ$ for all $t \leq T$, then in particular $\bar{x}_{T'} = x_{T'}^\circ$, so that

   $$\max\left\{\min\left\{r_1\left(\bar{x}_{T'}\right), v_{\text{R2}}^*\left(\bar{x}_{T'}\right)\right\}, \min\left\{r_2\left(\bar{x}_{T'}\right), v_{\text{R1}}^*\left(\bar{x}_{T'}\right)\right\}\right\} = \tilde{v}_{\text{R}}^*(s).$$

   But then

   $$\max\{r_1(\bar{x}_{T'}), r_2(\bar{x}_{T'})\} \geq \tilde{v}_{\text{R}}^*(s) \geq \tilde{v}_{\text{R}}^*(\bar{x}_{T'}).$$

   By our choice of $T$, we then have $T \leq T'$, creating a contradiction.

3. It follows from the previous step that

   $$\tilde{v}_{\text{R}}^*(\bar{x}_T) = \tilde{v}_{\text{R}}^*(x_T^\circ) = \tilde{v}_{\text{R}}^*(s).$$

   By our choice of $T$, there are two cases: $r_1(\bar{x}_T) \geq \tilde{v}_{\text{R}}^*(\bar{x}_T)$ and $r_2(\bar{x}_T) \geq \tilde{v}_{\text{R}}^*(\bar{x}_T)$. We assume the first case and prove the desired result, with case two following identically. To reach a contradiction, assume

   $$r_2(\bar{x}_t) < \tilde{v}_{\text{R}}^*(\bar{x}_T) \quad \forall t \in \mathbb{N}.$$

But then $\bar{\pi}(\bar{x}_t, \bar{y}_t, \bar{z}_t) = \theta_2(\bar{x}_t)$ for all $t \geq T$, so $v_{R2}^*(\bar{x}_T) = \max_{t \geq T} r_2(\bar{x}_t) < \tilde{v}_R^*(\bar{x}_T) \leq \tilde{v}_R^*(s)$. Thus $r_2(x_{T'}^\circ) \leq v_{R2}^*(x_{T'}^\circ) \leq v_{R2}^*(x_T^\circ) = v_{R2}^*(\bar{x}_T) < \tilde{v}_R^*(s)$. It follows that

$$\max\left\{\min\left\{r_1\left(x_{T'}^\circ\right), v_{R2}^*\left(x_{T'}^\circ\right)\right\}, \min\left\{r_2\left(x_{T'}^\circ\right), v_{R1}^*\left(x_{T'}^\circ\right)\right\}\right\} < \tilde{v}_R^*(s),$$

contradicting our choice of $T'$.

Thus $r_2(\bar{x}_t) \geq \tilde{v}_R^*(\bar{x}_T) = \tilde{v}_R^*(s)$ for some $t \in \mathbb{N}$ and also $r_1(\bar{x}_T) \geq \tilde{v}_R^*(\bar{x}_T) = \tilde{v}_R^*(s)$, so that (39) must hold by Lemma 9.

$\square$

**Corollary 8.** *For all $s \in \mathcal{S}$, we have $V_{RR}^*(s, r_1(s), r_2(s)) = v_{RR}^*(s)$.*

*Proof of Theorem 2.* The proof of this theorem immediately follows from the previous corollary together with Corollary 7 and Lemma 9. $\square$

## C    PROOF OF OPTIMALITY THEOREM

*Proof of Theorem 3.* The inequalities in both lines of the theorem follow from the fact that for each $\pi \in \Pi$, we can define a corresponding augmented policy $\bar{\pi} \in \overline{\Pi}$ by

$$\bar{\pi}(s, y, z) = \pi(s) \quad \forall s \in \mathcal{S}, y \in \mathcal{Y}, z \in \mathcal{Z},$$

in which case $V_{RAA}^\pi(s) = V_{RAA}^{\bar{\pi}}(s)$ and $V_{RR}^\pi(s) = V_{RR}^{\bar{\pi}}(s)$ for each $s \in \mathcal{S}$. Note that in general, we cannot define a corresponding policy for each augmented policy, so the reverse inequality does not generally hold (see Figure 3 for intuition regarding this fact).

The equalities in both lines of the theorem are simply restatements of Lemma 5 and Lemma 9. $\square$

## D    THE SRABE AND ITS POLICY GRADIENT

We first justify the SRABE from a theoretical perspective. For each $\gamma \in (0, 1)$ and stochastic policy $\pi : \mathcal{S} \to \Delta(\mathcal{A})$ (with $\Delta(\mathcal{A})$ the probability simplex on $\mathcal{A}$), let $\tilde{V}_{RA}^{\gamma,\pi}, V_{RA}^{\gamma,*} : \mathcal{S} \to \mathbb{R}$ be the (unique) solutions of the Bellman equations

$$\tilde{V}_{RA}^{\gamma,\pi}(s) = (1-\gamma)\min\{r(s), q(s)\} + \gamma\mathbb{E}_{a\sim\pi}\left[\min\left\{\max\left\{\tilde{V}_{RA}^{\gamma,\pi}\left(f(s,a)\right), r(s)\right\}, q(s)\right\}\right],$$

$$V_{RA}^{\gamma,*}(s) = (1-\gamma)\min\{r(s), q(s)\} + \gamma\left[\min\left\{\max\left\{\max_{a\in\mathcal{A}} V_{RA}^{\gamma,*}\left(f(s,a)\right), r(s)\right\}, q(s)\right\}\right],$$

respectively, where $r : \mathcal{S} \to \mathbb{R}$ and $q : \mathcal{S} \to \mathbb{R}$. Note that the above Bellman equations are indeed $\gamma$-contractive.

**Theorem 4.** *Let $\gamma \in (0, 1)$. Given any $\pi : \mathcal{S} \to \Delta(\mathcal{A})$, we have*

$$\tilde{V}_{RA}^{\gamma,\pi} \leq V_{RA}^{\gamma,*}.$$

*Moreover, there exists a $\pi^* : \mathcal{S} \to \Delta(\mathcal{A})$ such that*

$$\tilde{V}_{RA}^{\gamma,\pi^*} = V_{RA}^{\gamma,*}.$$

*Proof.* Let $\pi : \mathcal{S} \to \Delta(\mathcal{A})$. Define the Bellman operators $B_\gamma^\pi, B_\gamma^* : \mathbb{R}^{\mathcal{S}} \to \mathbb{R}^{\mathcal{S}}$ (where $\mathbb{R}^{\mathcal{S}}$ is the set of all maps $v : \mathcal{S} \to \mathbb{R}$) by

$$B_\gamma^\pi[v](s) = (1-\gamma)\min\{r(s), q(s)\} + \gamma\mathbb{E}_{a\sim\pi}\left[\min\left\{\max\left\{v\left(f(s,a)\right), r(s)\right\}, q(s)\right\}\right],$$

$$B_\gamma^*[v](s) = (1-\gamma)\min\{r(s), q(s)\} + \gamma\left[\min\left\{\max\left\{\max_{a\in\mathcal{A}} v\left(f(s,a)\right), r(s)\right\}, q(s)\right\}\right],$$

respectively. For each $v \in \mathbb{R}^{\mathcal{S}}$, we have $B_\gamma^\pi[v] \leq B_\gamma^*[v]$. Then $\tilde{V}_{RA}^{\gamma,\pi} = B_\gamma^\pi[\tilde{V}_{RA}^{\gamma,\pi}] \leq B_\gamma^*[\tilde{V}_{RA}^{\gamma,\pi}]$ and $V_{RA}^{\gamma,*} = B_\gamma^*[V_{RA}^{\gamma,*}]$.

Since $B_\gamma^*$ is a contraction and also a monotonic operator ($B_\gamma^*[v] \leq B_\gamma^*[w]$ when $v \leq w$), it follows from the comparison principle for Bellman operators that $\tilde{V}_{\text{RA}}^{\gamma,\pi} \leq V_{\text{RA}}^{\gamma,*}$.

Now let $\pi^* : \mathcal{S} \to \mathcal{A}$ be such that $\pi^*(s)$ is supported on $\arg\max_{a \in \mathcal{A}} v(f(s,a))$. Then $\tilde{V}_{\text{RA}}^{\gamma,\pi^*} = B_\gamma^{\pi^*}[\tilde{V}_{\text{RA}}^{\gamma,\pi^*}] = B_\gamma^*[\tilde{V}_{\text{RA}}^{\gamma,\pi^*}]$ and $V_{\text{RA}}^{\gamma,*} = B_\gamma^*[V_{\text{RA}}^{\gamma,*}]$, so that $\tilde{V}_{\text{RA}}^{\gamma,\pi^*} = V_{\text{RA}}^{\gamma,*}$. $\qquad\square$

To understand the significance of the above theorem, recall that when solving RA problems (as is needed during our solution of the RAA problem) we are interested in estimating $V_{\text{RA}}^{\gamma,*}$ in the limit $\gamma \to 1^-$ (see Proposition 3 in Fisac et al. (2019)). The above theorem tells us that to obtain $V_{\text{RA}}^{\gamma,*}$ we can search for a (possibly stochastic) policy $\pi$ that maximizes $\tilde{V}_{\text{RA}}^{\gamma,\pi}$. Doing so allows us to use the PPO adaptation described in the DOHJ-PPO algorithm for finding RA value functions. Analogous results hold for the R and A subproblems.

### D.1 POLICY GRADIENT

By analogy to the SRBE, the SRABE is given by

$$\hat{V}_{\text{RAA}}^\pi(s) = \mathbb{E}_{a \sim \pi}\left[\min\left\{\max\left\{\hat{V}_{\text{RAA}}^\pi\left(f(s,a)\right), \tilde{r}_{\text{RAA}}(s)\right\}, q(s)\right\}\right]. \qquad \text{(SRABE)}$$

The corresponding action-value function is

$$\hat{Q}_{\text{RAA}}^\pi(s,a) = \min\left\{\max\left\{\hat{V}_{\text{RAA}}^\pi\left(f(s,a)\right), \tilde{r}_{\text{RAA}}(s)\right\}, q(s)\right\}.$$

We define a modification of the dynamics $f$ involving an absorbing state $s_\infty$ as follows:

$$f'(s,a) = \begin{cases} f(s,a) & q\left(f(s,a)\right) < \hat{V}_{\text{RAA}}^\pi(s) < \tilde{r}_{\text{RAA}}\left(f(s,a)\right), \\ s_\infty & \text{otherwise.} \end{cases}$$

We then have the following proposition:

**Proposition 1.** *For each $s \in \mathcal{S}$ and every $\theta \in \mathbb{R}^{n_p}$, we have*

$$\nabla_\theta \hat{V}_{\text{RAA}}^{\pi_\theta}(s) \propto \mathbb{E}_{s' \sim d'_\pi(s), a \sim \pi_\theta}\left[\hat{Q}_{\text{RAA}}^{\pi_\theta}(s',a)\nabla_\theta \ln \pi_\theta(a|s')\right],$$

*where $d'_\pi(s)$ is the stationary distribution of the Markov Chain with transition function*

$$P(s'|s) = \sum_{a \in \mathcal{A}} \pi(a|s)\left[f'(s,\pi(a|s)) = s'\right],$$

*with the bracketed term equal to $1$ if the proposition inside is true and $0$ otherwise.*

Following Hsu et al. (2021), we then define the discounted value and action-value functions with $\gamma \in [0,1)$.

*Proof Sketch of Proposition 1 (adapted from So et al. (2024)).* We here closely follow the proof of Theorem 3 in So et al. (2024), which itself modifies the proofs of the Policy Gradient Theorems in Chapter 13.2 and 13.6 Sutton and Barto (2018). We only make the minimal modifications required to

adapt the PPO algorithm developed previously for the SRBE to on for the SRABE.

$$
\begin{aligned}
\nabla_\theta \hat{V}_{\text{RAA}}^{\pi_\theta}(s) =& \nabla_\theta \left( \sum_{a \in \mathcal{A}} \pi_\theta(a|s) \hat{Q}_{\text{RAA}}^{\pi_\theta}(s,a) \right) \\
=& \sum_{a \in \mathcal{A}} \left( \nabla_\theta \pi_\theta(a|s) \hat{Q}_{\text{RAA}}^{\pi_\theta}(s,a) \right. \\
& \left. + \pi_\theta(a|s) \nabla_\theta \min \left\{ \max \left\{ \hat{V}_{\text{RAA}}^\pi \left( f(s,a) \right), \tilde{r}_{\text{RAA}}(s) \right\}, q(s) \right\} \right) \\
=& \sum_{a \in \mathcal{A}} \left( \nabla_\theta \pi_\theta(a|s) \hat{Q}_{\text{RAA}}^{\pi_\theta}(s,a) \right. \\
& \left. + \pi_\theta(a|s) \left[ q(s) < \hat{V}_{\text{RAA}} \left( f(s,a) \right) < \tilde{r}_{\text{RAA}}(s) \right] \nabla_\theta \hat{V}_{\text{RAA}}^\pi \left( f(s,a) \right) \right) \quad (40) \\
=& \sum_{s' \in \mathcal{S}} \left[ \left( \sum_{k=0}^\infty \Pr(s \to s', k, \pi) \right) \sum_{a \in \mathcal{A}} \nabla_\theta \pi_\theta(a|s') \hat{Q}_{\text{RAA}}^{\pi_\theta}(s',a) \right] \quad (41) \\
=& \sum_{s' \in \mathcal{S}} \left[ \left( \sum_{k=0}^\infty \Pr(s \to s', k, \pi) \right) \sum_{a \in \mathcal{A}} \pi_\theta(a|s') \frac{\nabla_\theta \pi_\theta(a|s')}{\pi_\theta(a|s')} \hat{Q}_{\text{RAA}}^{\pi_\theta}(s',a) \right] \\
=& \sum_{s' \in \mathcal{S}} \left[ \left( \sum_{k=0}^\infty \Pr(s \to s', k, \pi) \right) \mathbb{E}_{a \sim \pi_\theta(s')} \left[ \nabla_\theta \ln \pi_\theta(a|s') \hat{Q}_{\text{RAA}}^{\pi_\theta}(s',a) \right] \right] \\
\propto& \ \mathbb{E}_{s' \sim d'_\pi(s)} \mathbb{E}_{a \sim \pi_\theta(s')} \left[ \nabla_\theta \ln \pi_\theta(a|s') \hat{Q}_{\text{RAA}}^{\pi_\theta}(s',a) \right],
\end{aligned}
$$

where the equality between (40) and (41) comes from rolling out the term $\nabla_\theta \hat{V}_{\text{RAA}}^\pi \left( f(s,a) \right)$ (see Chapter 13.2 in Sutton and Barto (2018) for details), and where $\Pr(s \to s', k, \pi)$ is the probability that under the policy $\pi$, the system is in state $s'$ at time $k$ given that it is in state $s$ at time 0. $\qquad \square$

Note, Proposition 1 is vital to updating the actor in Algorithm 1.

## E  THE DOHJ-PPO ALGORITHM

In this section, we outline the details of our Actor-Critic algorithm DOHJ-PPO beyond the details given in Algorithm 1.

In Algorithm 1, the Bellman update $B^\gamma[\tilde{Q}, \tilde{r}]$ differs for the RAA task and RR task, and the $B_i^\gamma[\tilde{Q}]$ differs between the reach, avoid, and reach-avoid tasks.

### E.1  THE SPECIAL BELLMAN UPDATES AND THE CORRESPONDING GAES

Akin to previous HJ-RL policy algorithms, namely RCPO Yu et al. (2022b), RESPO Ganai et al. (2023) and RCPPO So et al. (2024), DOHJ-PPO fundamentally depends on the discounted HJ Bellman updates Fisac et al. (2019). To solve the RAA and RR problems with the special rewards defined in Theorems 1 & 2, DOHJ-PPO utilizes the Reach, Avoid and Reach-Avoid Bellman updates, given by

$$
B_R^\gamma[Q \mid r](s,a) = (1-\gamma)r(s) + \gamma \max \left\{ r(s), Q(s,a) \right\}, \tag{42}
$$

$$
B_A^\gamma[Q \mid q](s,a) = (1-\gamma)q(s) + \gamma \min \left\{ q(s), Q(s,a) \right\}, \tag{43}
$$

$$
B_{RA}^\gamma[Q \mid r, q](s,a) = (1-\gamma)\min \left\{ r(s), q(s) \right\} + \gamma \min \left\{ q(s), \max \left\{ r(s), Q(s,a) \right\} \right\}. \tag{44}
$$

To improve our algorithm, we incorporate the Generalized Advantage Estimate corresponding to these Bellman equations in the updates of the Actors. As outlined in Section A of So et al. (2024), the GAE may be defined with a reduction function corresponding to the appropriate Bellman function which will be applied over a trajectory roll-out. We generalize the Reach GAE definition given in So et al. (2024) to propose a Reach-Avoid GAE (the Avoid GAE is simply the flip of the Reach GAE)

---

**Algorithm 1** : DOHJ-PPO (Actor-Critic)

---

**Require:** Composed and Decomposed Actor parameters $\theta$ and $\theta_i$, Composed and Decomposed Critic parameters $\omega$ and $\omega_i$, GAE $\lambda$, learning rate $\beta_k$ and discount factor $\gamma$. Let $B^\gamma$ amd $B_i^\gamma$ represent the Bellman update and decomposed Bellman update for the users choice of problem (RR or RAA).

1: Define *Composed* Actor and Critic $\tilde{Q}$
2: Define *Decomposed* Actor(s) and Critic(s) $\tilde{Q}_i$
3: **for** $k = 0, 1, \ldots$ **do**
4:      **for** $t = 0$ to $T - 1$ **do**
5:          Sample trajectories for $\tau_t : \{\hat{s}_t, a_t, \hat{s}_{t+1}\}$
6:          Define $\tilde{\ell}(s_t)$ with Decomposed Critics $\tilde{Q}_i(s_t)$ (Theorems 1 & 2)
7:          **Composed Critic update:**

$$\omega \leftarrow \omega - \beta_k \nabla_\omega \tilde{Q}(\tau_t) \cdot \left( \tilde{Q}(\tau_t) - B^\gamma[\tilde{Q}, \tilde{r}](\tau_t) \right)$$

8:          Compute Bellman-GAE $A_{HJ}^\lambda$ with $B^\gamma$
9:          (Standard) update Composed Actor
10:         **Decomposed Critic update(s):**

$$\omega \leftarrow \omega - \beta_k \nabla_\omega \tilde{Q}_i(\tau_t) \cdot \left( \tilde{Q}_i(\tau_t) - B_i^\gamma[\tilde{Q}_i](\tau_t) \right)$$

11:         Compute Bellman-GAE $A_i^\lambda$ with $B_i^\gamma$
12:         (Standard) update Decomposed Actor(s)
13:      **end for**
14: **end for**
15: **return** parameter $\theta, \omega$

---

as all will be used in DOHJ-PPO algorithm for either RAA or RR problems. Consider a reduction function $\phi_{RA}^{(n)} : \mathbb{R}^n \to \mathbb{R}$, defined by

$$\phi_{RA}^{(n)}(x_1, x_2, x_3, \ldots, x_{2n+1}) = \phi_{RA}^{(1)}(x_1, x_2, \phi_{RA}^{(n-1)}(x_3, \ldots, x_{2n+1})), \tag{45}$$

$$\phi_{RA}^{(1)}(x, y, z) = (1 - \gamma) \min \{x, y\} + \gamma \min \{y, \max \{x, z\}\}. \tag{46}$$

The $k$-step Reach-Avoid Bellman advantage $A_{RA}^{\pi(k)}$ is then given by,

$$A_{RA}^{(k)}(s) = \phi_{RA}^{(n)} \left( r(s_t), q(s_t), \ldots, r(s_{t+k-1}), q(s_{t+k-1}), V^(s_{t+k}) \right) - V^(s_{t+k}). \tag{47}$$

We may then define the Reach-Avoid GAE $A_{RA}^\lambda$ as the $\lambda$-weighted sum over the advantage functions

$$A_{RA}^\lambda(s) = \frac{1}{1 - \lambda} \sum_{k=1}^\infty \lambda^k A_{RA}^{(k)}(s) \tag{48}$$

which may be approximated over any finite trajectory sample. See So et al. (2024) for further details.

### E.2   Modifications from standard PPO

To address the RAA and RR problems, DOHJ-PPO introduces several key modifications to the standard PPO framework Schulman et al. (2017):

**Additional actor and critic networks are introduced to represent the decomposed objectives.** Rather than learning the decomposed objectives separately from the composed objective, DOHJ-PPO optimizes all objectives simultaneously. This design choice is motivated by two primary factors: (i) simplicity and minor computational speed-up, and (ii) coupling between the decomposed and composed objectives during learning.

**The decomposed trajectories are initialized using states sampled from the composed trajectory**, we refer to as *coupled resets*.

While it is possible to estimate the decomposed objectives independently—i.e., prior to solving the composed task—this approach might lead to inaccurate or irrelevant value estimates in on-policy settings. For example, in the RAA problem, the decomposed objective may prioritize avoiding penalties, while the composed task requires reaching a reward region without incurring penalties. In such a case, a decomposed policy trained in isolation might converge to an optimal strategy within a reward-irrelevant region, misaligned with the overall task. Empirically, we observe that omitting coupled resets causes DOHJ-PPO to perform no better than standard baselines such as CPPO, whereas their inclusion significantly improves performance.

**The special RAA and RR rewards are defined using the decomposed critic values and updated using their corresponding Bellman equations.**
This procedure is directly derived from our theoretical results (Theorems 1 and 2), which establish the validity of using modified rewards within the respective RA and R Bellman frameworks. These rewards are used to compute the composed critic target as well as the actor's GAE. In Algorithm 1, this process is reflected in the critic and actor updates corresponding to the composed objective.

# F  DDQN DEMONSTRATION

As described in the paper, we demonstrate the novel RAA and RR problems in a 2D $Q$-learning problem where the value function may be observed easily. We juxtapose these solitons with those of the previously studied RA and R problems which consider more simple objectives. To solve all values, we employ the standard Double-Deep $Q$ learning approach (DDQN) Van Hasselt et al. (2016) with only the special Bellman updates.

## F.1  GRID-WORLD ENVIRONMENT

The environment is taken from Hsu et al. (2021) and consists of two dimensions, $s = (x, y)$, and three actions, $a \in \{\text{left}, \text{straight}, \text{right}\}$, which allow the agent to maneuver through the space. The deterministic dynamics of the environment are defined by constant upward flow such that,

$$f((x_i, y_i), a_i) = \begin{cases} (x_{i-1}, y_{i+1}) & a_i = \text{left} \\ (x_i, y_{i+1}) & a_i = \text{straight} \\ (x_{i+1}, y_{i+1}) & a_i = \text{right} \end{cases} \tag{49}$$

and if the agent reaches the boundary of the space, defined by $x \geq |2|$, $y \leq -2$ and $y \geq 10$, the trajectory is terminated. The 2D space is divided into $80 \times 120$ cells which the agent traverses through.

**In the RA and RAA experiments**, the reward function $r$ is defined as the negative signed-distance function to a box with dimensions $(x_c, y_c, w, h) = (0, 4.5, 2, 1.5)$, and thus is negative iff the agent is outside of the box. The penalty function $q$ is defined as the minimum of three (positive) signed distance functions for boxes defined at $(x_c, y_c, w, h) = (\pm 0.75, 3, 1, 1)$ and $(x_c, y_c, w, h) = (0, 6, 2.5, 1)$, and thus is positive iff the agent is outside of all boxes.

**In the R and RR experiments**, one or two rewards are used. In the R experiment, the reward function $r$ is defined as the maximum of two negative signed-distance function of boxes with dimensions $(x_c, y_c, w, h) = (\pm 1.25, 0, 0.5, 2)$, and thus is negative iff the agent is outside of both boxes. In the RR experiment, the rewards $r_1$ and $r_2$ are defined as the negative signed distance functions of the same two boxes independently, and thus are positive if the agent is in one box or the other respectively.

## F.2  DDQN DETAILS

As per our theoretical results in Theorems 1 and 2, we may now perform DDQN to solve the RAA and RR problems with solely the previously studied Bellman updates for the RA Hsu et al. (2021) and R problems Fisac et al. (2019). We compare these solutions with those corresponding to the RA and R problems *without* the special RAA and RR targets, and hence solve the previously posed problems. For all experiments, we employ the same adapted algorithm as in Hsu et al. (2021), with no modification of the hyper-parameters given in Table 1.

Table 1: Hyperparameters for DDQN Grid World

| DDQN hyperparameters | Values |
| --- | --- |
| Network Architecture | MLP |
| Numbers of Hidden Layers | 2 |
| Units per Hidden Layer | 100, 20 |
| Hidden Layer Activation Function | tanh |
| Optimizer | Adam |
| Discount factor $\gamma$ | 0.9999 |
| Learning rate | 1e-3 |
| Replay Buffer Size | 1e5 transitions |
| Replay Batch Size | 100 |
| Train-Collect Interval | 10 |
| Max Updates | 4e6 |

## G    BASELINES

In both RAA and RR problems, we employ Constrained PPO (CPPO) Achiam et al. (2017a) as the major baseline as it can handle secondary objectives which are reformulated as constraints. The algorithm was not designed to minimize its constraints necessarily but may do so in attempting to satisfy them. As a novel direction in RL, few algorithms have been designed to optimize max/min accumulated costs and thus CPPO serves as the best proxy. Below we also include a naively decomposed STL algorithm to offer some insight into direct approaches to optimizing the max/min accumulated reward.

### G.1    CPPO BASELINES

Although CPPO formulations do not directly consider dual-objective optimization, the secondary objective in RAA (avoid penalty) or overall objective in RR (reach both rewards) may be transformed into constraints to be satisfied of a surrogate problem. For the RAA problem, this may be defined as

$$\max_\pi \mathbb{E}_\pi \left[ \sum_t^\infty \gamma^t \max_{t' \leq t} r(s_{t'}^\pi) \right] \quad \text{s.t.} \quad \min_t q(s_t^\pi) \geq 0. \tag{50}$$

For the RR problem, one might propose that the fairest comparison would be to formulate the surrogate problem in the same fashion, with achievement of both costs as a constraint, such that

$$\max_\pi \mathbb{E}_\pi \left[ \sum_t^\infty \gamma^t \min \left\{ \max_{t' \leq t} r_1(s_{t'}^\pi), \max_{t' \leq t} r_2(s_{t'}^\pi) \right\} \right] \quad \text{s.t.} \quad \min \left\{ \max_t r_1(s_t^\pi), \max_t r_2(s_t^\pi) \right\} \geq 0, \tag{51}$$

which we define as variant 1 (CPPO-v1). Empirically, however, we found this formulation to be the poorest by far, perhaps due to the abundance of the non-smooth combinations. We thus also compare with more naive formulations which relax the outer minimizations to summation in the reward

$$\max_\pi \mathbb{E}_\pi \left[ \sum_t^\infty \gamma^t \max_{t' \leq t} r_1(s_{t'}^\pi) + \max_{t' \leq t} r_2(s_{t'}^\pi) \right] \quad \text{s.t.} \quad \min \left\{ \max_t r_1(s_t^\pi), \max_t r_2(s_t^\pi) \right\} \geq 0, \tag{52}$$

which we define as variant 2 (CPPOv2), and additionally, in the constraint

$$\max_\pi \mathbb{E}_\pi \left[ \sum_t^\infty \gamma^t \max_{t' \leq t} r_1(s_{t'}^\pi) + \max_{t' \leq t} r_2(s_{t'}^\pi) \right] \quad \text{s.t.} \quad \max_t r_1(s_t^\pi) + \max_t r_2(s_t^\pi) \geq 0, \tag{53}$$

which we define as variant 3 (CPPOv3). This last approach, although naive and seemingly unfair, vastly outperforms the other variants in the RR problem.

### G.2    STL BASELINES

In contrast with constrained optimization, one might also incorporate the STL methods, which in the current context simply decompose and optimize the independent objectives. For the RAA problem,

the standard RA solution serves as a trivial STL baseline since we may attempt to continuously attempt to reach the solution while avoiding the obstacle. In the RR case, we define a decomposed STL baseline (DSTL) which naively solves both R problems, and selects the one with lower value to achieve first.

## H   DETAILS OF RAA & RR EXPERIMENTS: HOPPER

Table 2: Hyperparameters for Hopper Learning

| Hyperparameters for DOHJ-PPO | Values |
| --- | --- |
| Network Architecture | MLP |
| Units per Hidden Layer | 256 |
| Numbers of Hidden Layers | 2 |
| Hidden Layer Activation Function | tanh |
| Entropy coefficient | Linear Decay 1e-2 $\rightarrow$ 0 |
| Optimizer | Adam |
| Discount factor $\gamma$ | Linear Anneal 0.995 $\rightarrow$ 0.999 |
| GAE lambda parameter | 0.95 |
| Clip Ratio | 0.2 |
| Actor Learning rate | Linear Decay 3e-4 $\rightarrow$ 0 |
| Reward/Cost Critic Learning rate | Linear Decay 3e-4 $\rightarrow$ 0 |
| Number of Environments | 128 |
| Number of Steps | 400 |
| Total Timesteps (RAA) | 50M |
| Total Timesteps (RR) | 50M |
| Scan Steps | 4 |
| Update Epochs | 10 |
| Number of Minibatches | 32 |
| **Add'l Hyperparameters for CPPO** | |
| $K_P$ | 1 |
| $K_I$ | 1e-4 |
| $K_D$ | 1 |

The Hopper environment is taken from Gym Brockman et al. (2016) and So et al. (2024). In both RAA and RR problems, we define rewards and penalties based on the position of the Hopper head, which we denote as $(x, y)$ in this section.

In the RAA task, the reward is defined as

$$r(x, y) = \sqrt{|x - 2| + |y - 1.4|} - 0.1 \tag{54}$$

to incentive the Hopper to reach its head to the position at $(x, y) = (2, 1.4)$. The penalty $q$ is defined as the minimum of signed distance functions to a ceiling obstacle at $(1, 0)$, wall obstacles at $x > 2$ and $x < 0$ and a floor obstacle at $y < 0.5$. In order to safely arrive at high reward (and always avoid the obstacles), the Hopper thus must pass under the ceiling and not dive or fall over in the achievement of the target, as is the natural behavior.

In the RR task, the first reward is defined again as

$$r_1(x, y) = \sqrt{|x - 2| + |y - 1.4|} - 0.1 \tag{55}$$

to incentive the Hopper to reach its head to the position at $(x, y) = (2, 1.4)$, and the second reward as

$$r_2(x, y) = \sqrt{|x - 0| + |y - 1.4|} - 0.1 \tag{56}$$

to incentive the Hopper to reach its head to the position at $(x, y) = (0, 1.4)$. In order to achieve both rewards, the Hopper must thus hop both forwards and backwards without crashing or diving.

In all experiments, the Hopper is initialized in the default standing posture at a random $x \in [0, 2]$ so as to learn a position-agnostic policy. The DOHJ-PPO parameters used to train these problems can be found in Table 2.

## I   DETAILS OF RAA & RR EXPERIMENTS: F16

The F16 environment is taken from So et al. (2024), including a F16 fighter jet with a 26 dimensional observation. The jet is limited to a flight corridor with up to 2000 relative position north ($x_{PN}$), 1200 relative altitude ($x_H$), and $\pm 500$ relative position east ($x_{PE}$).

In the RAA task, the reward is defined as

$$r(x, y) = \frac{1}{5}|x_{PN} - 1500| - 50 \tag{57}$$

to incentivize the F16 to fly through the geofence defined by the vertical slice at 1500 relative position north. The penalty $q$ is defined as the minimum of signed distance functions to geofence (wall) obstacles at $x_{PN} > 2000$ and $|x_{PE}| > 500$ and a floor obstacle at $x_H < 0$. In order to safely arrive at high reward (and always avoid the obstacles), the F16 thus must fly through the target geofence and then evade crashing into the wall directly in front of it.

In the RR task, the rewards are defined as

$$r_1(x_{PN}, x_H) = \frac{1}{5}\sqrt{|x_{PN} - 1250| + |y - 850|} - 30 \tag{58}$$

and

$$r_2(x_{PN}, x_H) = \frac{1}{5}\sqrt{|x_{PN} - 1250| + |y - 350|} - 30 \tag{59}$$

to incentive the F16 to reach both low and high-altitude horizontal cylinders. In order to achieve both rewards, the F16 must thus aggressively pitch, roll and yaw between the two targets.

In all experiments, the F16 is initialized with position $x_{PN} \in [250, 750], x_H \in [300, 900], x_{PE} \in [-250, 250]$ and velocity in $v \in [200, 450]$. Additionally, the roll, pitch, and yaw are initialized with $\pm\pi/16$ to simulate a variety of approaches to the flight corridor. Further details can be found in *So et al.* (2024). The DOHJ-PPO parameters used to train these problems can be found in Table 3.

Table 3: Hyperparameters for F16 Learning

| **Hyperparameters for DOHJ-PPO** | **Values** |
|---|---|
| Network Architecture | MLP |
| Units per Hidden Layer | 256 |
| Numbers of Hidden Layers | 2 |
| Hidden Layer Activation Function | tanh |
| Entropy coefficient | Linear Decay 1e-2 $\rightarrow$ 0 |
| Optimizer | Adam |
| Discount factor $\gamma$ | Linear Anneal 0.995 $\rightarrow$ 0.999 |
| GAE lambda parameter | 0.95 |
| Clip Ratio | 0.2 |
| Actor Learning rate | Linear Decay 1e-3 $\rightarrow$ 0 |
| Reward/Cost Critic Learning rate | Linear Decay 1e-3 $\rightarrow$ 0 |
| Number of Environments | 256 |
| Number of Steps | 200 |
| Total Timesteps (RAA) | 50M |
| Total Timesteps (RR) | 100M |
| Scan Steps | 10 |
| Update Epochs | 10 |
| Number of Minibatches | 64 |
| **Add'l Hyperparameters for CPPO** | |
| $K_P$ | 1 |
| $K_I$ | 1e-4 |
| $K_D$ | 1 |

## J    DETAILS OF RAA & RR EXPERIMENTS: SAFETYGYM

The SafetyGym environment is taken from the SafetyGym Point Environment Brockman et al. (2016), reimplemented in So et al. (2024). In both RAA and RR problems, we define rewards and penalties based on the position of the Point position, which we denote as $(x, y)$ in this section.

In the RAA task, the reward is defined as

$$r(x, y) = \sqrt{|x| + |y|} - 0.3 \tag{60}$$

to incentive the Point to reach to the position at $(x, y) = (0., 0.)$. The penalty $q$ is defined as the minimum of signed distance functions to wall obstacles at $|x| \geq 3$ and $y \geq |3|$, and a scatter of eight random obstacles at $\{(1.4, 0.6), (0.4, 1.2), (-1.2, 0.8), (-0.9, 0.1.5), (-0.1, -1.1), (0.7, 0.2), (-0.3, 0.8), (-1.3, -1.3)\}$. In order to safely arrive at high reward (and always avoid the obstacles), the Point thus must drive through the obstacles and without leaving the box or crashing after arriving.

In the RR task, the first reward is defined again as

$$r_1(x, y) = \sqrt{|x - 2.5| + |y - 2.5|} - 0.3 \tag{61}$$

to incentive the Point to drive to the position at $(x, y) = (2.5, 2.5)$, and the second reward as

$$r_2(x, y) = \sqrt{|x + 2.5| + |y + 2.5|} - 0.3 \tag{62}$$

to incentive the Point to drive to the position at $(x, y) = (-2.5, -2.5)$. In order to achieve both rewards, the Point must thus drive to both targets.

In all experiments, the Point is initialized at a random $(x, y) \in [-2, 2]^2$ so as to learn a position-agnostic policy. The DOHJ-PPO parameters used to train these problems can be found in Table 4.

Table 4: Hyperparameters for SafetyGym Learning

| Hyperparameters for DOHJ-PPO | Values |
|---|---|
| Network Architecture | MLP |
| Units per Hidden Layer | 256 |
| Numbers of Hidden Layers | 2 |
| Hidden Layer Activation Function | tanh |
| Entropy coefficient | Linear Decay 5e-3 $\rightarrow$ 0 |
| Optimizer | Adam |
| Discount factor $\gamma$ | Linear Anneal 0.995 $\rightarrow$ 0.9995 |
| GAE lambda parameter | 0.95 |
| Clip Ratio | 0.2 |
| Actor Learning rate | Linear Decay 3e-4 $\rightarrow$ 0 |
| Reward/Cost Critic Learning rate | Linear Decay 3e-4 $\rightarrow$ 0 |
| Number of Environments | 128 |
| Number of Steps | 400 |
| Total Timesteps (RAA) | 50M |
| Total Timesteps (RR) | 100M |
| Scan Steps | 4 |
| Update Epochs | 10 |
| Number of Minibatches | 32 |
| **Add'l Hyperparameters for CPPO** | |
| $K_P$ | 1 |
| $K_I$ | 1e-4 |
| $K_D$ | 1 |

## K    DETAILS OF RAA & RR EXPERIMENTS: HALFCHEETAH

The HalfCheetah environment is taken from Gym Brockman et al. (2016), reimplemented in So et al. (2024). In both RAA and RR problems, we define rewards based on the HalfCheetah head position,

which we denote as $(x, y)$ in this section, and we define the penalties based on the position of all HalfCheetah body positions (back, neck, head, thighs, shins, and feet).

In the RAA task, the reward is defined as

$$r(x, y) = |x - 5.5| - 0.25 \tag{63}$$

to incentive the HalfCheetah to reach its head to the cylinder centered at $x = 5.5$. The penalty $q$ is defined as the minimum of signed distance functions to a floor obstacles at $(0, 4.5)$ and $(0, 6.5)$ which have height of $0.05$ and width $0.5$, and a back wall obstacle at $x < -0.9$. In order to safely arrive at high reward (and always avoid the obstacles), the HalfCheetah thus must jump over the front floor obstacle (at $x = 4.5$) and avoid crashing into, either by hopping over or standing, the back floor obstacle (at $x = 6.5$).

In the RR task, the first reward is defined again as

$$r_1(x, y) = \sqrt{|x - 5| + |y - 1|} - 0.1 \tag{64}$$

to incentive the HalfCheetah to reach its head to the position at $(x, y) = (5, 1)$, and the second reward as

$$r_2(x, y) = \sqrt{|x + 5| + |y - 1|} - 0.1 \tag{65}$$

to incentive the HalfCheetah to reach its head to the position at $(x, y) = (-5, 1)$. In order to achieve both rewards, the HalfCheetah must thus gallop both forwards and backwards.

In all experiments, the HalfCheetah is initialized in the default standing posture at a random $x \in [0.5, 1.5]$ so as to learn a position-agnostic policy. The DOHJ-PPO parameters used to train these problems can be found in Table 5.

Table 5: Hyperparameters for HalfCheetah Learning

| Hyperparameters for DOHJ-PPO | Values |
|---|---|
| Network Architecture | MLP |
| Units per Hidden Layer | 256 |
| Numbers of Hidden Layers | 2 |
| Hidden Layer Activation Function | tanh |
| Entropy coefficient | Linear Decay 5e-3 $\rightarrow$ 0 |
| Optimizer | Adam |
| Discount factor $\gamma$ | Linear Anneal 0.995 $\rightarrow$ 0.9995 |
| GAE lambda parameter | 0.95 |
| Clip Ratio | 0.2 |
| Actor Learning rate | Linear Decay 3e-4 $\rightarrow$ 0 |
| Reward/Cost Critic Learning rate | Linear Decay 3e-4 $\rightarrow$ 0 |
| Number of Environments | 128 |
| Number of Steps | 400 |
| Total Timesteps (RAA) | 100M |
| Total Timesteps (RR) | 150M |
| Scan Steps | 4 |
| Update Epochs | 10 |
| Number of Minibatches | 32 |
| **Add'l Hyperparameters for CPPO** | |
| $K_P$ | 100 |
| $K_I$ | 1e-4 |
| $K_D$ | 1 |

## L  CONTRAST WITH DECOMPOSITIONS IN TEMPORAL LOGIC

Temporal Logic (TL) predicates, including those for reach (eventually $\varphi$), avoid (always not $\psi$), and reach-avoid (not $\psi$ until $\varphi$) problems, enjoy useful algebraic properties. These properties provide a convenient way to decompose complex TL predicates into simpler predicates. Unfortunately, the

decomposition of these predicates does not generally translate to valid decompositions of the optimal value functions in HJR.

More explicitly, it is indeed possible to phrase HJR problem formulations, such as our RR and RAA problems, using the language of quantitative semantics from the TL literature. In particular, the standard HJR problem is to find the sequence of actions that maximizes some objective function, and this objective function can generally be written as a quantitative semantic corresponding to the specification of the desired system behavior (however, this is not explicitly written in TL notation in the HJR literature). That said, the optimal value function decomposition results for the RR and RAA problems are distinct from the decomposition of the corresponding quantitative semantics. In other words, our decomposition results are statements about the optimal control associated with the quantitative semantic, not the quantitative semantic itself.

This subtle distinction can be clarified by the following counter-examples, where a valid decomposition of the quantitative semantic (on the TL side) for the RR and RAA problem does not translate to a valid decomposition of the optimal value functions (on the HJR side) for the RAA problem. We will use $F$ as the eventually operator, $G$ as the always operator, and $\land / \lor / \neg$ as logical and/or/not.

## L.1 RAA CASE

Consider an RAA problem where my friend is holding a piñata which I would like to break with a bat. We can always decompose the RAA quantitative semantic into R and A quantitative semantics. However, suppose there is some state of the system from which I can either eventually hit the piñata or always avoid hitting my friend, but not both. In this case, the optimal value function for the R and A problems will both be non-negative, even though I cannot actually achieve the RAA task from this state.

To make this argument explicit, define the atomic predicates $\varphi$ and $\psi$ to represent hitting the piñata and hitting my friend, respectively:

$$(\mathbf{x}, t) \models \varphi \iff r(\mathbf{x}(t)) \geq 0,$$
$$(\mathbf{x}, t) \models \psi \iff p(\mathbf{x}(t)) \geq 0.$$

Given a predicate $\mu$, let $\rho_\mu$ be the corresponding quantitative semantic. Thus, $\rho_\varphi[\mathbf{x}, t] = r(\mathbf{x}(t))$ and $\rho_\psi[\mathbf{x}, t] = p(\mathbf{x}(t))$. The quantitative semantics for the R, A, and RAA problems are then, respectively:

$$\rho_{\mathrm{R}}[\mathbf{x}, t] := \rho_{F\varphi}[\mathbf{x}, t] = \max_{\tau \geq t} r(\mathbf{x}(\tau)), \tag{66}$$

$$\rho_{\mathrm{A}}[\mathbf{x}, t] := \rho_{G\neg\psi}[\mathbf{x}, t] = \min_{\tau \geq t} -p(\mathbf{x}(\tau)), \tag{67}$$

$$\rho_{\mathrm{RAA}}[\mathbf{x}, t] := \rho_{(F\varphi)\land(G\neg\psi)}[\mathbf{x}, t] = \min\left\{ \max_{\tau \geq t} r(\mathbf{x}(\tau)), \min_{\tau \geq t} -p(\mathbf{x}(\tau)) \right\}. \tag{68}$$

As mentioned earlier, the optimal value functions for the R, A, and RAA problems can then be written in terms of the quantitative semantics, i.e.

$$V_{\mathrm{R}}^*(s) = \max_\pi \rho_{\mathrm{R}}[\mathbf{x}_s^\pi, 0],$$
$$V_{\mathrm{A}}^*(s) = \max_\pi \rho_{\mathrm{A}}[\mathbf{x}_s^\pi, 0],$$
$$V_{\mathrm{RAA}}^*(s) = \max_\pi \rho_{\mathrm{RAA}}[\mathbf{x}_s^\pi, 0],$$

where $\mathbf{x}_s^\pi$ is the trajectory of the bat corresponding to the initial state $s$ and policy $\pi$.

But here is the key point: although it is always true that

$$\rho_{\mathrm{RAA}}[\mathbf{x}, t] = \min\{\rho_{\mathrm{R}}[\mathbf{x}, t], \rho_{\mathrm{A}}[\mathbf{x}, t]\},$$

in general we only have that

$$V_{\text{RAA}}^*(s) \leq \min\{V_{\text{R}}^*(s), V_{\text{A}}^*(s)\}.$$

Indeed this inequality is sometimes strict. Again, consider the case where there is some state of the system $s$ from which I can either eventually hit the piñata or always avoid hitting my friend, but cannot do both. Then

$$V_{\text{RAA}}^*(s) < 0 \leq \min\{V_{\text{R}}^*(s), V_{\text{A}}^*(s)\}.$$

In other words, the algebra that applies to the TL predicates translates to an algebra on the TL quantitative semantics, but it does not generally translate to an algebra on the optimal value functions. This is why our decomposition required thorough justification.

### L.2 RR CASE

Next, suppose that we are solving an RR problem for a small robotic boat that must deliver supplies to two downstream islands in a wide river that flows from north to south. The islands are at the same latitude, but one is to the west and one is to the east. Suppose the boat starts far enough upstream that it can reach either island, but the current is too strong to move from one island to the other. Then the boat can satisfy the R task for the west island (by ignoring the east island) and it can satisfy the R task for the east island (by ignoring the west island), but it cannot satisfy the RR task. The decomposition of the quantitative semantics for the RR problem into the minimum of the quantitative semantics for the two R problems will then not translate to a valid decomposition of the value functions.

More explicitly, with $r_1$ and $r_2$ the signed distance functions to either island, let

$$(\mathbf{x}, t) \models \varphi \iff r_1(\mathbf{x}(t)) \geq 0, \tag{69}$$
$$(\mathbf{x}, t) \models \psi \iff r_2(\mathbf{x}(t)) \geq 0. \tag{70}$$

Thus, $\rho_\varphi[\mathbf{x}, t] = r_1(\mathbf{x}(t))$ and $\rho_\psi[\mathbf{x}, t] = r_2(\mathbf{x}(t))$. The quantitative semantics for reaching the west island, reaching the east island, and reaching both islands are then respectively,

$$\rho_{\text{R1}}[\mathbf{x}, t] := \rho_{F\varphi}[\mathbf{x}, t] = \max_{\tau \geq t} r_1(\mathbf{x}(\tau)), \tag{71}$$

$$\rho_{\text{R2}}[\mathbf{x}, t] := \rho_{F\psi}[\mathbf{x}, t] = \max_{\tau \geq t} r_2(\mathbf{x}(\tau)), \tag{72}$$

$$\rho_{\text{RR}}[\mathbf{x}, t] := \rho_{(F\varphi) \wedge (F\psi)}[\mathbf{x}, t] = \min\left\{\max_{\tau \geq t} r_1(\mathbf{x}(\tau)), \max_{\tau \geq t} r_2(\mathbf{x}(\tau))\right\}. \tag{73}$$

As before, the optimal value functions for the two R problems and the RR problems can then be written in terms of the quantitative semantics, i.e.

$$V_{\text{R1}}^*(s) = \max_\pi \rho_{\text{R1}}[\mathbf{x}_s^\pi, 0], \tag{74}$$

$$V_{\text{R2}}^*(s) = \max_\pi \rho_{\text{R2}}[\mathbf{x}_s^\pi, 0], \tag{75}$$

$$V_{\text{RR}}^*(s) = \max_\pi \rho_{\text{RR}}[\mathbf{x}_s^\pi, 0]. \tag{76}$$

But we still have the analogous issue: although it is always true that

$$\rho_{\text{RR}}[\mathbf{x}, t] = \min\{\rho_{\text{R1}}[\mathbf{x}, t], \rho_{\text{R2}}[\mathbf{x}, t]\},$$

in general we only have that

$$V_{\text{RR}}^*(s) \leq \min\{V_{\text{R1}}^*(s), V_{\text{R2}}^*(s)\}.$$

When we begin from some state of the system $s$ from which I can eventually reach the east island or I can eventually reach the west island, but not both, then the inequality is strict:

$$V_{\mathrm{RR}}^*(s) < 0 \leq \min\{V_{\mathrm{R1}}(s), V_{\mathrm{R2}}^*(s)\}.$$

## M BROADER IMPACTS

This paper touches on advancing fundamental methods for Reinforcement Learning. In particular, this work falls into the class of methods designed for Safe Reinforcement Learning. Methods in this class are primarily intended to prevent undesirable behaviors in virtual or cyber-physical systems, such as preventing crashes involving self-driving vehicles or potentially even unacceptable speech among chatbots. It is an unfortunate truth that safe learning methods can be repurposed for unintended use cases, such as to prevent a malicious agent from being captured, but the authors do not foresee the balance of potential beneficial and malicious applications of this method to be any greater than other typical methods in Safe Reinforcement Learning.

## N ACKNOWLEDGMENTS

We would like to thank several people for assisting the development of the work. We thank Dr. Jaime Fisac and Dr. Kai-Chieh Hsu for lending their knowledge on bridging Hamilton-Jacobi reachability and reinforcement learning. We thank Dr. Chuchu Fan and Oswin So for offering their knowledge on PPO-based HJ-RL and algorithm design and providing us with starter code. Lastly, we thank Dr. Morteza Lahijanian, Dr. Sean Gao and Dr. Nikolay Atanasov for useful discussions regarding the paper.

### FUNDING

Research reported in this publication was work is supported by the Office of Naval Research under grant N00014-24-1-2661, the Naval Innovation, Science, and Engineering Center under grant N00014-23-1-2831, and the Society of Hellman Fellows. Research reported in this publication was also supported by NIBIB of the National Institutes of Health under award number T32EB009380. The content is solely the responsibility of the authors and does not necessarily represent the official views of the National Institutes of Health.

