# Supplementary Material

# Contents

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

}^*_{\mathrm{R}}(\bar{x}_T) \leq \tilde{v}^*_{\mathrm{R}}(s)$. Thus $r_2(x^\circ_{T'}) \leq v^*_{\mathrm{R2}}(x^\circ_{T'}) \leq v^*_{\mathrm{R2}}(x^\circ_T) = v^*_{\mathrm{R2}}(\bar{x}_T) < \tilde{v}^*_{\mathrm{R}}(s)$. It follows that

$$\max\left\{\min\left\{r_1\left(x^\circ_{T'}\right), v^*_{\mathrm{R2}}\left(x^\circ_{T'}\right)\right\}, \min\left\{r_2\left(x^\circ_{T'}\right), v^*_{\mathrm{R1}}\left(x^\circ_{T'}\right)\right\}\right\} < \tilde{v}^*_{\mathrm{R}}(s),$$

contradicting our choice of $T'$.

Thus $r_2(\bar{x}_t) \geq \tilde{v}^*_{\mathrm{R}}(\bar{x}_T) = \tilde{v}^*_{\mathrm{R}}(s)$ for some $t \in \mathbb{N}$ and also $r_1(\bar{x}_T) \geq \tilde{v}^*_{\mathrm{R}}(\bar{x}_T) = \tilde{v}^*_{\mathrm{R}}(s)$, so that (36) must hold by Lemma 9.

$\square$

**Corollary 8.** *For all $s \in \mathcal{S}$, we have $V^*_{\mathrm{RR}}(s, r_1(s), r_2(s)) = v^*_{\mathrm{RR}}(s)$.*

*Proof of Theorem 2.* The proof of this theorem immediately follows from the previous corollary together with Corollary 7 and Lemma 9.

$\square$

## C  PROOF OF OPTIMALITY THEOREM

*Proof of Theorem 3.* The inequalities in both lines of the theorem follow from the fact that for each $\pi \in \Pi$, we can define a corresponding augmented policy $\bar{\pi} \in \overline{\overline{\Pi}}$ by

$$\bar{\pi}(s, y, z) = \pi(s) \quad \forall s \in \mathcal{S}, y \in \mathcal{Y}, z \in \mathcal{Z},$$

in which case $V^\pi_{\mathrm{RAA}}(s) = V^{\bar{\pi}}_{\mathrm{RAA}}(s)$ and $V^\pi_{\mathrm{RR}}(s) = V^{\bar{\pi}}_{\mathrm{RR}}(s)$ for each $s \in \mathcal{S}$. Note that in general, we cannot define a corresponding policy for each augmented policy, so the reverse inequality does not generally hold (see Figure 3 for intuition regarding this fact).

The equalities in both lines of the theorem are simply restatements of Lemma 5 and Lemma 9. $\square$

## D  THE SRABE AND ITS POLICY GRADIENT

*Proof of Proposition 1.* We here closely follow the proof of Theorem 3 in (4), which itself modifies the proofs of the Policy Gradient Theorems in Chapter 13.2 and 13.6 (60). We only make the minimal modifications required to adapt the PPO algorithm developed previously for the SRBE to on for the SRABE.

$$\begin{aligned}
\nabla_\theta \tilde{V}^{\pi_\theta}_{\mathrm{RAA}}(s) =& \nabla_\theta \left( \sum_{a \in \mathcal{A}} \pi_\theta(a|s) \tilde{Q}^{\pi_\theta}_{\mathrm{RAA}}(s, a) \right) \\
=& \sum_{a \in \mathcal{A}} \Big( \nabla_\theta \pi_\theta(a|s) \tilde{Q}^{\pi_\theta}_{\mathrm{RAA}}(s, a) \\
& \qquad + \pi_\theta(a|s) \nabla_\theta \min\left\{ \max\left\{ \tilde{V}^\pi_{\mathrm{RAA}}\left(f(s,a)\right), r_{\mathrm{RAA}}(s) \right\}, q(s) \right\} \Big) \\
=& \sum_{a \in \mathcal{A}} \Big( \nabla_\theta \pi_\theta(a|s) \tilde{Q}^{\pi_\theta}_{\mathrm{RAA}}(s, a) \\
& \qquad + \pi_\theta(a|s) \left[ q(s) < \tilde{V}_{\mathrm{RAA}}\left(f(s,a)\right) < r_{\mathrm{RAA}}(s) \right] \nabla_\theta \tilde{V}^\pi_{\mathrm{RAA}}\left(f(s,a)\right) \Big) \

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

## K    ACKNOWLEDGMENTS

This section has been redacted for the purpose of anonymous review.