# OpenReview forum: "Dual-Objective Reinforcement Learning with Novel Hamilton-Jacobi-Bellman Formulations"
_ICLR.cc/2026/Conference — ICLR 2026 Poster_

### Official Review · Reviewer_jprC · 2025-10-30

**Soundness:** 4
**Presentation:** 4
**Contribution:** 3
**Rating:** 6
**Confidence:** 4

**Summary:**

This paper proposes a novel RL framework that integrates Hamilton-Jacobi equations from optimal control theory to handle dual-objective tasks involving safety and goal achievement.
The authors introduce two new value formulations: Reach-Always-Avoid, which requires reaching a goal while maintaining safety thereafter, and Reach-Reach, which requires achieving two distinct goals sequentially.
These value functions are defined via min-max extremal operators, enabling direct reasoning about best- and worst-case outcomes.
The authors show that the Reach-Always-Avoid and Reach-Reach problems can be decomposed into simpler subproblems (Reach, Avoid, and Reach-Avoid), each corresponding to known HJ-Bellman equations.
To make these formulations tractable, they augment the MDP state with historical extrema to restore the Markov property and propose an RL algorithm DO-HJ-PPO, a variant of PPO that jointly learns decomposed and composed actor-critic pairs.
The algorithm employs a Reach-Avoid GAE to approximate the extremal Bellman differences.
Experiments on discrete and continuous control tasks demonstrate that DO-HJ-PPO outperforms baselines.

**Strengths:**

1. The paper builds on HJ optimal control theory and connects it elegantly to RL. The proposed formulations (Reach-Always-Avoid and Reach-Reach) are mathematically rigorous and clearly grounded in control-theoretic Bellman principles.

2. The theoretical decomposition of Reach-Always-Avoid and Reach-Reach problems into subproblems is clean and provable, allowing modular reuse of existing HJ-RL methods.

3. By directly modeling safety and goal satisfaction in the value function, the approach avoids the delicate tuning required by Lagrangian or constrained RL methods.

4. Experiments span both discrete and continuous control tasks. The method consistently outperforms strong baselines, showing robustness and good generalization.

**Weaknesses:**

1. While the theoretical model is elegant, many experiments (e.g., Hopper-RAA, Hopper-RR) use geometrically simple rewards that do not fully reflect the difficulty of real-world control problems. Thus, the results may overstate performance in realistic scenarios.

2. The algorithm involves multiple coupled actor-critic networks (for decomposed and composed objectives), which can significantly increase computational and memory costs. The paper lacks ablation studies or discussion on scalability or efficiency.

3. The baselines (e.g., CPPO, RESPO) are standard but may not be fully optimized under the same hyperparameter or reward conditions, making the fairness of some comparisons uncertain.

**Questions:**

1. The paper extends the HJ-Bellman framework to stochastic and learned environments, but most derivations assume deterministic dynamics. How robust is the theoretical foundation when applied to stochastic MDPs?

2. The RAA and RR formulations are proven decomposable, but the algorithm jointly learns all sub-critics with bootstrapped targets.
Is there any theoretical guarantee that this joint training converges to the same fixed point as the original HJ decomposition?

3. The continuous-control tasks use simplified geometric rewards.
Would DO-HJ-PPO still outperform baselines if trained with the dense, physically meaningful rewards used in standard MuJoCo benchmarks?

---

> ### Author Response · Authors · 2025-11-20
>
> We thank the reviewer for their analysis of our submission and their feedback for the betterment of the work. Below, we address the weaknesses posed by the reviewer.
>
> ***While the theoretical model is elegant, many experiments (e.g., Hopper-RAA, Hopper-RR) use geometrically simple rewards that do not fully reflect the difficulty of real-world control problems. Thus, the results may overstate performance in realistic scenarios.***
>
> We appreciate the reviewer's consideration and we agree that safety can be highly nuanced in special applications and real-world settings. In this work, we defined the reward and constraint to be (dense) functions of distance to regions of interest and hazard. We made this choice for two major reasons.
>
> First, these functions arise naturally in safe robotics applications in the wild. For example, consider an autonomous vehicle attempting to cross a busy intersection where it must get to a lane on the far side without colliding with pedestrians entering the street. Here, one might choose a distance reward for the target lane and a penalty distance to any observed pedestrians. Moreover, in real-world applications, these functions are frequently used because of their immediate availability from sensor data, for example, from a lidar sensor on top of the autonomous car.
>
> Second, the tasks arising from the distance functions, namely those corresponding to reachability and avoidability, have a history in safe RL and robotics research [Ganai et al. 2024, So et al. 2024, Fisac et al. 2019, Mitchell et al. 2005, Altman et. al 2021], making them the standard benchmark for comparison with previous algorithms. In fact, these functions define the standard tasks in OpenAI’s safety-gym suite [Ray et al. 2019], which use simulated lidar to define distance to targets (eg. buttons) and obstacles (eg. vases).
>
> From this perspective, the reward and penalty distance functions represent a broad class of real-world problems and a clear standard of comparison with the history of algorithms developed for safety in RL and robotics.
>
> ***The algorithm involves multiple coupled actor-critic networks (for decomposed and composed objectives), which can significantly increase computational and memory costs. The paper lacks ablation studies or discussion on scalability or efficiency.***
>
> The reviewer is correct in noting that our proposed algorithm involves additional actors and critics to represent the decompositions arising in the novel HJB theory, and indeed this increases the computational load per gradient step. An important facet of these decompositions is that they are independent during the policy roll-out, and hence, each iteration of the algorithm (eg. policy evaluation, advantage computation, model update) can be parallelized completely, reducing the ideal time to solve the increased compute load. As the reviewer keenly notes, however, this does not guarantee that this approach will not increase complexity of the learning problem, possibly delaying convergence. To interrogate this phenomenon, particularly in stochastic settings, we have added an ablation study to our paper, showcasing the per-epoch task success percent for an increasing quantity of dynamic noise to the paper. From this ablation, one can observe that our algorithm always achieves its peak performance earliest or second earliest in comparison with the top three performing baselines. Moreover, this peak performance is not only fast, but also the maximum of peak performances across all algorithms in 6 of 8 ablation conditions with increasing stochasticity.

---

> ### Author Response · Authors · 2025-11-20
>
> ***The baselines are standard but may not be fully optimized ..., making the fairness of some comparisons uncertain.***
>
> Indeed, the augmented Lagrangian methods, such as CPPO, PPOLAG and others, need a significant amount of tuning to achieve good performance, leading to the so-called reward-engineering that is predominant in real-world RL usage. Accordingly, the authors spent a considerable amount of time tuning these algorithms to achieve the observed success rates (in fact, this was the most laborious task in this paper by far). For example, for CPPO we scanned $K_p$ coefficients from 1e-2 to 1e3 and soft-constraint thresholds of -0.5 to 1, and we experimented with constraint conditions that included both minimums, sums and sum-minimums of the penalty function $q$ and the reward function $r$. See the Baselines section of the Appendix for further discussion and the top-performing form.
>
> In general, it is precisely because the augmented Lagrangian approaches blend the reward and constraint objectives that one must iteratively tune for a desirable balance. Without structured prioritization, the augmented Lagrangian landscape corresponds to an a-physical combination of the independent objectives, and maximization of this “net” goal can be achieved by unintuitive combinations of the safety and liveness tasks.
>
> In contrast, the RR and RAA value functions are defined by temporal maxima and minima that encode hard rules on which objectives dominate the overall value. Notably, this yields an algorithm (DOHJ-PPO) that requires little to no tuning to balance the task objectives. In the RAA setting, for example, one may tune the reward function to change the emergent reaching behavior, however, in states where the robot will crash regardless of its action, the overall value must still be dominated by the same penalty term (as it will define the minimum objective). In other words, changing $r$ does not change the optimization landscape where the agent will crash into an obstacle, and hence, learns with respect to worst-case collisions in either case.
>
> In comparison with the HJ-RL baselines, RESPO and RCPPO are indeed imperfect for the problems at hand of considering multiple objectives. That is, however, because their value functions (and the Safety Bellman equations they yield) are defined by single reaching or avoiding objectives. Prior to this work, the dual-objective problem was unconsidered and perhaps unknown in HJ literature. In fact, it was the deficiency of these algorithms and other existing HJ methods that inspired the authors to study the dual-objective scenarios and propose novel Safety-Bellman equations for these problems.
>
> ***The paper extends the HJ-Bellman ... but most derivations assume deterministic dynamics. How robust is the theoretical foundation when applied to stochastic MDPs?***
>
> We appreciate the reviewer’s question and agree this raises an interesting point. While the HJB equations are derived in deterministic settings, ultimately they are relaxed to the stochastic approximations (the SRBE and SRABE), which perform well with stochastic policies but may behave differently with stochastic dynamics. To interrogate this question, **we have added to the paper an ablation experiment** in which a scaled Gaussian disturbance is added to the dynamics of the HalfCheetah environment for both RAA and RR problems. Specifically, the Gaussian term is added to the velocities and angular velocities (avoiding position directly to prevent teleportation) of the HalfCheetah body at each time-point. With an increasing magnitude of noise, we compare the task achievement (success percentage) of DOHJ-PPO with the top three performing baselines in either task. At the time of this reviewer response, we have the data from two seeds and will continue to update the plot as the third seeds finish. **These results may be found in Fig. 5.**
>
> In this ablation, we find that the proposed approach, using the novel Bellman equations with stochastic relaxations, offers a significant performance improvement even amongst sizable randomness (< 2 units/step). In the HalfCheetah RAA problem, DOHJ-PPO dominates the top performing baselines with a 8%-22% peak-performance gap between it and the second best algorithm, and is the fastest to peak-performance, for noise levels up to 1 unit per time step, beyond which all algorithms perform equally poor. In the RR case, we find an even starker result, with all but one experiment demonstrating a >30% improvement in peak-performance even in the high-noise regimes, with an exception at 1 unit per time step where DOHJ-PPO performs 5% worse than the best baselines, DSTL. In all cases, DOHJ-PPO is the second fastest algorithm, behind DSTL, but often achieves a significantly higher peak performance. These results are interesting, demonstrating that the stochastic relaxations offer a significantly competitive advantage over existing methods and also perhaps might be improved in future work.

---

> ### Author Response · Authors · 2025-11-20
>
> ***The RAA and RR formulations are proven decomposable, but the algorithm jointly learns all sub-critics with bootstrapped targets. Is there any theoretical guarantee that this joint training converges to the same fixed point as the original HJ decomposition?***
>
> Indeed, our approach to learning RR and RAA the value functions is founded on PPO [Schulman et al. 2017] and therefore lacks a proof of convergence, even without the coupled gradient steps. We do note that as an alternative to the bootstrapping approach, one could run PPO separately on the inner actor-critic until convergence and subsequently the outer one. In this case, sample complexity, and thus performance, would likely be worse, but the algorithm would inherit the same convergence properties as PPO.
>
> For the curiosity of the reviewer towards future directions we have been considering, one might extend the novel Hamilton-Jacobi-Bellman equations to a similarly bootstrapped Soft-Actor Critic algorithm, akin to the HJ-RL algorithm in [Oh et al. 2025], where convergence is simpler to certify. Namely, because the discounted HJ-RL update also induces a contraction then under the same (strong) assumptions where SAC is guaranteed to converge, perhaps one might show that the decompositions are also guaranteed to converge. Finally, one might show that at some point, the dynamic nature of the bootstrapped decompositions will be sufficiently small as to induce a contraction on the composed update.
>
> ***The continuous-control tasks use simplified geometric rewards. Would DOHJ-PPO still outperform baselines if trained with the dense, physically meaningful rewards used in standard MuJoCo benchmarks?***
>
> Please see our response above discussing the definition of these rewards and their relevance to real-world applications.

---

### Official Review · Reviewer_BuMr · 2025-11-01

**Soundness:** 3
**Presentation:** 3
**Contribution:** 2
**Rating:** 6
**Confidence:** 2

**Summary:**

The paper extends Hamilton–Jacobi-style RL beyond the classic Reach (R), Avoid (A), and Reach–Avoid (RA) objectives to two harder, compositional objectives: Reach–Always–Avoid (RAA) (reach a goal and keep avoiding hazards forever afterward) and Reach–Reach (RR) (reach two distinct goals, in any order). The key theoretical result is that both RAA and RR can be decomposed into already-solved HJ-RL subproblems (avoid + reach–avoid for RAA; three reach problems for RR), yielding explicit Bellman forms (Thm. 1–2). Because optimal actions depend on history (e.g. “have I already reached goal 1?”), the paper introduces a specific state augmentation that tracks max-reward and min-penalty over time and shows this augmentation is sufficient (Thm. 3). On top of this, they build DO-HJ-PPO, an on-policy method that learns the decomposed critics alongside the composed RAA/RR critic and couples rollouts so subproblems stay relevant. Experiments on gridworld, Hopper, F-16, SafetyGym, and HalfCheetah show higher “both-objectives achieved” rates and faster completion than Lagrangian and prior HJ-RL baselines.

**Strengths:**

1. The decompositions are stated cleanly, proved in the appendix, and tied directly to implementable Bellman operators; the need for history and the exact augmentation are justified, not hand-waved.
2. Many safe / task-spec RL problems really are “reach X and stay safe” or “reach X then Y”; having explicit Bellman forms and a working PPO variant for those is useful, especially since baselines mostly get only “partial success.”

**Weaknesses:**

1. Assumptions are narrow: Main theory is for deterministic, finite MDPs, yet real tasks are stochastic/continuous, but the paper only gives a heuristic stochastic variant (SRABE) without matching guarantees. A short discussion of what breaks in stochastic dynamics is needed.
2. State augmentation cost: The proposed augmentation grows the state with running max/min signals; for higher-dim tasks (real robots, multi-goal specs) this could be heavy, and the paper doesn’t study scalability.
3. The paper lacks sample-complexity or contraction analysis for the composed operators. Prior HJ-RL work leans on discounted versions to get contraction, and here the composed PPO version is justified mostly by analogy.

**Questions:**

Please see weaknesses.

---

> ### Author Response · Authors · 2025-11-20
>
> We would first like to thank the reviewer for their clear helpful feedback, and the time they have put into their review.
>
> ***Assumptions are narrow: Main theory is for deterministic, finite MDPs, yet real tasks are stochastic/continuous, but the paper only gives a heuristic stochastic variant (SRABE) without matching guarantees. A short discussion of what breaks in stochastic dynamics is needed.***
>
> We appreciate the reviewer’s feedback on the limitation with regard to stochastic dynamics. It is true that we only develop the theory for deterministic, finite MDPs, and the mathematical proofs do not cleanly carry over in the stochastic case. At a high level, the challenge is that maximum and minimum operations do not generally commute with expectations over probability distributions, complicating any algebraic manipulation for decomposition. As suggested, we will add a short discussion regarding these limitations in the conclusion of the paper.
>
> To address the concern with stochasticity at a practical level, **we have added in ablation experiments** where increasing amounts of noise are injected into the Half Cheetah dynamics. Specifically, we sample a Gaussian vector and add it to the dynamics of the Half Cheetah velocities and angular velocities at each time point (avoiding positions to prevent teleportation), and we assess the proposed algorithm against the top performing baselines as the standard deviation scales along [0.0, 0.5, 1.0, 2.0]. In both RAA and RR problems, we observe that for low and moderate levels of noise (< 2), the performance (measured by task success percent) is similar to the deterministic case, and we still outperform baselines significantly. When the levels of noise are quite large, in the RAA case, we observe all algorithms to achieve equally poor performance, and in the RR case, we maintain a >20% improvement over the next best baseline. **The full results can be seen Figure 5.**
>
> With regard to the variant for stochastic *policies*, rather than stochastic dynamics, the SRABE is more than just a heuristic. If the reviewer is interested, we have outlined the theoretical justification, which will be added to the appendix in the final draft:
>
> First, pick a discount factor $\\gamma \\in (0,1).$
> Next, given a reward function $r$ and negated penalty $q$, we define the Bellman operator $B_{\\gamma}^{\*}$ by
>
> $$B_{\\gamma}^{\*}\[v\](s) = (1 - \\gamma) \\min \\{r(s), q(s)\\} + \\gamma  \\left\[ \\min\\left\\{ \\max \\left\\{ \\max_{a} v(f(s,a)), r(s)  \\right\\},  q(s) \\right\\} \\right\].$$
>
> We define the value function $V_{\\gamma}^{\*}$ to be the solution to the Bellman equation
> $V_{\\gamma}^{\*} = B_{\\gamma}^{\*}\[V_{\\gamma}^{\*}\]$.
> Proposition 3 in \[Hsu et al., RSS 2021\] then states that $V_{\\gamma}^{\*} \\to V^{\*}$ ($V^{\*}$ here is the true RA optimal value function) as $\\gamma \\to 1^-$.
> Hence we are interested in obtaining $V_{\\gamma}^{\*}$.
>
> Now, for each policy $\\pi$ define the Bellman operator $B_{\\gamma}^\\pi$ by
>
> $$B_{\\gamma}^\\pi\[v\](s) = (1 - \\gamma) \\min \\{r(s), q(s)\\} + \\gamma  \\mathbb{E}_{a \\sim \\pi} \\left\[ \\min\\left\\{ \\max \\left\\{ v(f(s,a)), r(s)  \\right\\},  q(s) \\right\\} \\right\].$$
>
> We define the value function $V_{\\gamma}^\\pi$ to be the solution to the discounted SRABE, i.e. the Bellman equation
> $V_{\\gamma}^\\pi = B_{\\gamma}^\\pi\[V_{\\gamma}^\\pi\]$.
> Notice that from the above definitions, for all states $s$ we have
> $V_{\\gamma}^{\\pi}(s) = B_{\\gamma}^\\pi\[V_{\\gamma}^\\pi\](s) = B_{\\gamma}^{\*}\[V_{\\gamma}^{\\pi}\](s)$. Thus, we have
>
> $$V_{\\gamma}^{\\pi} \\le B_{\\gamma}^{\*}\[V_{\\gamma}^{\\pi}\]\\quad\\text{and}\\quad V_{\\gamma}^{\*} = B_{\\gamma}^{\*}\[V_{\\gamma}^{\*}\].$$
>
>
> Since $B$ is a contraction and is also is monotonically increasing (i.e. $v_1 \\le v_2 \\implies B\[v_1\] \\le B\[v_2\]$), the comparison principle for the Bellman equation $v = B_{\\gamma}^{\*}\[v\]$ applies. In particular,
>
> $$V_{\\gamma}^{\\pi} \\le V_{\\gamma}^{\*}.$$
>
> Thus the SRABE allows us to consider stochastic policies (which we need to run PPO), and by what we have shown above no such policy will not overestimate $V_{\\gamma}^{\*}$.
>
> Now let $\\pi^{\*}$ be a deterministic policy which for each state $s$ produces an action $a$ that maximizes $V_{\\gamma}^{\*}(f(s,a))$.
> It can then be shown that $V_{\\gamma}^{\\pi^{\*}} = V_{\\gamma}^{\*}$.
> So not only will no solution to the SRABE overshoot $V_{\\gamma}^{\*}$, but it is in addition not conservative to search for the optimal policy $\\pi^{\*}$ satisfying the SRABE.

---

> ### Author Response · Authors · 2025-11-20
>
> ***State augmentation cost: The proposed augmentation grows the state with running max/min signals; for higher-dim tasks (real robots, multi-goal specs) this could be heavy, and the paper doesn’t study scalability.***
>
> This is a great question. One of the nice aspects about the decomposition is that the augmented state space is only required to decide when to switch between optimal policies for the decomposed problems. We did not emphasize this enough in the paper, and will add a remark in the main text to do so.
>
> In particular, this means that the basic R, A, and RA problems can be solved on the original state-space. One could still run into scaling issues regarding learning when to switch between policies, but for many practical applications there are heuristics to do so (e.g. switching policies after reaching a target).
>
> That said, as tasks become more complex than those we have explored in the scope of the text (e.g. hitting a large number of targets), there will almost certainly be scaling issues. This is at some level a fundamental limitation of almost all algorithms that attempt to solve similar problems, and our approach unfortunately is not able to circumvent it. We will clarify this in the conclusion section.

---

> ### Author Response · Authors · 2025-11-20
>
> ***The paper lacks sample-complexity or contraction analysis for the composed operators. Prior HJ-RL work leans on discounted versions to get contraction, and here the composed PPO version is justified mostly by analogy.***
>
> While it is true that we do not consider discounted versions of the composed problems to obtain contraction, discounting and contraction are indeed fundamental to our ability to compute the optimal value functions for these objectives. It is just that the discounting is employed after the decomposition, not before it.
>
> More explicitly, after performing the relevant decomposition into (undiscounted) R/A/RA objectives, we follow [Hsu et al. 2021], approximating the optimal value functions for these objectives using discounted versions to get contraction (with $\gamma = 1 - \varepsilon$, $\varepsilon \ll 1$). These approximations are justified using that work’s theoretical results, which establish that the true R, A, and RA optimal value functions are achieved in the limit $\gamma \to 1^{-}$.
>
> In DOHJ-PPO, a PPO-variant (modified similarly to [So et al. 2024]) is run on these decomposed problems, with bootstrapping between them. Unfortunately, even standard PPO does not generally come with convergence guarantees, so we must justify this piece of our methodology empirically rather than theoretically. We do indeed draw inspiration from [So et al 2024] for the design of our PPO-variant DOHJPPO, but these authors also justified their PPO-variant using empirical evidence (once the GAE estimator is included) for this reason. We will clarify these limitations in the conclusion.
>
> With regard to sample-complexity, we appreciate the reviewer pointing out the lack of such an analysis. We have included time-courses of learning progress in the ablation experiment mentioned above. Please refer to Figure 5 in the updated text for results.

---

### Official Review · Reviewer_Cdby · 2025-11-02

**Soundness:** 3
**Presentation:** 3
**Contribution:** 2
**Rating:** 6
**Confidence:** 4

**Summary:**

The paper defines two dual-objective, extremal value functions in reach-avoid optimal control problems: "**reach and always avoid**" (RAA) and "**reach and reach**" (RR),  and shows each can be decomposed into known HJ-RL primitives of Reach (R), Avoid (A), Reach-Avoid (RA), yielding explicit Bellman targets and corresponding algorithms that train a composed actor–critic together with additional critics/actors for the sub-objectives.
The paper argues history is necessary and introduces a minimal augmentation (running max/min) with a sufficiency result.
The proposed value function decompositions was combined with deep Q-learning (DQN) and proximal policy optimization (PPO) and tested in a simple grid-world and modified continuous control environments (Hopper, F16, SafetyGym, and HalfCheetah).

**Strengths:**

- **Originality and significance**: moderate. The problems studied in this paper are classical in logic terms, but the decomposition is novel and practically attractive.
- **Quality and correctness**: the reductions using min/max identities and distributivity/commutativity under deterministic transition and policy assumptions (relaxed later) seem sound and reasonable for the stated setting. Proofs were not carefully checked.
- **Clarity**: This paper is clear, easy to follow, and contextualized well. Section 2 related works is comprehensive yet concise, very useful for readers. The most relevant works (e.g., `Fisac et al. (2019)`, `Hsu et al. (2021)`) were cited throughout and explained well. Figure 1 is a bit confusing (Figure 2 is more understandable for me) and dense, but it illustrates the problem setting to some extent.
- **Compositional formulation**: converting RAA and RR to R, A, and RA can be helpful for compositionality and generalization.

**Weaknesses:**

## Formal relation to temporal logic

I believe the value function decompositions are based on some algebra/logic implicitly.
It would help readers if the paper states the specs more explicitly, potentially using temporal logic, even if the proposed method does not explicitly use automata.

My feeling is that this paper is just doing quantitative semantics for temporal modal operators $F$ **f**inally, $G$ **g**lobally/always, and $U$ **u**ntil.

Let $r, p, q$ be temporal propositions.
Then,
- R is $F r$ "finally reach r"
- A is $G \lnot p$ "always avoid (not reach) p"
- RA is $\lnot p U r$ "avoid p until reach r"
- RAA is $F r \land G \lnot p$ "(finally reach r) and (always avoid p)"
- RR is $F p \land F q$ "(finally reach p) and (finally reach q)"

The first three have quantitative semantics:
- R: $F r \leftrightarrow \max_t r_t$
- A: $G \lnot p \leftrightarrow \min_t -p_t$
- RA: $\lnot p U r \leftrightarrow \max_t \min\\{r_t, \min_{t' \leq t} -p_{t'}\\}$
which were shown in Section 4.

It is possible to prove
- RAA: $F r \land G \lnot p \equiv (\lnot p U r) \land (G \lnot p)$  = "(avoid p until reach r) and (always avoid p)" (?)
- RR: $F p \land F q \equiv F((p \land F q) \lor (q \land F p))$ "finally ((reach p and finally reach q) or (reach q and finally reach p))"

The LHSs have quantitative semantics given in Section 3:
- RAA: $\min\\{\max_t r_t, \min_t -p_t\\}$
- RR: $\min\\{\max_t p_t, \max_t q_t\\}$

The RHSs have quantitative semantics given in Section 6. (small gaps)

The point is, there exists a more principled approach to derive these value functions and Bellman equations than the lengthy *10+ page proofs* in the appendix, which are more error-prone.

The author stated that "*While the problems we attempt to solve (e.g. reaching multiple goals) can be thought of as specific instantiations of LTL specifications, our approach to solving these problems is fundamentally different from those in this line of work.*," but being "fundamentally different" doesn't mean it's better, more efficient, or more generalizable.
There's no evidence that LTL cannot express RAA or RR.
Even if it's true, there alternatives such as metric temporal logic (MTL) or signal temporal logic (STL).
I'm not an expert in temporal logic, but the concept of **monitoring** is very relevant to the state augmentation issue.
The author is encouraged to further connect this paper with the existing logic literature;
if they decide not to do so, at least justify the "min/max algebra" approach by spelling out the axioms and operations (commutativity, associativity, distributivity, idempotence, absorption, monotonicity, etc.).

## Complexity: why RA?

The second issue is pertinent to the first one.
This paper decided to convert RAA and RR in to R, A, and RA to "leverage new HJ-based methods on the subproblems", because these HJ-RL "primitives" have been previously studied.
However, after scrutinizing Sections 3, 4 and 6, it seems to me this approach unnecessarily complexifies the problem?
Why not directly derive the value functions and Bellman equations from the specifications in Section 3?
Especially, the use of RA in RAA seems strange.
Why not just R and A?

"Focusing on extremal values rather than discounted sums", "directly specifying desired behavior", and composing task specs, values, and policies is a nice direction.
However, the RL community has developed several methods for general learning objectives `Wang et al. (2020)`, `Cui & Yu (2023)`, `Tang et al. (2025)`.
Please compare this work with them, and explain why the proposed method is preferred for the stated tasks.

## MDP without rewards and need for augmented states

Figure 2 and Section 5 might be misleading because each node seems to be considered as a state, and whether a node has been reached isn't part of the state.
"Non-augmented" MDPs/policies are flawed because this paper is actually using a "**MDP without rewards**" setting: the MDP $M = (S, A, f)$ is a triple and only consists of the transition function, and the reward function is not on the environment side but designed by the agent.
Although I believe this setting is indeed more natural and should be used, but since the more widely used formulation considers the reward function as a part of MDP, it's incorrect to consider a node as a state, otherwise the reward function is not well defined.

"Node not reached/with an item" and "node reached/without an item" should be two distinct states;
"Reaching an unvisited node/collecting an item" and "reaching a previously visited node" should generate different observations/rewards (e.g., locations + a bunch of flags).
The flaw is not due to the limitations of "memoryless" policies; it's because of the wrongly chosen states.

## Stochastic reach-avoid

This paper extended reach-avoid problem formulations using min/max to the stochastic setting in Section 7.1.
However, the value function and Bellman equation (SRABE) were given without sufficient justification.

Caveat: *expected maximum* and *maximum expectation* are two different objectives, and consequently their value functions are also different.
One may argue that both are meaningful objectives, but the corresponding value functions and Bellman equations should not be *defined* but *derived* from the objective.
The error of interchanging max and expectation in `Quah & Quek (2006)` was noticed by `Gottipati et al. (2020)` and `Cui & Yu (2023)` and some fixes were proposed.
The author is encouraged to explicitly state the learning objective in Section 5 like what they did in Section 3 and confirm the Bellman equation truly corresponds to this objective.

## References

- Kian Hong Quah and Chai Quek. **Maximum reward reinforcement learning: A non-cumulative reward criterion.** Expert Systems with Applications, 31(2):351–359, 2006. https://doi.org/10.1016/j.eswa.2005.09.054.
- Sai Krishna Gottipati, Yashaswi Pathak, Rohan Nuttall, Raviteja Chunduru, Ahmed Touati, Sriram Ganapathi Subramanian, Matthew E Taylor, and Sarath Chandar. **Maximum reward formulation in reinforcement learning.** arXiv preprint, 2020. https://arxiv.org/abs/2010.03744.
- Ruosong Wang, Peilin Zhong, Simon S Du, Russ R Salakhutdinov, and Lin Yang. **Planning with general objective functions: Going beyond total rewards.** In Neural Information Processing Systems, 2020. https://proceedings.neurips.cc/paper/2020/hash/a6a767bbb2e3513233f942e0ff24272c-Abstract.html
- Wei Cui and Wei Yu. **Reinforcement learning with non-cumulative objective.** IEEE Transactions on Machine Learning in Communications and Networking, 1:124–137, 2023. https://doi.org/10.1109/TMLCN.2023.3285543.
- Yuting Tang, Yivan Zhang, Johannes Ackermann, Yu-Jie Zhang, Soichiro Nishimori, Masashi Sugiyama. **Recursive Reward Aggregation.** In Reinforcement Learning Conference, 2025. https://openreview.net/forum?id=13lUcKpWy8

**Questions:**

Minor issues
- The introduction employs strong intensifiers (e.g., “performing far more safely”) without quantitative context. This reads like forward-referencing results.
- Apostrophes for plurals. "SBE’s and RABE’s" $\to$ SBEs and RABEs (no apostrophes for plural acronyms)
- l.049: user?
- l.050: "Figure 1, top middle-left" is awkward. Consider using subcaptions (e.g., Figure 1a).
- l.050-051: Consider avoiding F16 and Hopper, because readers may not know them if they are not familiar with these environments.

---

> ### Author Response · Authors · 2025-11-20
> **Formal relation to temporal logic**
>
> Firstly, let us thank the author for their extensive consideration of our work. We will address each critique, one at a time:
>
> ***I believe the value function decompositions are based on some algebra/logic implicitly. It would help readers if the paper states the specs more explicitly, potentially using temporal logic, even if the proposed method does not explicitly use automata... The point is, there exists a more principled approach to derive these value functions and Bellman equations than the lengthy 10+ page proofs in the appendix, which are more error-prone...***
>
> We thank the reviewer for this question and this suggestion. It is indeed possible to phrase the Hamilton-Jacobi reachability (HJR) problem formulation using the language of Temporal Logic (TL) quantitative semantics. In particular, the standard HJR problem is to find the sequence of actions that maximizes some objective function, and this objective function can generally be written as a quantitative semantic corresponding to the specification of the desired system behavior (however, this is not explicitly written in TL notation in the HJR literature).
>
> That said, the optimal value function decomposition results for the RR ($F \psi_1 \land F \psi_2$) and RAA ($F \psi \land G \lnot \phi$) are distinct from the decomposition of the corresponding quantitative semantics. In other words, our decomposition results are statements about the optimal control associated with the quantitative semantic, not the quantitative semantic itself.
>
> This subtle distinction can be clarified by the following counter-example, where a valid decomposition of the quantitative semantic (TL) for the RAA problem does not translate to a valid decomposition of the optimal value functions (HJ) for the RAA problem.
>
> Consider an RAA problem where my friend is holding a piñata which I would like to break with a bat. We can always decompose the RAA quantitative semantic into R and A quantitative semantics.
> However, suppose there is some state of the system from which I can either eventually hit the piñata or always avoid hitting my friend, but not both. In this case, the optimal value function for the R and A problems will both be non-negative, even though I cannot actually achieve the RAA task from this state.
>
> To make this argument explicit, define the atomic predicates $\\varphi$ and $\\psi$ to represent hitting the piñata and hitting my friend, respectively:
>
> $$(\\mathbf{x}, t) \\models \\varphi \\iff r(\\mathbf{x}(t)) \\ge 0,$$
> $$(\\mathbf{x}, t) \\models \\psi \\iff p(\\mathbf{x}(t)) \\ge 0.$$
>
> Given a predicate $\\mu$, let $\\rho_\\mu$ be the corresponding quantitative semantic.
> Thus, $\\rho_\\varphi \[\\mathbf{x}, t\] = r(\\mathbf{x}(t))$ and $\\rho_\\psi\[\\mathbf{x}, t\] = p(\\mathbf{x}(t))$.
> The quantitative semantics for the R, A, and RAA problems are then, respectively:
> \\begin{align}
>     \\rho_{\\mathrm{R}}\[\\mathbf{x}, t\] &:= \\rho_{F\\varphi}\[\\mathbf{x}, t\] = \\max_{\\tau \\ge t} r(\\mathbf{x}(\\tau)), \\\\
>     \\rho_{\\mathrm{A}}\[\\mathbf{x}, t\] &:= \\rho_{G\\lnot \\psi}\[\\mathbf{x}, t\] =  \\min_{\\tau \\ge t} -p(\\mathbf{x}(\\tau)), \\\\
>     \\rho_{\\mathrm{RAA}}\[\\mathbf{x}, t\] &:= \\rho_{(F\\varphi) \\land (G\\lnot \\psi)}\[\\mathbf{x}, t\] = \\min\\left\\{\\max_{\\tau \\ge t} r(\\mathbf{x}(\\tau)) , \\min_{\\tau \\ge t} -p(\\mathbf{x}(\\tau)) \\right\\}.
> \\end{align}
>
> As mentioned earlier, the optimal value functions for the R, A, and RAA problems can then be written in terms of the quantitative semantics, i.e.
>
> $$V_{\\mathrm{R}}^{\*}(s) = \\max_{\\pi} \\rho_{\\mathrm{R}}\[\\mathbf{x}_{s}^{\\pi}, 0\],$$
>
> $$V_{\\mathrm{A}}^{\*}(s) = \\max_{\\pi} \\rho_{\\mathrm{A}}\[\\mathbf{x}_{s}^{\\pi}, 0\],$$
>
> $$V_{\\mathrm{RAA}}^{\*}(s) = \\max_{\\pi} \\rho_{\\mathrm{RAA}}\[\\mathbf{x}_{s}^{\\pi}, 0\],$$
>
> where $\\mathbf{x}_{s}^{\\pi}$ is the trajectory of the bat corresponding to the initial state $s$ and policy $\\pi$.
>
> But here is the key point: although it is always true that
>
> $$\\rho_{\\mathrm{RAA}}\[\\mathbf{x}, t\] = \\min\\{\\rho_{\\mathrm{R}}\[\\mathbf{x}, t\],  \\rho_{\\mathrm{A}}\[\\mathbf{x}, t\]\\},$$
>
> in general we only have that
>
> $$V_{\\mathrm{RAA}}^{\*}(s) \\le \\min\\{V_{\\mathrm{R}}^{\*}(s), V_{\\mathrm{A}}^{\*}(s) \\}.$$
>
> Indeed this inequality is sometimes strict.
> Again, consider the case where there is some state of the system $s$ from which I can either eventually hit the piñata or always avoid hitting my friend, but cannot do both.
> Then
>
> $$V_{\\mathrm{RAA}}^{\*}(s) < 0 \\le \\min\\{V_{\\mathrm{R}}^{\*}(s), V_{\\mathrm{A}}^{\*}(s) \\}.$$
>
> In other words, the algebra that applies to the TL predicates translates to an algebra on the TL quantitative semantics, but it does not generally translate to an algebra on the optimal value functions. This is why our decomposition required thorough justification.

---

> > ### Comment · Reviewer_Cdby · 2025-11-28
> >
> > Thank you for this detailed counter-example.
> >
> > > the algebra that applies to the TL predicates translates to an algebra on the TL quantitative semantics, but it does not generally translate to an algebra on the optimal value functions
> >
> > - Is this an intrinsic limitation of TL quantitative semantics?
> > - Is it because of the "quantitative negation"?
> > - Do we also have such a counter-example for RR problems?

---

> ### Author Response · Authors · 2025-11-20
> **Complexity: why RA?**
>
> ***The second issue is pertinent to the first one. This paper decided to convert RAA and RR in to R, A, and RA to "leverage new HJ-based methods on the subproblems", because these HJ-RL "primitives" have been previously studied. However, after scrutinizing Sections 3, 4 and 6, it seems to me this approach unnecessarily complexifies the problem? Why not directly derive the value functions and Bellman equations from the specifications in Section 3? Especially, the use of RA in RAA seems strange. Why not just R and A?***
>
> This is a great question, which gets at the heart of the decomposition of the RR and RAA optimal value functions. In the last example, we demonstrated we can indeed decompose the RAA quantitative semantic into quantitative semantics for the R and A problems. As discussed above, however, this *does not* generally translate to a decomposition of the RAA optimal value function into R and A optimal value functions. In the previous example, this was highlighted by the following *false* statement: I can complete the RAA task if and only if I am in a state from which I can reach the piñata and I am in a state from which I can avoid hitting my friend.
>
> Note, the minimum of the two optimal value functions corresponds to simultaneously considering two different action sequences, each optimal for reaching and avoiding independently. In the dual-objective problem, however, we only have one set of actions but simultaneously need to accomplish both tasks.
>
> The RAA task *can* be completed if and only if I can (1) hit the piñata without first hitting my friend (RA) in a fashion that (2) also allows me to avoid hitting my friend after I have hit the piñata (A). In other words, I want my bat to reach the set of states for which the bat is both touching the piñata and is in the avoid set of my friend, while also avoiding hitting my friend before reaching this set.
>
> Although this may seem intuitive, we have added a direct derivation of the RAA Bellman equation to the appendix which highlights the added mathematical complexity when simultaneously dealing with the quantitative semantics ($\\max_t$ and $\\min_t$) and optimal control ($\\max_{a_0, a_1, \\dots}$). For details, see Appendix A1. A Direct Derivation of the RAA Bellman Equations.
>
> ***"Focusing on extremal values rather than discounted sums", "directly specifying desired behavior", and composing task specs, values, and policies is a nice direction. However, the RL community has developed several methods for general learning objectives Wang et al. (2020), Cui & Yu (2023), Tang et al. (2025)...***
>
> We thank the reviewer for connecting our work to some existing methods for clear comparison. These works consider maximum-reward formulations and reproduce some of the same results in [Fisac et al. 2019], however, the aforementioned works are not tailored to the kind of problems we are studying.
>
> A few key differences are that (1) our methods are specifically designed for problems which involve multiple rewards / rewards and penalties, and (2) our methods are fundamentally rooted in decomposing larger safety problems into fundamental ones for which we have efficient solution methods. Also, on a practical level, we translate the decomposition theory into the DOHJ-PPO algorithm, which involves boot-strapping between decomposed components, each encoded in an actor-critic pair, using an adaptation of PPO. This algorithm is built to take advantage of the decompositional results.
>
> More specifically, the work in [Wang et al. 2020]  is only formulated for single scalar reward functions. Even if one were to overlook this issue, the aggregation functions $f$ used to integrate these rewards / penalties are not symmetric (in either the RR or RAA case), which is a key assumption of this paper. This lack of symmetry is due to the presence of the distinct rewards (RR) / both rewards and penalties (RAA).
>
> While it is possible to cast the RR and RAA problems into the form investigated in [Cui & Yu 2023], one must first at least perform some sort of state augmentation to do so, as to keep track of the quality function in the Bellman update (8) in [Cui & Yu]. Even if one were to do so, however, the combination of the multiple rewards / penalties without a decompositional approach would be likely to result in inefficient learning, as the value function only increases when the lesser of the two objectives in the dual-objective problem increases. Moreover, the algorithm in [Cui & Yu 2023] operates on the whole, augmented state-space during the entire learning process.
>
> By contrast, our algorithm proceeds by decomposition. In each decomposed objective, the above issues with competing objectives and larger state-spaces are not present. This is the key to how our algorithm is able to approach dual-objective problems in a way that the others are not tailored for. Similar differences hold when comparing our approach to that of [Tang et al. 2025]

---

> > ### Comment · Reviewer_Cdby · 2025-11-28
> >
> > Thank you for the explanation. I'm not sure if I fully understand why RA and RAA are chosen, but I understand better that
> >
> > > In the dual-objective problem, however, we only have one set of actions but simultaneously need to accomplish both tasks.
> >
> > Now, let's consider some more complex, hypothetical real-world scenarios. In applications such as human-robot interaction and autonomous driving, the specs of the goals can be more complex, can come and change continuously, or need to be defined on the fly. Maybe it's not dual-objective but Reach-A-Always-Avoid-B-After-Reach-C. Or a car should always drive within the speed limit except in an emergency. Or simply a malicious instruction "give up the piñata and hit your friend Alice and then Bob now". Can we decompose these goals (using RA and RAA or something else) to obtain an optimal policy without extra training?
> >
> > While this does not affect the decision for this paper, such a discussion can help readers understand the importance of RA, RAA, and goal decomposition.

---

> > > ### Author Response · Authors · 2025-12-03
> > >
> > > **Thank you for the explanation... While this does not affect the decision for this paper, such a discussion can help readers understand the importance of RA, RAA, and goal decomposition.**
> > >
> > > This is a great question that helps contextualize the RR and RAA problems in a manner that also points towards possible extensions. The RR and RAA problems indeed provide two of the fundamental building blocks needed when solving larger tasks. For example (if the authors are understanding your specification correctly), the Reach-A-Always-Avoid-B-After-Reach-C can be decomposed into an RAA problem and an R problem. Specifically, the inner operation is Reach-A-Always-Avoid-B and the outer operation is to reach the intersection of this RAA set and C.
> > >
> > > As more layers are added to the specifications, the training process will become more complex. For example, computing the RAA involves solving an R, A, and RA problem. To add in the After-Reach-C part requires yet another R problem. As such, ways to scale toward very complex tasks in general will require more work, including algorithmic approaches to decomposition, more efficient representations, and likely heuristics based on system knowledge to improve sampling efficiency, as is typical in such problems.
> > >
> > > On a related note, this work focused on extending the HJR approach to RL to the most mathematically fundamental compositional tasks: the RR and RAA. While this framework can be leveraged toward the more complex tasks discussed above, more work would be needed to be able to solve all TL specifications using HJR. We will note that future work is required to solve such general problems.
> > >
> > > We have added information regarding these future directions to the conclusion.

---

> > ### Comment · Reviewer_Cdby · 2025-11-28
> >
> > Indeed, [Wang et al. 2020] [Cui & Yu 2023] only study scalar reward functions. [Wang et al. 2020] even requires the aggregation function to be symmetric, so maybe it's not suitable for temporal goals. There are also works that study the composition and decomposition of multiple reward signals (not necessarily multi-objective), but they mainly focus on discounted sum. Including a broader contextualization and literature review in the revised version can highlight the unique challenges and contributions of this work.
> >
> > - Hybrid Reward Architecture for Reinforcement Learning
> > - Consistent Aggregation of Objectives with Diverse Time Preferences Requires Non-Markovian Rewards
> > - RD^2: Reward Decomposition with Representation Disentanglement

---

> > > ### Author Response · Authors · 2025-12-03
> > >
> > > **Indeed, [Wang et al. 2020] [Cui & Yu 2023] only study scalar reward functions. [Wang et al. 2020] even requires the aggregation function to be symmetric, so maybe it's not suitable for temporal goals. There are also works that study the composition and decomposition of multiple reward signals (not necessarily multi-objective), but they mainly focus on discounted sum. Including a broader contextualization and literature review in the revised version can highlight the unique challenges and contributions of this work.**
> > >
> > >
> > > We thank the reviewer for the literature suggestions. We wil havel added these articles and the previous ones discussed to our Related Works section, contrasting the goals of these works with the contributions of our own.

---

> ### Author Response · Authors · 2025-11-20
> **MDP without rewards and need for augmented states**
>
> ***Figure 2 and Section 5 might be misleading because each node seems to be considered as a state, and whether a node has been reached isn't part of the state. "Non-augmented" MDPs/policies are flawed because this paper is actually using a "MDP without rewards" setting: the MDP is a triple and only consists of the transition function, and the reward function is not on the environment side but designed by the agent. Although I believe this setting is indeed more natural and should be used, but since the more widely used formulation considers the reward function as a part of MDP, it's incorrect to consider a node as a state, otherwise the reward function is not well defined. "Node not reached/with an item" and "node reached/without an item" should be two distinct states; "Reaching an unvisited node/collecting an item" and "reaching a previously visited node" should generate different observations/rewards (e.g., locations + a bunch of flags). The flaw is not due to the limitations of "memoryless" policies; it's because of the wrongly chosen states.***
>
> We thank the reviewer for this detailed and thorough question. The answer here is nuanced and merits a lengthy response. The reviewer is correct that it is a common mistake in RL for a user to work with a stochastic process that either does not have a Markovian state transition or a Markovian reward, and a common way to repair such a problem is to add extra states. For example, if an agent gets a positive reward upon first visiting a state and a zero reward thereafter, this reward is not Markovian, and a “copy” of that state should be added as the reviewer suggests.
>
> However, this is not the situation in our work. In the examples in Figure 3, the agent receives the same reward/penalty each time a target/hazard is hit (not just the first time). **In general, even without state-augmentation, the rewards and penalties are Markovian** in the sense that they only depend on the current state, not state history. The underlying issue is not that the rewards and penalties fail to be Markovian, but instead that, unlike a traditional RL setup, these rewards and penalties are not summed over time in the agent’s objective. Rather, they are aggregated via multiple min and max operations.
>
> **The reason that we do augment the state space is that – unlike discounted-sum rewards (or  R, A, and RA problems) – when considering the RAA and RR problems, the optimal path may revisit a state, and thus an agent may be overly constrained with only a policy on the original state space.** This is most clear in the RR problem on the left of Figure 3, because here the agent cannot find a deterministic policy which hits both targets, solely due to the topology of the state space. A similar issue also applies in the RAA problem on the right of Figure 3. One may note that allowing for stochastic policies would help in these small examples, but on larger state spaces similar problems still arise.
>
> Thus the solution is to augment the state space to allow the agent to choose from a richer set of policies. We prove in the appendix that our particular choice of augmentation is ideal in the sense that there is a policy which can achieve the same value of the objective function as any sequence of actions over time.
> This need for augmentation to achieve optimal behavior is one of the odd but interesting nuances of these compositional max/min problems which motivated some of the extensive theoretical work in the appendix. As a final note, one may at first be concerned with the burden of the growing state-space, but we show in the appendix (in the course of proving Lemmas 5 and 10) that these extra states are needed solely to decide when to transition between the value functions in the decomposition. This observation helps keep the learning process efficient in practice.

---

> ### Author Response · Authors · 2025-11-20
> **Stochastic reach-avoid**
>
> ***This paper extended reach-avoid problem formulations using min/max to the stochastic setting in Section 7.1. However, the value function and Bellman equation (SRABE) were given without sufficient justification.***
>
> ***Caveat: expected maximum and maximum expectation are two different objectives, and consequently their value functions are also different. One may argue that both are meaningful objectives, but the corresponding value functions and Bellman equations should not be defined but derived from the objective. The error of interchanging max and expectation in Quah & Quek (2006) was noticed by Gottipati et al. (2020) and Cui & Yu (2023) and some fixes were proposed. The author is encouraged to explicitly state the learning objective in Section 5 like what they did in Section 3 and confirm the Bellman equation truly corresponds to this objective.***
>
> This is a great point, and we appreciate the reviewer’s careful consideration and observation of our stochastic relaxations. Before we get into theoretical justification, let us note that the SRABE is not directly related to the RR or RAA problems. We introduced the SRABE for a practical reason: to obtain an extension of the SRBE (introduced in [So. et al. 2024, NeurIPS]), which is for a Reach problem, to a Reach-Avoid problem. We do this because Reach-Avoid problems are used in the decomposition, but no Reach-Avoid equivalent of the SRBE was included in the previous work. In some sense, the true motivation for the SRABE was the strong performance of using the SRBE with PPO in the previous work, although any other method to estimate RA value functions using PPO could have been used instead.
>
> That said, if the reviewer is interested in theoretical justification for the use of the SRBE / SRABE, the following argument may be of interest. As the reviewer mentions in the caveat, the value function for the expected value of the RA objective function (i.e. quantitative semantic) under a stochastic policy will not generally satisfy the SRABE, since the expected max/min and max/min expectation are not equal. The theoretical justification for the use of the SRABE in our context comes not from an explicit value function but instead from a comparison principle:
>
> First, pick a discount factor $\\gamma \\in (0,1).$
> Next, given a reward function $r$ and negated penalty $q$, we define the Bellman operator $B_{\\gamma}^{\*}$ by
>
> $$B_{\\gamma}^{\*}\[v\](s) = (1 - \\gamma) \\min \\{r(s), q(s)\\} + \\gamma  \\left\[ \\min\\left\\{ \\max \\left\\{ \\max_{a} v(f(s,a)), r(s)  \\right\\},  q(s) \\right\\} \\right\].$$
>
> We define the value function $V_{\\gamma}^{\*}$ to be the solution to the Bellman equation
> $V_{\\gamma}^{\*} = B_{\\gamma}^{\*}\[V_{\\gamma}^{\*}\]$.
> Proposition 3 in \[Hsu et al., RSS 2021\] then states that $V_{\\gamma}^{\*} \\to V^{\*}$ ($V^{\*}$ here is the true RA optimal value function) as $\\gamma \\to 1^-$.
> Hence we are interested in obtaining $V_{\\gamma}^{\*}$.
>
> Now, for each policy $\\pi$ define the Bellman operator $B_{\\gamma}^\\pi$ by
>
> $$B_{\\gamma}^\\pi\[v\](s) = (1 - \\gamma) \\min \\{r(s), q(s)\\} + \\gamma  \\mathbb{E}_{a \\sim \\pi} \\left\[ \\min\\left\\{ \\max \\left\\{ v(f(s,a)), r(s)  \\right\\},  q(s) \\right\\} \\right\].$$
>
> We define the value function $V_{\\gamma}^\\pi$ to be the solution to the discounted SRABE, i.e. the Bellman equation
> $V_{\\gamma}^\\pi = B_{\\gamma}^\\pi\[V_{\\gamma}^\\pi\]$.
> Notice that from the above definitions, for all states $s$ we have
> $V_{\\gamma}^{\\pi}(s) = B_{\\gamma}^\\pi\[V_{\\gamma}^\\pi\](s) = B_{\\gamma}^{\*}\[V_{\\gamma}^{\\pi}\](s)$. Thus, we have
>
> $$V_{\\gamma}^{\\pi} \\le B_{\\gamma}^{\*}\[V_{\\gamma}^{\\pi}\]\\quad\\text{and}\\quad V_{\\gamma}^{\*} = B_{\\gamma}^{\*}\[V_{\\gamma}^{\*}\].$$
>
>
> Since $B$ is a contraction and is also is monotonically increasing (i.e. $v_1 \\le v_2 \\implies B\[v_1\] \\le B\[v_2\]$), the comparison principle for the Bellman equation $v = B_{\\gamma}^{\*}\[v\]$ applies. In particular,
>
> $$V_{\\gamma}^{\\pi} \\le V_{\\gamma}^{\*}.$$
>
> Thus the SRABE allows us to consider stochastic policies (which we need to run PPO), and by what we have shown above no such policy will not overestimate $V_{\\gamma}^{\*}$.
>
> Now let $\\pi^{\*}$ be a deterministic policy which for each state $s$ produces an action $a$ that maximizes $V_{\\gamma}^{\*}(f(s,a))$.
> It can then be shown that $V_{\\gamma}^{\\pi^{\*}} = V_{\\gamma}^{\*}$.
> So not only will no solution to the SRABE overshoot $V_{\\gamma}^{\*}$, but it is in addition not conservative to search for the optimal policy $\\pi^{\*}$ satisfying the SRABE.
>
> The above gives the true intuition for why the SRABE is so useful: it allows one to optimize over stochastic policies, and thus use PPO, without either overshooting the value function one is actually interested in, nor being overly conservative.

---

> ### Author Response · Authors · 2025-12-03
>
> **Thank you for this detailed counter-example...**
>
> The authors are pleased to hear we have been able to clarify this subtle but important point regarding how the decompositions in our work cannot be derived from algebraic manipulations of the quantitative semantics.
>
> ***Is this an intrinsic limitation of TL quantitative semantics?***
>
> The simple answer to this question is yes, but such an oversimplification is somewhat unfair to the TL literature. A more complete answer would be that TL is a useful language for describing what needs to happen in the environment for a task to be completed, but it does not on its own specify how the agent should go about choosing actions to complete the desired task. HJR methods focus on the latter problem, with the optimal value function being the central object of interest. In other words, these two frameworks, while complimentary, were designed to address different problems.
>
> ***Is it because of the "quantitative negation"?***
>
>  “Negation”, “and”, and “always” operators in a TL formula can all result in similar issues because the maximization over the actions does not generally commute with the minimums or negatives corresponding to such operations that appear in the quantitative semantic. Indeed, see the counter-example below for the RR problem, which has no “negation” or “always” operator, but where there is still an issue due to the “and” operation.
>
> ***Do we also have such a counter-example for RR problems?***
>
> Yes, we do. Suppose that we are solving an RR problem for a small robotic boat that must deliver supplies to two downstream islands in a wide river that flows from north to south. The islands are at the same latitude, but one is to the west and one is to the east. Suppose the boat starts far enough upstream that it can reach either island, but the current is too strong to move from one island to the other. Then the boat can satisfy the R task for the west island (by ignoring the east island) and it can satisfy the R task for the east island (by ignoring the west island), but it cannot satisfy the RR task. The decomposition of the quantitative semantics for the RR problem into the minimum of the quantitative semantics for the two R problems will then not translate to a valid decomposition of the value functions.
>
> More explicitly, with $r_1$ and $r_2$ the signed distance functions to either island, let
>
> $$(\\mathbf{x}, t) \\models \\varphi \\iff r_1(\\mathbf{x}(t)) \\ge 0,$$
> $$(\\mathbf{x}, t) \\models \\psi \\iff r_2(\\mathbf{x}(t)) \\ge 0.$$
>
> Thus, $\\rho_\\varphi \[\\mathbf{x}, t\] = r_1(\\mathbf{x}(t))$ and $\\rho_\\psi\[\\mathbf{x}, t\] = r_{2}(\\mathbf{x}(t))$.
> The quantitative semantics for reaching the west island, reaching the east island, and reaching both islands are then respectively,
> \\begin{align}
>     \\rho_{\\mathrm{R1}}\[\\mathbf{x}, t\] &:= \\rho_{F\\varphi}\[\\mathbf{x}, t\] = \\max_{\\tau \\ge t} r_1(\\mathbf{x}(\\tau)), \\\\
>     \\rho_{\\mathrm{R2}}\[\\mathbf{x}, t\] &:= \\rho_{F\\psi}\[\\mathbf{x}, t\] =  \\max_{\\tau \\ge t} r_2(\\mathbf{x}(\\tau)), \\\\
>     \\rho_{\\mathrm{RR}}\[\\mathbf{x}, t\] &:= \\rho_{(F\\varphi) \\land (F\\psi)}\[\\mathbf{x}, t\] = \\min\\left\\{\\max_{\\tau \\ge t} r_1(\\mathbf{x}(\\tau)) , \\max_{\\tau \\ge t} r_2(\\mathbf{x}(\\tau)) \\right\\}.
> \\end{align}
>
> As before, the optimal value functions for the two R problems and the RR problems can then be written in terms of the quantitative semantics, i.e.
>
> $$V_{\\mathrm{R1}}^{\*}(s) = \\max_{\\pi} \\rho_{\\mathrm{R1}}\[\\mathbf{x}_{s}^{\\pi}, 0\],$$
>
> $$V_{\\mathrm{R2}}^{\*}(s) = \\max_{\\pi} \\rho_{\\mathrm{R2}}\[\\mathbf{x}_{s}^{\\pi}, 0\],$$
>
> $$V_{\\mathrm{RR}}^{\*}(s) = \\max_{\\pi} \\rho_{\\mathrm{RR}}\[\\mathbf{x}_{s}^{\\pi}, 0\].$$
>
> But we still have the analogous issue: although it is always true that
>
> $$\\rho_{\\mathrm{RR}}\[\\mathbf{x}, t\] = \\min\\{\\rho_{\\mathrm{R1}}\[\\mathbf{x}, t\],  \\rho_{\\mathrm{R2}}\[\\mathbf{x}, t\]\\},$$
>
> in general we only have that
>
> $$V_{\\mathrm{RR}}^{\*}(s) \\le \\min\\{V_{\\mathrm{R1}}^{\*}(s), V_{\\mathrm{R2}}^{\*}(s) \\}.$$
>
> When we begin from some state of the system $s$ from which I can eventually reach the east island or I can eventually reach the west island, but not both, then the inequality is strict:
>
> $$V_{\\mathrm{RR}}^{\*}(s) < 0 \\le \\min\\{V_{\\mathrm{R1}}^{\*}(s), V_{\\mathrm{R2}}^{\*}(s) \\}.$$
>
> This counter-example was added to Section L of the appendix.

---

> ### Author Response · Authors · 2025-12-03
>
> Finally, we would like to thank this reviewer for catching the typos in the minor comments from their original review! We have amended these errors.

---

### Official Review · Reviewer_FUvU · 2025-11-06

**Soundness:** 3
**Presentation:** 3
**Contribution:** 3
**Rating:** 8
**Confidence:** 2

**Summary:**

This paper investigates two dual-objective satisfactory problems in RL: the reach–reach problem and the reach–always–avoid problem. These problems extend the previously studied reach, avoid, and reach–avoid problems analyzed from the Hamilton–Jacobi perspective of dynamic programming. The authors derive the corresponding Bellman equations for both problems and demonstrate that each can be decomposed into compositions of the previously studied HJ-based reach, avoid, and reach–avoid problems. Furthermore, the paper extends the framework to handle stochastic policies and provides a formulation for implementing it using PPO. Experiments conducted across several environments validate the effectiveness of the proposed method.

**Strengths:**

- The RR and RAA problems are two interesting dual-objective satisfactory problems, and the paper reveals their unique structures that require specialized treatment.

- The paper presents a solid theoretical analysis by deriving the Bellman equations for the RR and RAA problems and elucidating their connections to the previously studied reach, avoid, and reach–avoid problems.

- Empirical results across multiple environments show that the proposed PPO-based method outperforms baseline approaches.

**Weaknesses:**

I do not have major concerns about the paper (possibly due to limited familiarity with the related literature). Some minor points are as follows:
- The paper considers a deterministic transition function. It is unclear whether the proposed method can be extended to the stochastic transition case.
- For stochastic policies, it remains uncertain whether the proposed PPO-based update can guarantee convergence to an optimal policy.

**Questions:**

Please refer to the weaknesses above.

---

> ### Author Response · Authors · 2025-11-20
>
> We thank the reviewer for thoughtful comments and insightful questions. Below, we have addressed the listed weaknesses. Thank you for your time and effort.
>
> ***The paper considers a deterministic transition function. It is unclear whether the proposed method can be extended to the stochastic transition case.***
>
> This is an important observation and opportunity for an extension of this work. Currently, our mathematical theory is only derived for deterministic transitions. At a high level, it is challenging to directly adapt such theory in the case of stochastic transitions because maximum/minimum operations do not generally commute with expectations over a distribution. As such, more intricate theoretical work is required to extend the formal guarantees to the stochastic case.
>
> That said, we can still empirically test how our algorithms perform under stochasticity in the dynamics. Toward this end, **we have added in ablation experiments where increasing amounts of noise are injected into the system dynamics.** We observe that for low levels of noise, the performance is similar to the deterministic case, and we still outperform baselines. When the levels of noise are quite large, it is possible for our performance to further decrease and the baselines to become competitive again. **The full results can be seen in Figure 5 of the updated text.**
>
> ***For stochastic policies, it remains uncertain whether the proposed PPO-based update can guarantee convergence to an optimal policy.***
>
> The reviewer is correct, unfortunately we do not have theoretical guarantees regarding convergence of PPO to an optimal policy. Unfortunately, this is not just a limitation of DOHJ-PPO but of PPO in general, which typically does not come with convergence guarantees.
> Even if one were to obtain such a convergence guarantee, the likelihood of converging to an optimal policy would be small due to the non-convexity of the optimization problem.
>
> On the other hand, the reviewer also suggests a great point: one would indeed like to have some theoretical guarantees when considering optimality with regard to stochastic policies rather than restricting to deterministic policies. We include the following theoretical argument to justify why it makes sense optimize the SRABE (which applies to stochastic policies, so that we can run PPO in the first place) to obtain the optimal value function for the deterministic policy case:
>
> First, pick a discount factor $\\gamma \\in (0,1).$
> Next, given a reward function $r$ and negated penalty $q$, we define the Bellman operator $B_{\\gamma}^{\*}$ by
>
> $$B_{\\gamma}^{\*}\[v\](s) = (1 - \\gamma) \\min \\{r(s), q(s)\\} + \\gamma  \\left\[ \\min\\left\\{ \\max \\left\\{ \\max_{a} v(f(s,a)), r(s)  \\right\\},  q(s) \\right\\} \\right\].$$
>
> We define the value function $V_{\\gamma}^{\*}$ to be the solution to the Bellman equation
> $V_{\\gamma}^{\*} = B_{\\gamma}^{\*}\[V_{\\gamma}^{\*}\]$.
> Proposition 3 in \[Hsu et al., RSS 2021\] then states that $V_{\\gamma}^{\*} \\to V^{\*}$ ($V^{\*}$ here is the true RA optimal value function) as $\\gamma \\to 1^-$.
> Hence we are interested in obtaining $V_{\\gamma}^{\*}$.
>
> Now, for each policy $\\pi$ define the Bellman operator $B_{\\gamma}^\\pi$ by
>
> $$B_{\\gamma}^\\pi\[v\](s) = (1 - \\gamma) \\min \\{r(s), q(s)\\} + \\gamma  \\mathbb{E}_{a \\sim \\pi} \\left\[ \\min\\left\\{ \\max \\left\\{ v(f(s,a)), r(s)  \\right\\},  q(s) \\right\\} \\right\].$$
>
> We define the value function $V_{\\gamma}^\\pi$ to be the solution to the discounted SRABE, i.e. the Bellman equation
> $V_{\\gamma}^\\pi = B_{\\gamma}^\\pi\[V_{\\gamma}^\\pi\]$.
> Notice that from the above definitions, for all states $s$ we have
> $V_{\\gamma}^{\\pi}(s) = B_{\\gamma}^\\pi\[V_{\\gamma}^\\pi\](s) = B_{\\gamma}^{\*}\[V_{\\gamma}^{\\pi}\](s)$. Thus, we have
>
> $$V_{\\gamma}^{\\pi} \\le B_{\\gamma}^{\*}\[V_{\\gamma}^{\\pi}\]\\quad\\text{and}\\quad V_{\\gamma}^{\*} = B_{\\gamma}^{\*}\[V_{\\gamma}^{\*}\].$$
>
>
> Since $B$ is a contraction and is also is monotonically increasing (i.e. $v_1 \\le v_2 \\implies B\[v_1\] \\le B\[v_2\]$), the comparison principle for the Bellman equation $v = B_{\\gamma}^{\*}\[v\]$ applies. In particular,
>
> $$V_{\\gamma}^{\\pi} \\le V_{\\gamma}^{\*}.$$
>
> Thus the SRABE allows us to consider stochastic policies (which we need to run PPO), and by what we have shown above no such policy will not overestimate $V_{\\gamma}^{\*}$.
>
> Now let $\\pi^{\*}$ be a deterministic policy which for each state $s$ produces an action $a$ that maximizes $V_{\\gamma}^{\*}(f(s,a))$.
> It can then be shown that $V_{\\gamma}^{\\pi^{\*}} = V_{\\gamma}^{\*}$.
> So not only will no solution to the SRABE overshoot $V_{\\gamma}^{\*}$, but it is in addition not conservative to search for the optimal policy $\\pi^{\*}$ satisfying the SRABE.
>
> This outline will be added to the appendix in the final version.

---

### Author Response · Authors · 2025-12-03
**Final Comments from Authors (summary for AC)**

First, we would like to express our gratitude to the reviewers and the AC for the time and effort they have invested in evaluating our contributions and helping us to improve this work. We were encouraged by the reviewers’ responses to our results and exposition.

$$\quad$$

The reviewers particularly appreciated our work's:
1. **Interesting problems** -- The “RR and RAA problems are two interesting dual-objective satisfactory problems” [`FUvU`] and “many safe / task-spec RL problems really are ‘reach X and stay safe’ or ‘reach X then Y … [such that] explicit Bellman forms and a working PPO variant for those is useful” [`BuMR`]
2. **Novel, practical theory** -- The “decomposition is novel and practically attractive” [`Cdby`] and “tied directly to implementable Bellman operators” [`BuMR`].
3. **Connections to other work** -- This approach builds-off previous work, “elucidating their connections to the previously studied reach, avoid, and reach–avoid problems” [`FUvU`] and done in a manner that is “clean and provable, allowing modular reuse of existing HJ-RL methods” [`jprC`].
4. **Robustness and generalization in experiments** -- The “experiments span both discrete and continuous control tasks” and our algorithm “consistently outperforms strong baselines, showing robustness and good generalization” [`jprC`].
5. **Clear exposition** -- The work is “clear, easy to follow, and contextualized well… comprehensive yet concise, very useful for readers” [`Cdby`].

$$\quad$$

We have used their feedback to both clarify our work and guide additional experiments and theoretical arguments, which have been added to our draft. Our changes to the manuscript include additions related to questions on:

1. **How our algorithm would perform under stochastic dynamics** -- We performed ablation experiments in which an increasing an amount of noise was added to the system, and compared our performance with top baselines (Figure 5). The results show that our algorithm outcompetes all tested baselines under low-to-moderate noise levels, while some baselines begin to perform competitively with our method for high amounts of noise.
2. **How our decompositions relate to decompositions in Linear Temporal Logic (LTL)** -- We added simple counter-examples which clarify that LTL-based decompositions fail to translate in the safe optimal control case (Section L in the Appendix). For this latter class of problems, our specialized decompositions are required instead.
3. **What motivates/justifies the use of the Stochastic Reach-Avoid Bellman Equation (SRABE)** -- We clarified that we introduce the SRABE to generalize the Stochastic Reachability Bellman Equation introduced in (So et al. 2023). We also provided a new theorem for the SRABE and SRBE (Theorem 4 in Section D of the Appendix), now justifying its use both theoretically and empirically.
4. **Whether state augmentation is needed because rewards and penalties are not Markovian** -- We made explicit in the text that all rewards and penalties in our work are deterministic, and thus Markovian. We clarified that instead, state augmentation is used to achieve optimal performance for the dual-objective problems (rather than discounted sum-of-rewards).
5. **Related works** -- One reviewer suggested including certain works on non-traditional RL objective functions. We added their suggested references to the Related Works section.
6. **Future work** -- Some reviewers were curious about extensions to higher-level task specifications and scaling. We have now added further discussion to the Conclusion on future directions for / extensions of this work.

---

### Meta-Review · Area_Chair_ExgA · 2026-01-07

**Summary:**

The authors provide a mathematically rigorous framework for Reach-Always-Avoid (RAA) and Reach-Reach (RR) problems by elegantly connecting Hamilton-Jacobi optimal control theory to reinforcement learning through sound Bellman forms via decomposition.

The reviewers seem to have reached a consensus that the paper introduces a novel and clean decomposition into modular subproblems that enhances compositionality and eliminates the delicate parameter tuning often required by traditional Lagrangian-based methods. The presentation is clear and well-contextualized, offering a comprehensive literature review and experiments that bridge the gap between theoretical findings and practical implementation. Empirical evaluations across diverse environments demonstrate that the proposed PPO-based method outperforms baselines.

**Reviewer Concerns:**

Most reviewers concern seem to have been addressed during the rebuttal period. The changes have been succinctly summarized by the authors in their final comment. I do not think there are any significant concerns that remain outstanding.

**Reviewer Scores:**

None of the reviewers are likely to have changed their score. There is consensus that the paper meets the bar for publication, without being outstanding. Some of the suggestions for improvement would have made the paper stronger but are not strictly required to meet the threshold for acceptance. For example, an extension of the method to solve all TL specifications using HJR would significantly improve the paper, but the authors have chosen to frame this as an avenue for future work.

---

### Decision · Program_Chairs · 2026-01-26

Accept (Poster)